# Fragmentation and multithreading of experience in the default-mode network

Fahd Yazin [1] ✉, Gargi Majumdar [2], Neil Bramley [1] & Paul Hoffman [1]

Reliance on internal predictive models of the world is central to many theories of human cognition. Yet it is unknown whether humans acquire multiple separate internal models, each evolved for a specific domain, or maintain a globally unified representation. Using fMRI during naturalistic experiences (movie watching and narrative listening), we show that three topographically distinct midline prefrontal cortical regions perform distinct predictive operations. The ventromedial PFC updates contextual predictions (States), the anteromedial PFC governs reference frame shifts for social predictions (Agents), and the dorsomedial PFC predicts transitions across the abstract state spaces (Actions). Prediction-error-driven neural transitions in these regions, indicative of model updates, coincided with subjective belief changes in a domain-specific manner. We find these parallel top-down predictions are unified and selectively integrated with visual sensory streams in the Precuneus, shaping participants' ongoing experience. Results generalized across sensory modalities and content, suggesting humans recruit abstract, modular predictive models for both vision and language. Our results highlight a key feature of human world modeling: fragmenting information into abstract domains before global integration.

In our lives, we encounter a wide range of situations with complex and ever-changing properties – spatial, temporal and social. To understand and predict future events, we form internal models of our experiences. A well-tuned internal model[1–4] allows us to interact with the world efficiently by generating accurate future predictions[5,6]. Understanding how these models are structured and represented are central question in cognitive neuroscience. A hallmark of cortical computation is the prevalence of functionally specialised modules geared towards processing specific kinds of information[7,8]. This may be because different types of environmental variables require different kinds of inductive biases[9] for efficient computation or because the computations themselves differ. For example, relational properties like reference frames are better captured by graph structures, whereas temporal characteristics of action sequences might require a more sequential structure. Different domains, thus, may necessitate different prior constraints, and successful generalisation requires getting these constraints right[10]. Organising world knowledge efficiently[11–13] (e.g.,

cognitive maps) might therefore require a modular approach to internal model construction, leveraging domain-appropriate inductive biases. Introducing such modular principles has also been shown to enhance artificial learning systems[14,15]. Yet it is unknown the extent to which humans use an assembly of many separable models, each specialised for different kinds of prediction[2,16,17], versus a single unified model. Modular representation of internal models through parallelisation of domains would, however, create a coordination problem: how are the contents of these distinct models unified to provide coherent behaviour, let alone our integrated experience of the current properties of the world[18]?

There are computational reasons to presume humans possess multiple distinct cognitive maps[11] or model spaces[19], since this allows each to be specialised for particular kinds of inference. In this study, we introduce three such models, tuned to different aspects of the world: states, agents and actions (illustrated in Fig. 1a). First, navigating a complex world requires an accurate representation of one's current

[1]School of Philosophy, Psychology & Language Sciences, University of Edinburgh, Edinburgh, UK. [2]Biological Psychology and Neuropsychology, University of Hamburg, Hamburg, Germany. ✉e-mail: fahd7yazin@gmail.com

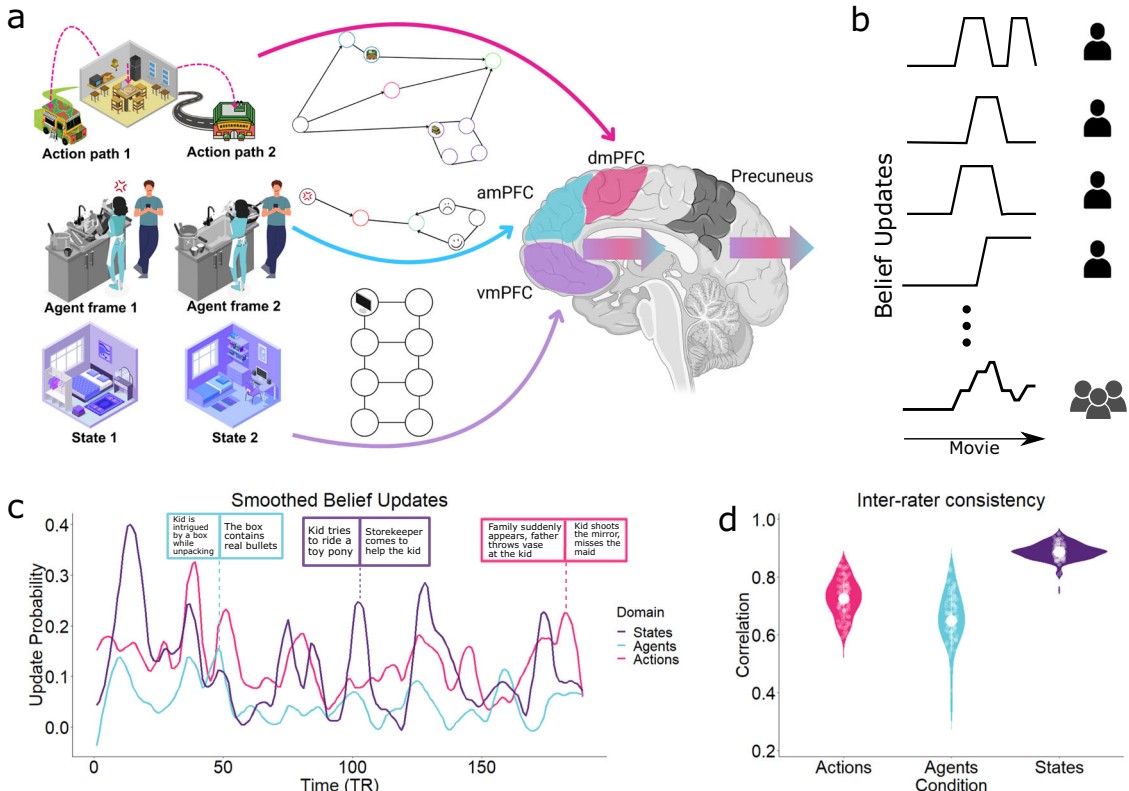

**Fig. 1 | Fragmentation of World Models. a** Any experience can be divided into models of states (abstract contexts), agents (others' beliefs/goals), and actions (temporal paths through state space). The midline prefrontal cortex can be viewed as assembling such a model space, configuring the best model for each world abstraction. It generates top-down predictions in a tripartite organisation, with domain-specific belief updates recruiting each region selectively. Consider visiting a friend's home for lunch. You walk into what you assume is the bedroom, only to discover a fully equipped work PC. The context then updates from "bedroom" (State 1) to "home office" (State 2). You observe the friend's apparent unhappiness during cleaning, possibly due to not offering help (Agent frame 1), changing your representation of their mood (Agent frame 2). Consequentially you consider alternative dining options than eating in, e.g., grabbing food from a nearby food truck (Action path 1) or restaurant (Action path 2). The experience itself appears fused, but its deeper compositionality is implicit in the narrative structure of human experience (and later memories). The authors have applied the CC-BY 4.0 license to illustrations generated by Biorender: Created in BioRender. Yasin, F. (2025) https://biorender.com/f0o8ol8. **b** Design schematic for obtaining belief update time-courses by aggregating reported updates over multiple participants. **c** Smoothed, group-level belief update time-courses peaking when participants signalled their predictions were being updated in each domain. (Inset) Movie scenes as text descriptions depict different domain-updates (Due to copyright restrictions of the movie "Bang! You're Dead", the screenshots are replaced with text descriptions of the scene here; participants watched the full movie inside the scanner). **d** Interrater reliability in update time-courses computed through split-half correlation (100 times) for State ($n = 18$), Agent ($n = 21$) and Action ($n = 19$) updates. Error bars denote the standard error of the mean.

state. However, it may not be possible to obtain (or observe) all the variables required for this, requiring strong background information (e.g., memory). A mapping between prior knowledge and observed sensory information underlies the inference of the environmental state. These states provide abstract contexts to situate events and are crucial for accurate future state predictions and learning; their disruption can bias inference[20].

Representing these contextual State models seems necessary but insufficient for a full world model. This is because states are populated by other people (or generally agents) who each possess a different reference frame, and thus have distinct goals and perspectives. Modelling these reference frames is crucial to reasoning about the mental states, beliefs and intentions of other agents, ourselves and any interactions. These Agent models may then be represented quite separately to state representations. This facilitates perspective-taking and simplifies joint inference across various combinations of states and agents. In group settings, that form a large portion of human life, an accurate representation of relational properties of each agent to oneself and others are key.

Third, for a given State and Agent-reference frame, the space of (abstract) transitions or paths one might take through them is vast. Thus, a separable representation of temporally abstract actions, or Action models, allows the mapping of previously learned action paths onto newly learned states or agents. In sum, we suggest three core abstract domains of cognitive representation are needed – States, Agents, and Actions. Each operates on different sources of information about the world. They depend on one another, but generate distinct predictions about the unfolding environment. The errors in each domain demand a fundamentally different kind of update, lending itself to a modular architecture. Thus, a neural specialisation for these domains allows fast and flexible inference[7,8] to the near-infinite permutations of context, people and plans that we encounter in our lives.

Different sectors of midline prefrontal cortex are sensitive to different features of the world, making this cortical territory a likely site for specialised world models[21] (Fig. 1a). State model estimation has been extensively shown to be centred around the ventromedial prefrontal cortex[22], which encodes cognitive maps[23] or low-dimensional schemas[24]. This region is also associated with reward processing; however, recent perspectives suggest these effects may stem from the more general function of state estimation[25]. Sitting dorsal and anterior to vmPFC, the anteromedial prefrontal cortex is involved in social cognition, theory of mind[26,27], social hierarchy learning[28] and goal processing[29]. All of these activities require referential modelling[30,31] and computing goals (self/others). Further dorsal and posterior, the

dorsomedial prefrontal cortex is critical for (high-level) action planning[32], strategic decisions[33], and formulating hierarchical plans[34], all of which fall under the notion of modelling temporal properties over longer time-scales[35]. Taken together, it is appealing to position this triumvirate of regions as the core model space within the midline prefrontal cortex. These also form the anterior nodes of the Default-Mode Network (DMN), which is thought to process internal models of experience[12,13,16,18,36,37].

As mentioned above, a modular architecture creates the challenge of integrating different predictions together into a unified format and merging top-down priors with sensory data. The Precuneus, the core node of DMN[38] is in a strategic position to meet this demand due to its hypothesised role in global integration[39], interfacing with other cortical networks and being a sensory hub[18]. Lesions to this region often lead to integratory deficits[40,41]. We propose that the Precuneus is where distinct prefrontal predictions relating to states, agents and actions are combined. Integrating these with sensory data allows the brain to maintain a coherent, unified, and up-to-date model of its physical and social environment. In other words, the evolving unified experience may be integrated and maintained here, while fragmented modular models persist in the PFC.

In this study, we test this proposal: that the midline PFC regions operate as a partitioned domain-specialised, tripartite model space. This system generates top-down predictions from each model in a parallel, independent manner. We focus on prediction errors; moments when an aspect of one's current world model proves inaccurate and is thus updated. One would expect brain regions supporting a particular domain or prediction to show increased activity during such updates. Folding this notion onto three distinct models means different kinds of errors trigger updates to unique parts of the world model. We hypothesise that these predictions are integrated in the Precuneus, contributing to a unified representation of one's current environment. This process may underlie the subjective experience of the world, which we operationalise in this study by measuring participants' moment-to-moment arousal levels.

We explore three fundamental questions about internal model representation using fMRI data collected while participants watched a short movie, where all three domains are intermingled. We collected behavioural data from a different sample of participants watching the movie to determine when people generally updated their beliefs about states (movie situations), agents (movie characters) and actions (what transpired). Using these updates as predictors of neural activity, we investigated whether humans possess a single global representation of the world model or a modular, domain-specific organisation. Next, through hidden Markov modelling, we explored whether region-specific neural transitions coincided with subjective belief changes. Third, we tested whether these domain-specific predictions are integrated within core DMN regions by analysing shared connectivity profiles. Finally, we replicated our key findings in a second dataset that involved a different sensory modality (a spoken story) and level of emotional content. Our results specifically outline how the human prefrontal cortex performs domain-specific world modelling and, more generally, how the DMN integrates these to shape our subjective experience.

## Results

Our approach (Fig. 2) was to use fMRI data from young participants ($n = 111$) passively watching a short movie "Bang you're dead", taken from the Cam-CAN project[42]. We obtained continuous belief-update (prediction updates) time courses for this movie from separate groups watching online and used this for our analysis of the neuroimaging participants' BOLD data. Later, we generalise the main results to a separate cohort listening to a story[43] ($n = 52$) while undergoing scanning, using the same methods.

We performed a number of analyses designed to identify the neural correlates of belief updates in the domains of states, agents and actions, and to investigate the within- and between-region neural dynamics associated with them. Each of these analyses is described in more detail later, but we begin with a general overview. First, we used update probability time-courses as predictors of BOLD response, to identify regions that show heightened activity during updates of each type (Fig. 2b). Next, we isolated the most significant update moments in each domain by thresholding and thus binarizing the update time-courses (Fig. 2c). We then used Hidden Markov Modelling to test whether these domain update moments coincide with transitions in the neural states of specific PFC regions. This analysis tested whether the transient activity increases observed during model updates were associated with sustained shifts in the activation states of the region, specific to each domain. We also performed a number of analyses based on inter-subject correlations (ISC; Fig. 2d). The rationale of ISC analyses is that when multiple participants are scanned while experiencing the same stimulus, neural responses that are consistent (i.e., correlated) across participants reflect reliable brain responses to the content of the stimuli[44]. We tested ISC in the activity of PFC regions to determine whether correlation increased during a region's preferred updates, indicating heightened stimulus-driven activity. Next, we used a related method, inter-subject functional connectivity, to investigate whether stimulus-driven PFC-Precuneus activity correlations were increased during belief updates, compared to other regions and non-updates. And finally, we used inter-subject pattern correlations to quantify the alignment of multi-voxel activation patterns between participants, and test the degree to which these pattern time-courses of PFC and Precuneus displayed representational alignment at similar times.

### Measuring belief updates in state, agent and action models

Our analysis approach is to identify changes in neural activity, patterns and connectivity and assess whether these coincide with subjective markers of internal model updates of movie content in each domain. We first probed when belief updates typically occurred in each domain, during the movie. Participants from three independent groups watched the movie and pressed a button whenever they felt their beliefs changed in one of the three domains (Fig. 1b). Briefly, these were contextual updates[45] (for States; $n = 18$), belief updates about people (for Agents; $n = 21$) and belief change due to an action taken, that could affect the trajectory of the movie (for Actions; $n = 19$) (see Supplementary Table 1 for complete instructions). We assume that people signal a belief update whenever their existing model of the movie mismatches incoming information (i.e., after prediction errors occur). Update time-courses from individual participants were combined and smoothed to give a continuous time-course (Fig. 1c). Higher values indicate a greater proportion of individuals watching the movie marked a belief update at that point. These time courses indicate that the predictions (and the experience) of each domain fluctuate considerably throughout this movie. By collecting these ratings in separate groups of participants to those who provided the fMRI data, we ensured that the fMRI data reflected processing of a fully naturalistic experience, free from instructional, meta-monitoring and responding effects. Group-level ratings were only weakly correlated with one another (States vs Agents, $r = 0.31$, Agents vs Actions, $r = 0.29$, Actions vs States, $r = 0.28$). This indicates that models in different domains were being updated at different times during movie-watching. In addition, we performed mutual information and cross-correlation analysis on the ratings, which indicate only weak associations between different domains (Supplementary Figs. 2, 3). We also conducted a split-half correlation analysis for each rating with the Spearman-Brown correction applied, which indicated generally good levels of agreement between participants in the timing of updates (Fig. 1d) (States $r = 0.88$, Agents $r = 0.65$, Actions $r = 0.72$, see "Methods").

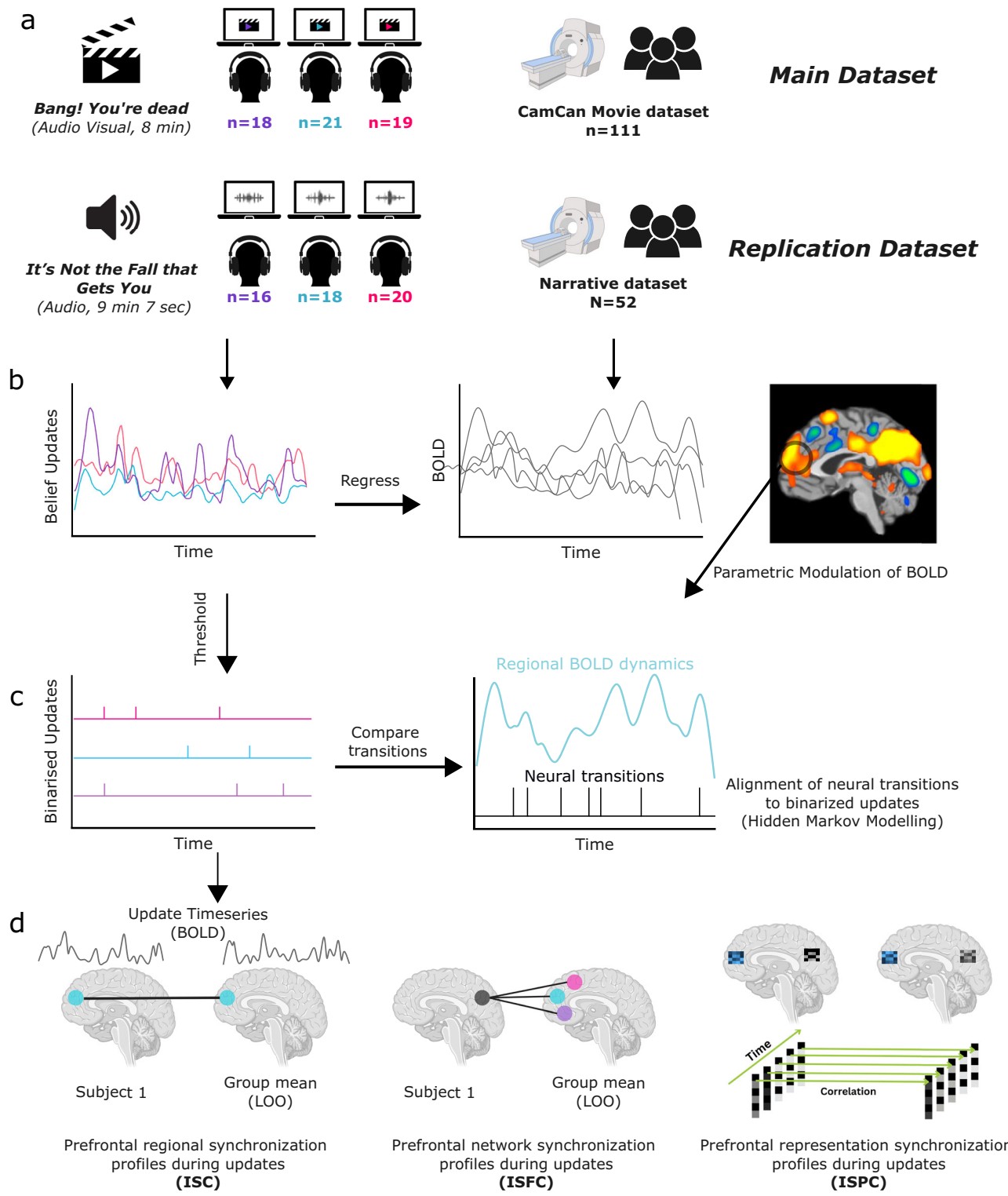

**Fig. 2 | Illustration of analysis pipeline. a** Distinct rating and scanning group watched a movie (main dataset) and a story (replication) dataset. The authors have applied the CC-BY 4.0 license to illustrations generated by Biorender: Created in BioRender. Yasin, F. (2025) https://biorender.com/438by9r. **b** Continuous measures of domain-specific belief updates were used to predict BOLD across the brain (GLM with parametric modulation), identifying regions that show increased activity during periods with a high likelihood of model update. **c** Belief update probabilities were binarised to identify update points, which were compared to transitions in neural dynamics in each ROI and (**d**) later used for investigating shared response properties during update events using intersubject analyses. The authors have applied the CC-BY 4.0 license to illustrations generated by Biorender: Created in BioRender. Yasin, F. (2025) https://biorender.com/w149eob.

## Topographically distinct internal models in the midline prefrontal cortex

Our neuroimaging analysis began by identifying activations that were parametrically modulated by each domain-specific belief update.

Updates involve revision and reconfiguration of the current internal model. Thus, these are periods where we would expect heightened processing demand when regions representing domain-specific model content should show increased activity. To test this, all three

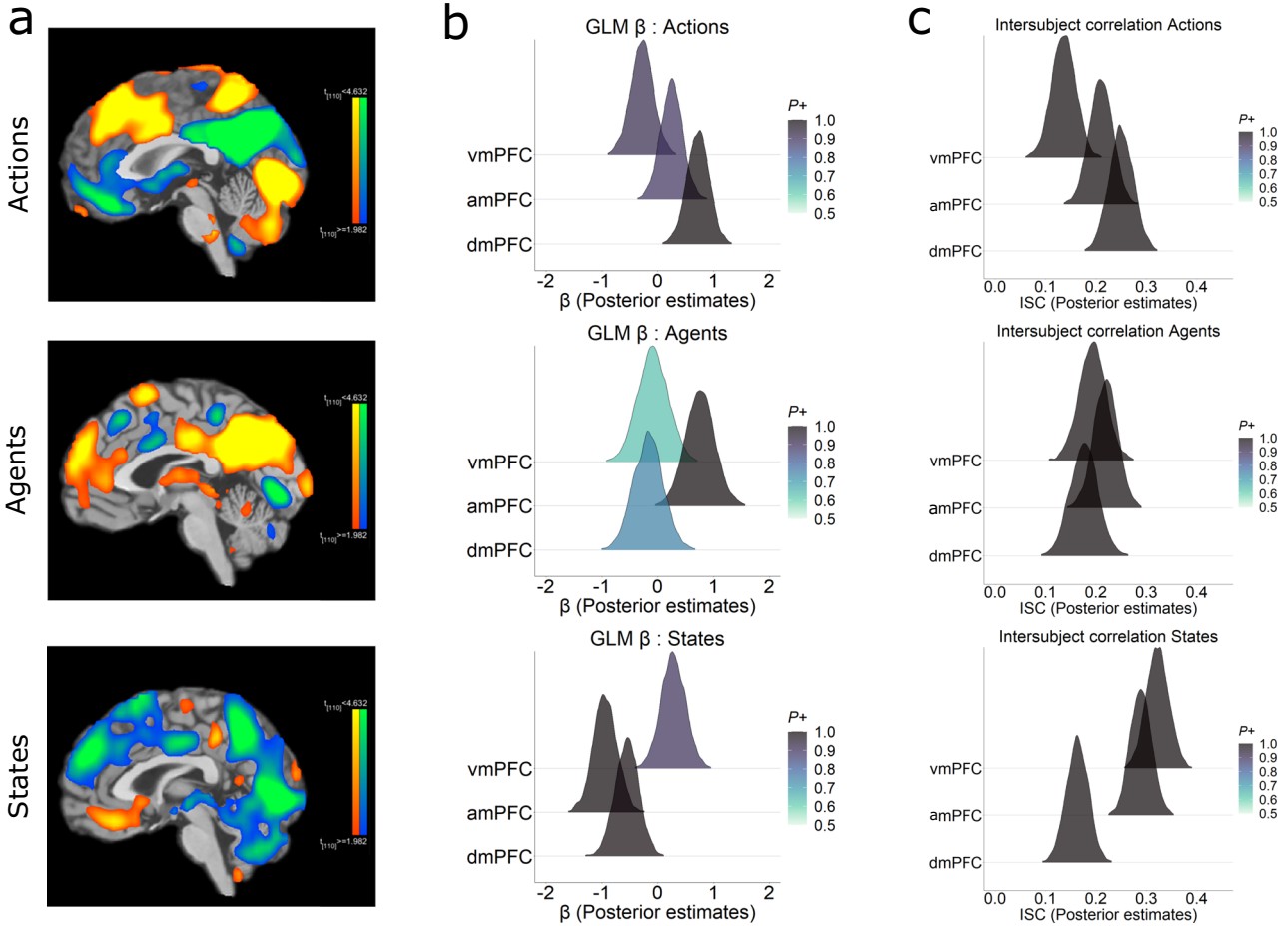

**Fig. 3 | Fragmentation of Predictions in Prefrontal Cortex. a** Whole brain maps ($p < 0.05$ FDR corrected) show a topographically distinct segmentation in the midline prefrontal nodes responding to revisions of predictions in Actions (top), Agents (middle) and States (bottom) domains. Warm colours indicate positive effects for each domain relative to the other two, and cool colours denote negative effects. **b** Posterior distributions of an ROI analysis (Bayesian regression) confirming a domain-specificity within these regions. **c** Posterior distributions of an Intersubject correlation (ISC) reveal that, in each region, activity is more synchronised across participants during updates in that region's preferred domain.

smoothed update probability time courses (Fig. 1c) were used to simultaneously predict neural activity. We then contrasted the effects of each domain against the other two, allowing us to tease apart whether the regions were particularly involved in each type of belief update.

Whole brain maps (Fig. 3a) revealed a topographically distinct activation profile in the midline prefrontal sector. vmPFC activity was most strongly correlated with State updates, amPFC with Agent updates and dmPFC with Action updates, in line with our hypothesised model space. A further ROI-level Bayesian analysis (Fig. 3b and Supplementary Fig. 6) was conducted on the effects of each predictor (beta values) in each region, using anatomically defined ROIs derived from the Brainnetome atlas[46] (Supplementary Fig. 1). We used a hypothesis-driven Bayesian hierarchical regression[47,48] to test this (see "Methods").

In State updates, beta values showed strong evidence in favour of vmPFC showing higher activation than dmPFC (vmPFC > dmPFC Estimate = 0.83, 95% CI 0.57-1.09, BFfor > 150, $P = 0.99$) and amPFC (vmPFC > amPFC, Estimate = 1.22, 95% CI 0.96–1.48, BFfor > 150, $P = 0.99$). Similarly, for Agent updates, amPFC had more activity than dmPFC (amPFC > dmPFC, Estimate = 0.93 95% CI 0.66 − 1.18, BFfor > 150, $P = 0.99$) and vmPFC (amPFC > vmPFC, Estimate = 0.84, 95% CI 0.57 − 1.12, BFfor > 150, $P = 0.99$). Finally, in Actions updates, there was evidence for dmPFC being higher than vmPFC (dmPFC > vmPFC, Estimate = 1, 95% CI 0.74 − 1.24, BFfor > 150, $P = 0.99$) and amPFC

(dmPFC > amPFC, Estimate = 0.47, 95% CI 0.23-0.72, BFfor > 150, $P = 0.99$). These results suggest a topographically distinct pattern of effects within the PFC for different kinds of model updates. That is, different PFC regions responded to different types of prediction updates during naturalistic experience. The scanning cohort had no instructions to watch the movie in any special way. Yet they showed our predicted separation in the PFC when revisions of their beliefs occurred (as indicated in the ratings of independent groups of participants).

## Domain-specific increase in Prefrontal Shared Activity during prediction updates

The activation effects suggest that specific PFC regions show heightened processing in response to domain-specific belief updates. In naturalistic neuroimaging paradigms, synchronisation in the temporal profile of activity across participants is often used as evidence that a region is engaging in stimulus-driven processing[44]. If these regions are indeed generating top-down predictions, then periods of belief update should also have particularly high levels of synchronisation, i.e., increased shared response across participants. This is because all participants should update or calibrate their predictions in response to events in the movie in similar ways.

To tackle these arguments rigorously, we took a principled approach to the update ratings that stayed constant in all further analyses unless specified otherwise. We identified a set of updating

moments in each domain by applying a threshold to the group-averaged time courses shown in Fig. 1c. Specifically, we assumed an update occurred whenever the update probability exceeded 2 SD of its mean level (see Methods for rationale). The underlying logic being, since model updates are a subjective judgement, there is no consensus as to what exactly is an objective update (no matter how many raters are present). At the same time, an update that is most prevalent across raters has the highest likelihood to be present in the experience shifts of the movie-watching cohort. This ensures that the 'base rate' of updates for each domain is respected and only those updates that were widely shared across the cohort are analysed.

We then calculated the intersubject correlation (ISC) values[44] for these update periods, in our three PFC regions. ISC values were obtained by constructing a 7 TR (scan) window around these time points. These update segments were concatenated and ISC computed on this one long segment, for each participant.

We predicted that in each PFC region, ISC would be higher during its preferred domain's updates relative to other domains. We used a hypothesis driven Bayesian hierarchical regression[47,48] to test this (see "Methods").

We found strong evidence (Fig. 3c) for these regions being synchronised in a domain-specific manner across participants. In State updates, ISC showed strong evidence in favour of vmPFC showing higher ISC than dmPFC (vmPFC > dmPFC Estimate = 0.16, 95% CI 0.11-0.21, BFfor > 150, $P = 0.99$) and amPFC (vmPFC > amPFC, Estimate = 0.03, 95% CI $-0.01 - 0.08$, BFfor = 7.35 $P = 0.88$). Similarly, in Agent updates, amPFC had more ISC than vmPFC (amPFC > vmPFC, Estimate = 0.03 95% CI $-0.02 - 0.07$, BFfor = 4.81, $P = 0.83$) and dmPFC (amPFC > dmPFC, Estimate = 0.04, 95% CI $0 - 0.09$, BFfor = 15.39, $P = 0.94$). Finally, in Actions updates, there was evidence for dmPFC being higher than vmPFC (dmPFC > vmPFC, Estimate = 0.11, 95% CI $0.06 - 0.17$, BFfor > 150, $P = 0.99$) and amPFC (dmPFC > amPFC, Estimate = 0.04, 95% CI $-0.01$-0.09, BFfor = 12.02, $P = 0.92$). In addition, we repeated the ISC analysis over a range of windows to assess robustness to window size (Supplementary Fig. 22).

These results indicate a topographically tripartite profile in midline PFC that showed domain-specific increase in activation and shared-response during moments of stimulus- driven belief updates.

## Prefrontal neural transitions coincide with experienced belief updates

The previous analyses revealed increased activity when participants updated their predictions and consequently showed an increased shared response in the PFC. Updating one's current internal models would involve rebuilding the model representations on the fly, leading to newer interpretations and experience of the ongoing stimuli. If such a model rebuilding occurs, then each update should also be associated with sustained shifts to the prefrontal neural dynamics of the ongoing experience. The signatures of this could be extracted from the BOLD latent dynamics/transitions. To test this, we used Hidden Markov models (HMMs) to identify transitions in the neural activation states of each PFC region. We then tested how well these transitions aligned with belief updates in the domains of State, Agent and Action.

Hidden Markov models are well-suited to tackling such problems in cognitive neuroscience and has been successfully applied onto naturalistic stimuli[12,13,48]. We deployed an HMM (see "Methods") designed to be highly resistant to the timing of participant update variability. We did this by repeatedly model-fitting across subsets of participants over a range of possible latent states and selecting the most statistically efficient number of states. The final HMM, configured with this number of states, was then estimated multiple times, averaging the latent state transition time points, producing a transition time-course for each ROI. We only used the most reliable and consistent update points for a rigorous comparison (see "Methods"). We aimed to identify which type of belief update most closely aligned with

neural state transitions in each PFC region. To do this, we counted the belief updates occurring immediately after a neural transition, using various TR/scan window sizes (Fig. 4a).

Figure 4b shows the proportion of belief updates that occurred immediately after neural state transitions in each PFC region, for a range of temporal window sizes. Strong domain- specificity was observed. vmPFC transitions captured State updates more than Agent or Action updates, amPFC was most attuned to Agent updates and dmPFC showed a preference for Action updates. These effects were largely consistent across the size of the temporal windows used.

These results emphasise that shifts in the prefrontal neural dynamics coincided with updated model predictions. Three separate internal models in the prefrontal cortex appear to coincide with these shifts in three key domains during unguided naturalistic experience.

## Precuneus selectively integrates updated prefrontal representations

So far, we have provided evidence consistent with our hypothesis that internal world models relating to States, Agents and Actions are represented in distinct regions of PFC. How do prediction threads from these simultaneous yet spatially distinct systems get integrated and distributed globally?

There are two distinct subproblems here. First, prefrontal model contents must be unified to form a global representation of the current state of the world. This can occur within the PFC, or outside it. Second, these integrated representations must be used to constrain processing of incoming sensory information in other parts of the cortex.

Functional integration of updated predictions across domains may happen in several ways. First, a decentralised manner within the Prefrontal cortex, suggesting coordination driven by anatomical proximity. Second, centralised integration with the Hippocampus, leveraging its extensive connectivity[49]. Third, network integration in DMN[18,50], with the Precuneus playing a pivotal role due to its integrative dynamics[18,38,40] and network centrality[39].

To explore these potential integration mechanisms, we utilised intersubject-functional connectivity[50,51] (ISFC) analysis between domain-specific PFC regions and other ROIs during belief updates. ISFC is particularly effective in naturalistic paradigms as it removes stimulus-unrelated connectivity influences. This allowed us to measure and compare coupling strengths during model updating within the above three hypothesised ways. Using a similar approach to the earlier ISC analysis, we assessed the correlation between each PFC region and other regions during update periods for its preferred domain (e.g., during State updates for vmPFC). We correlated each PFC region with (a) Hippocampus, (b) Precuneus (PCN) and (c) other PFC regions. Employing a Bayesian hierarchical regression, we then compared posterior evidence for each coupling hypothesis: within-PFC, PFC-Hippocampus and PFC-PCN.

We found evidence (Fig. 5a, left column) for higher functional coupling between PFC nodes and the Precuneus than the other two forms during updates. State updates showed vmPFC having more connectivity with PCN than the other regions (PCN-vmPFC > HPC-vmPFC Estimate = 0.03 95% CI $0.01 - 0.05$, BFfor = 216.39, $P = 0.99$ & PCN-vmPFC > within-PFC Estimate = 0.2 95% CI $0.18 - 0.22$, BFfor > 1000, $P = 0.99$). Similarly, in Agent updates, amPFC showed more evidence of coupling with PCN (PCN-amPFC > HPC-amPFC Estimate = 0.06 95% CI $0.04 - 0.08$, BFfor > 1000, $P = 0.99$ & PCN-amPFC > within-PFC Estimate = 0.16 95% CI $0.14 - 0.18$, BFfor > 1000, $P = 0.99$). Finally, Action updates had dmPFC displaying more synchrony with PCN (PCN-dmPFC > HPC-dmPFC Estimate = 0.08 95% CI $0.06 - 0.1$, BFfor > 1000, $P = 0.99$ & PCN-dmPFC > within-PFC Estimate = 0.09 95% CI $0.07 - 0.11$, BFfor > 1000, $P = 0.99$). Moreover, the model could account for a fairly large variance in the data (Bayes adjusted $R^2 = 0.85$).

Importantly, this coupling between PFC and PCN was much more during the updates than during nonupdated events (Fig. 5a, right

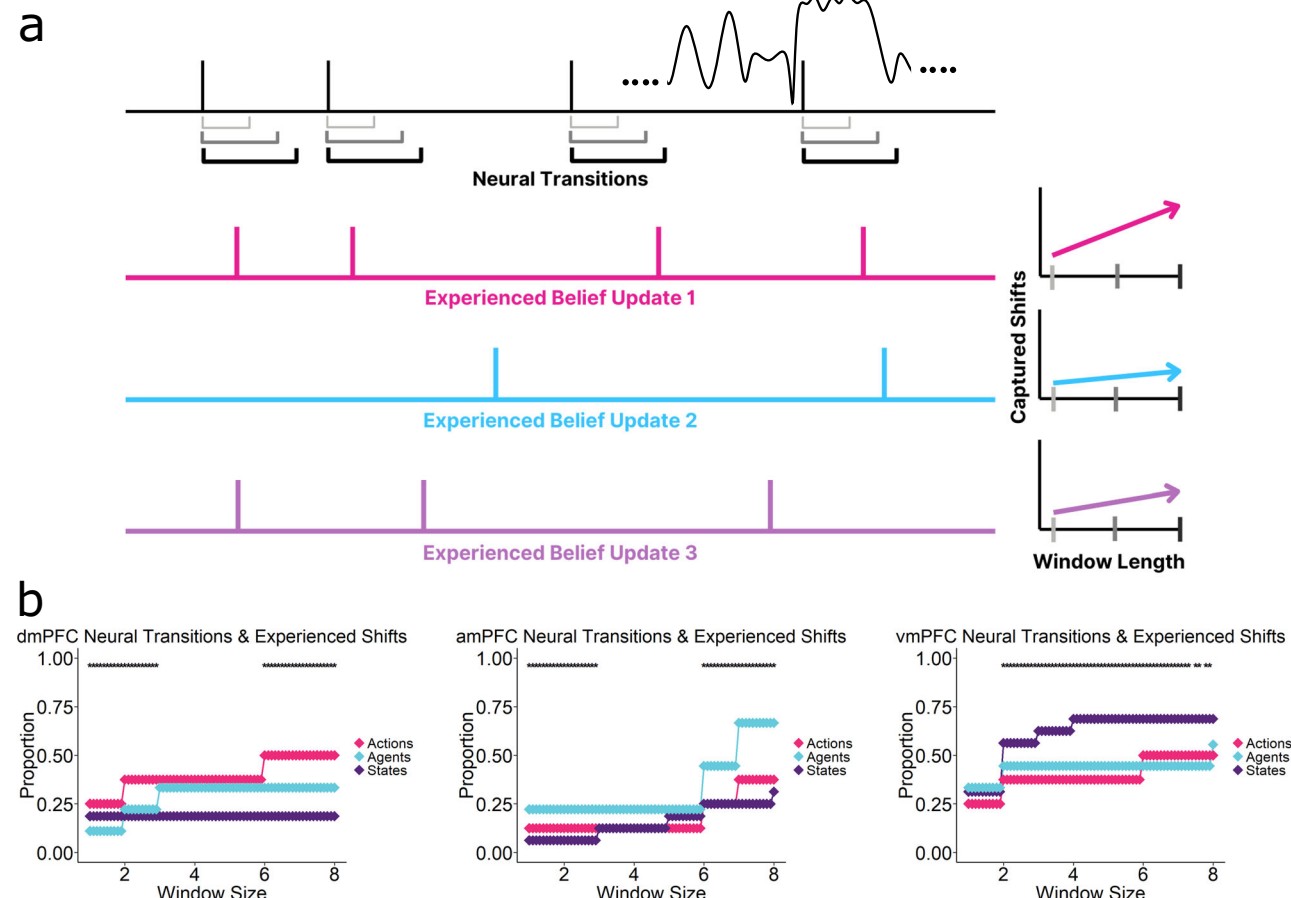

**Fig. 4 | Neural Transitions Coincide with Subjectively Experienced Belief Updates. a** Using Hidden Markov Models, we identified neural transition points within an ROI during the movie (top row). We then compared the timing of these transitions with the belief updates by counting how many updates occurred within different time windows (TRs/scans) following these neural transitions. This varied from 1 to 8 scans. As the window length increases (light grey to black), more belief updates are naturally included. However, the critical prediction is that each region's transition windows will contain more updates from its preferred domain than from the other domains. **b** Proportion of belief updates that fall within neural transition windows in each PFC region, for various window sizes. (Left) dmPFC neural transitions are most closely aligned with Action model updates, while amPFC (middle) and vmPFC (right) transitions capture more Agent and State updates, respectively. Statistical significance was assessed via a permutation test on phase-randomised neural time-series. (Asterisks denote window sizes whose observed statistic exceeded the 95th percentile of its permutation-null distribution ($p < 0.05$, one-tailed test)).

column). This is crucial to rule out since these regions are part of the DMN, and thus can exhibit generally high functional coupling. We find that States had the highest difference in update times, having more ISFC values than NonUpdate points (Update > Nonupdate Estimate = 0.36 95% CI 0.30 − 0.42, BFfor > 1000, $P = 0.99$). This was followed by Actions (Update > Nonupdate Estimate = 0.09 95% CI 0.03 − 0.15, BFfor = 216.39, $P = 0.99$), and less by Agents (Update > Nonupdate Estimate = 0.04 95% CI −0.04 − 0.12, BFfor = 4, $P = 0.80$). Overall, there was more evidence for Update events having more coupling within the core DMN than non-update points.

The above results suggest prefrontal representations are preferentially coupled with the Precuneus in all the updates. Interestingly, the Hippocampus also showed more coupling with the vmPFC (States) than the other two PFC regions, suggesting greater hippocampal involvement in more contextual, State-related representation. Strong PFC-Precuneus coupling is perhaps not surprising given the usually high connectivity between the PFC and Precuneus within DMN. However, it is still unknown how exactly this integration is carried out within the DMN. Specifically, in our design, how are these updated top-down representations unified and integrated with sensory data? The Precuneus is in a position to achieve this due to its anatomical proximity with visual cortex and functional coupling with the prefrontal sectors.

Given the parallel nature of the domains, the Precuneus likely has access to all running prefrontal prediction threads. This is helpful if sensory evidence required only one domain to be updated, while (mostly) keeping the other two untouched. For instance, upon finding a restaurant closed, one might update the action strategy without changing the reference frame of wanting food. We hypothesise that integration in the Precuneus follows similar logic, prioritising updates to representations within the relevant domain of the PFC. Since this involves accessing and switching between multiple running prediction threads, we term this Multithreaded integration. This interpretation allows us to directly test if such a form of functional integration is occurring on the neural representational level during the updates.

To test this, we utilised intersubject spatial pattern correlation[52] (ISPC). This measure is the spatial equivalent of the ISC measure used earlier. It indexes the degree to which the pattern of activation across the voxels in a region is similar across participants (Fig. 5b). By computing ISPC in the Precuneus for each TR, we can construct a time course of shared patterns across participants. High ISPC values indicate points during the movie when (stimulus-driven) neural representations are similar across participants. This similarity suggests that regions are actively engaged in stimulus-driven processing, such as updating beliefs about movie content. To determine whether the Precuneus is involved in global integration across domains, we

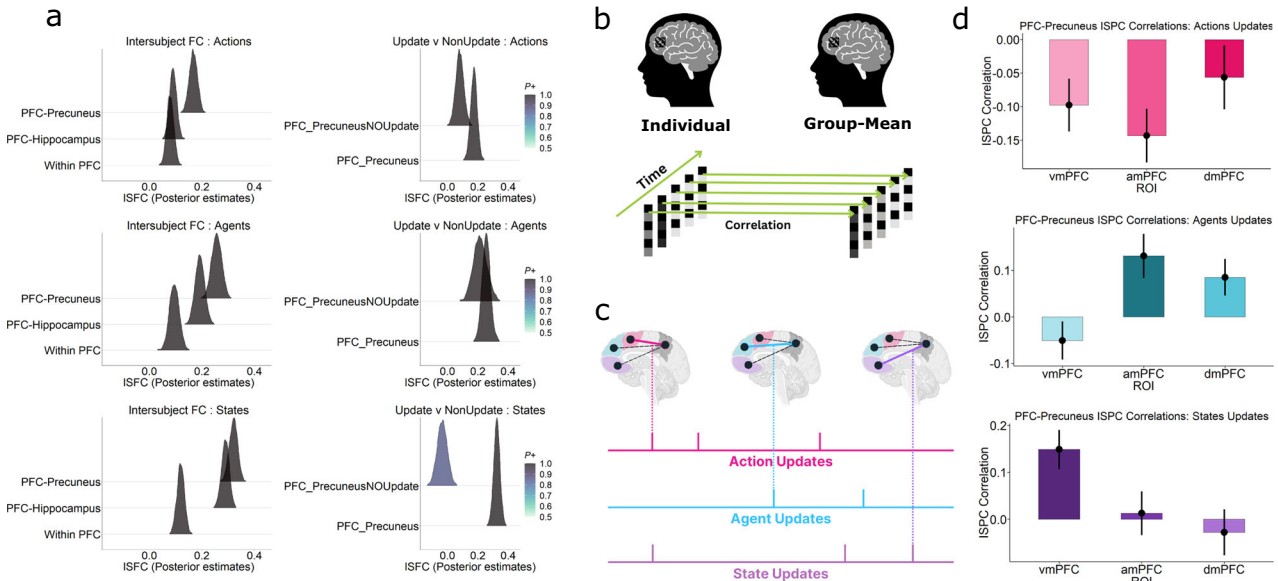

**Fig. 5 | Prioritised Integration of Prefrontal Predictions within the Precuneus.** **a** Posterior estimates of different forms of functional integration. (Left column) The Precuneus had more shared functional integration (ISFC) during the belief updates with each of the domain-specific PFC regions. Bayes factors showed more evidence (see text) for such a within-DMN functional integration than an integration within the PFC (Within PFC) or with the Hippocampus (PFC-Hippocampus). (Right column) This connectivity was more with each of the PFC subregions during update, compared to time points without belief updates. **b** Pattern-level Integration. Intersubject pattern correlation (ISPC) was computed by correlating voxel patterns at each time point in the movie across participants. This provides a time course of shared patterns for different ROIs. **c** During moments of updates, pattern representations in the PFC suggest new predictions. An increased correlation of the time-course between a PFC region and another ROI (here, Precuneus), during the

update, suggests functional coupling on the representational level. If an ROI has high similarity selectively with each of the PFC regions for its domain update, then it integrates new prefrontal representations in a prioritised, multithreaded manner. Threading here refers to switching between multiple prefrontal prediction threads. The authors have applied the CC-BY 4.0 license to illustrations generated by Biorender: Created in BioRender. Yasin, F. (2025) https://biorender.com/f0o8ol8. **d** Bar plots show the correlation strength of each region's ISPC time course during updates with that of the Precuneus ($n = 111$). (Top) dmPFC displaying higher similarity with during Action updates than other regions. (Middle) amPFC showing more similarity than the other two during Agent updates, and (bottom) vmPFC showing similar specificity with the Precuneus during State updates. Error bars denote the standard error of the mean.

compared the ISPC values in this region with the ISPC values in PFC regions. If the Precuneus performs a global integration role, then we would expect its ISPC dynamics to resemble those seen in different PFC regions in a domain-specific manner (Fig. 5c). Around State updates, we would expect the Precuneus ISPC time-course to be most aligned to that of vmPFC, since these regions should both be engaged in processing State-related changes at these times. For Agent updates, it should be most similar to amPFC and for Action updates, to dmPFC. Thus, we tested the correlation between the two regions' ISPC values during belief updates of each type.

To examine this, we computed ISPC time courses for all regions and computed correlations between them for different segments of the movie. During State updates, vmPFC ISPC (Fig. 5d bottom) was more correlated with the Precuneus ISPC than amPFC and dmPFC ($p < 0.001$). For Agent updates, amPFC ISPC (Fig. 5d middle) displayed stronger correlations with the Precuneus ISPC than vmPFC and dmPFC ($p < 0.001$). Action updates elicited higher dmPFC ISPC (Fig. 5d top) correlations with the Precuneus ISPC than vmPFC and amPFC ($p < 0.001$). However, it is important to note that the PFC-Precuneus correlations were negative during Action updates (though least negative in dmPFC). We consider reasons for this in the Discussion.

Finally, we extended this ISPC analysis over a range of temporal windows surrounding the updates (Supplementary Fig. 14) to assess whether these trends persist over other window lengths. The same overall pattern of results was observed for windows of length 4 to 10 TRs, before dissipating when the window length exceeded 10 TRs. It should be emphasised that for the intersubject analyses, at very short windows, the small number of observations could lead noise to drive the results, while at very long windows, the window will include neural data that occur sometime after the update and are unlikely to be

related to it. Updates themselves are somewhat transient events, but our data suggest that their effects on neural patterns persist for some time after they occur.

These results indicate that the representational dynamics of the Precuneus resembles that of different PFC regions at different points during the movie, with the resemblance determined by which domain is currently engaged in model updating. To determine the specificity of this result, we performed similar analyses comparing PFC regions with the Visual cortex, Hippocampus and with other parts of the DMN: Angular Gyrus, Middle Temporal Gyrus, Retrosplenial Cortex and Posterior Cingulate (Supplementary Figs. 15–20). None of these regions showed the same domain-specific changes in ISPC correlations with PFC regions, suggesting that the global unification of multi-domain prefrontal predictions here is specific to the Precuneus. These results cannot confirm whether there is a unidirectional or bidirectional flow of information between these two zones. However, they do highlight a network-centric representation of the unified internal model. Next, we investigate if the resulting integrated representations here might be interfaced with the incoming sensory data online during the updates.

### Update-driven coupling of precuneus with visual regions
The above results suggest that the Precuneus might selectively unify the updated prefrontal representations. We hypothesise that it integrates these predictions to form a unified world model. This is then used to influence and constrain processing in visual associative regions, also shaping the ongoing subjective experience of the movie. No other node of the DMN, nor key sensory regions, showed the same pattern of domain-selective similarity with PFC. Thus, the Precuneus appears to be in a unique position to integrate top-down predictions

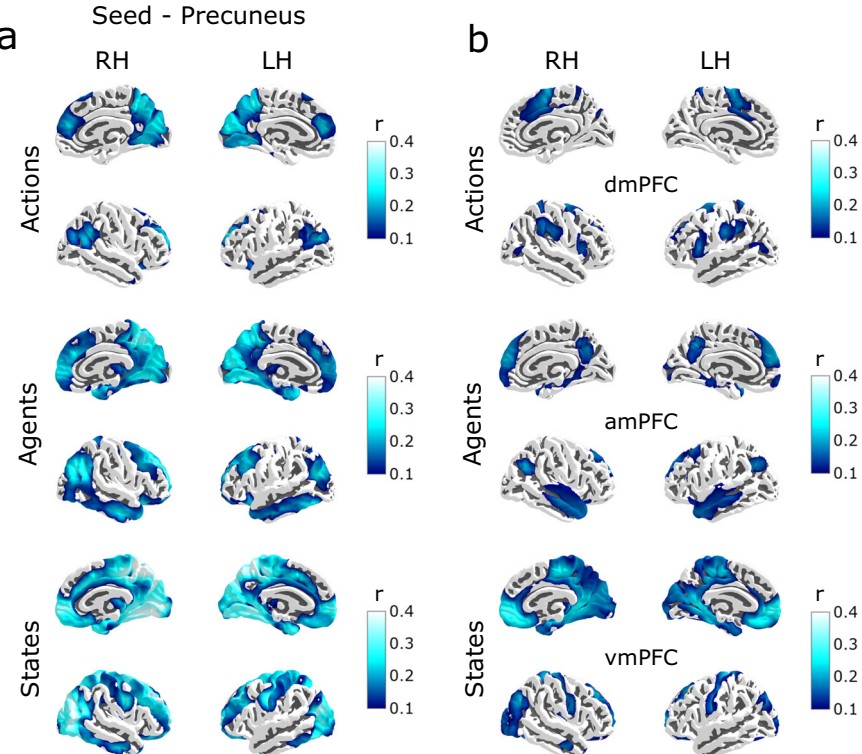

**Fig. 6 | Visual sensory regions synchronise more with the Precuneus than PFC during belief updates.** Intersubject functional connectivity (ISFC) between whole-brain voxels and (**a**) precuneus seed, and (**b**) domain-specific PFC seed (for its domain). Values suggest an increased visual cortex functional connectivity with the Precuneus than the PFC. Maps are visualised at ($r > 0.1$, FDR $p < 0.001$). Updates in (Top) Actions (Middle) Agents (Bottom) States.

with the sensory stimuli processed throughout the cortex during these updates. To test this idea, we took the time periods when updates occurred using the same approach as the preceding analysis and performed a seed-based ISFC[44,51] analysis for each domain.

Connectivity maps (Fig. 6a) show activity in the Precuneus was correlated with visual sensory regions (Visual Cortex) during updates in each domain (we later show that this connectivity is absent when participants listened to an auditory story). In contrast, domain-specific Prefrontal nodes showed less connectivity with visual regions (Fig. 6b). Whether seeding in the Precuneus or PFC, coupling with a unique set of networks specific to a domain occurred; Hippocampal/Para-hippocampal regions during States, Temporo-Parietal junction and Anterior Temporal lobe for Agents, in addition to other heteromodal and associative regions. When combined with the previous analysis, these results indicate the Precuneus is highly coupled with both the prefrontal top-down predicting regions and the bottom-up visual information. This suggests a role for integrating both of these during belief updates, shaping the ongoing subjective experience.

The results so far are broadly consistent with our conjecture. The midline PFC updates the world models in a modular way, fragmented into three domains, actively generating and adapting predictions of it. These separate classes of predictions are then unified and integrated with visual sensory regions, through the Precuneus. Such a network-level process hints to the Precuneus as a hub having access to the integrated form of prefrontal predictions. Thus, this region could potentially be an important neural correlate of unified subjective experience.

### Integrated representations in the Precuneus track ongoing subjective experience

Integration of top-down predictions with bottom-up sensory information is key for a unified current model of the world. Since the Precuneus connectivity seems to suggest this integration occurs here, we predicted that this region would have a unified representation of the movie experience.

Previous studies have observed that this region had similar representations, which were relatively higher than other cortical regions, during movie watching and subsequent recall, for both within and across participants[53]. This suggests the experiential changes due to the movie might be reflected in its neural dynamics. We used emotional Arousal ratings as our measure of overall movie experience. People experience high arousal around moments when they have high uncertainty that change their understanding of a situation[48]. Therefore, we used levels of emotional arousal as a proxy for the degree to which participants are engaged with the unified experience of the movie.

First, we repeated our bootstrapped HMM analysis, investigating whether neural state transitions coincide with times when participants experience high Arousal. We predicted that the Precuneus would show neural shifts linked with high arousal moments, while transitions in the prefrontal regions would be more specific to their respective domains (as shown previously). Other than setting the threshold θ to 1 SD for Arousal (due to no points surviving at 2 SD threshold we previously used; see "Methods"), the exact same procedure was applied here.

We observed that periods of high arousal captured a strikingly large proportion of neural state transitions in the Precuneus, more than periods of domain-specific belief updates (Fig. 7a, top). This difference was significant throughout all the temporal windows used to model updates/high arousal. Crucially, none of the prefrontal regions showed transitions that coincided with Arousal in the same way. Instead, each PFC region's transitions coincided with updates in its specific domain (Fig. 7a). The Precuneus effect persisted also when we used an alternative definition of high arousal (times where arousal was greater than mean arousal, rather than more than 1 standard deviation higher than the mean) (Supplementary Fig. 13).

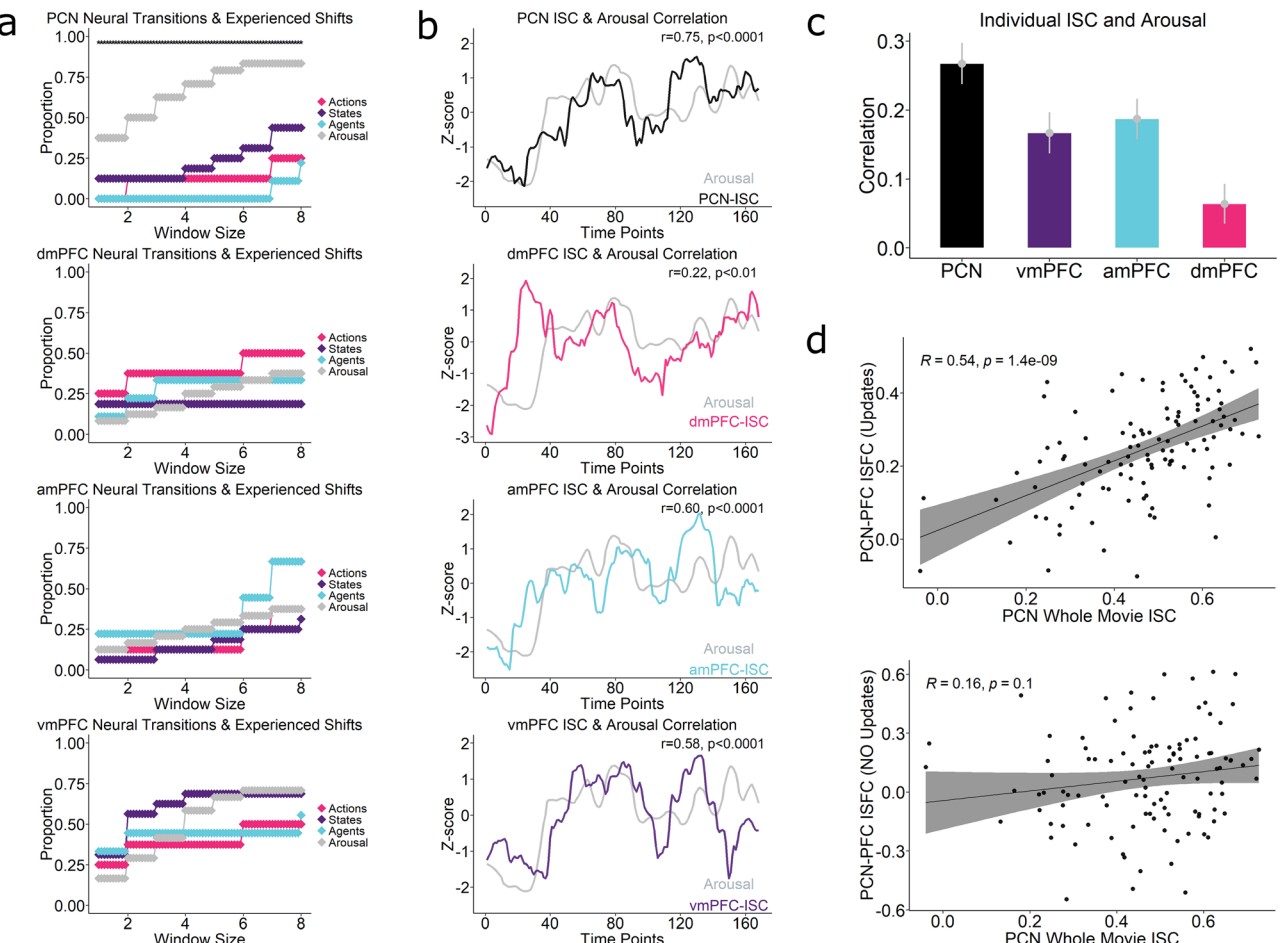

**Fig. 7 | Precuneus Unifies Fragmented Predictions into Global Experience.**
**a** Neural transitions analysis applied on the Precuneus and PFC regions included transitions linked with periods of heightened Arousal, a measure of global experience. Arousal shifts was captured much more than each of the belief updates. PFC regions do not capture Arousal more than their specialised domains.
**b** Correlation between Arousal and Group-averaged ISC time-courses showing Precuneus having larger correlations ($r = 0.75$) than dmPFC ($r = 0.22$), vmPFC ($r = 0.58$) and amPFC ($r = 0.60$) ($r$-values shown uncorrected for multiple comparisons). **c** Correlation between Arousal and participant-level ISC time-courses

($n = 111$). Precuneus had more correlation than each of the PFC subregions. Error bars denote the standard error of the mean. **d** Relation between PFC-PCN coupling during updates and shared neural responses in the Precuneus. Scatterplot shows average functional connectivity between PFC and the Precuneus (integration of prefrontal predictions across all domains), during updates along $y$-axis, correlated (Pearson) with the whole-movie Precuneus ISC (movie shared experience) along $x$-axis during (**top**) update time points and (**bottom**) nonupdate time points f. Dots represent individual participants. Shaded bands denote the 95% confidence interval of the fitted linear model.

Next, we obtained a dynamic intersubject correlation time-course[44] (sISC) for the Precuneus and compared the Arousal time series to this. The rationale was to compare moment-to-moment fluctuations in the shared response of this region with the evolving movie experience. This analysis uses a sliding window to compute ISC at each point in time, thus providing temporal information about shared, movie-driven activity. If a region is tracking the unified experience, then it should correlate more with Arousal, compared to regions carrying only the fragmented experience. The Precuneus showed more correlation with Arousal than each of the Prefrontal subregions on participant-level (Fig. 7c) (PCN vs vmPFC $t = 2.8342$, $p$-value $= 0.005467$, $d = 0.27$, PCN vs amPFC $t = 2.4467$, $p$-value $= 0.016$, $d = 0.23$, PCN vs dmPFC $t = 5.1409$, $p$-value $= 1.198e-06$ $d = 0.49$). Group-level time courses (Fig. 7b, top) showed strong correlations between the Precuneus ($r = 0.75$, $p = 1.62e-31$) and Arousal, more than each of the prefrontal regions individually (Fig. 7b), suggesting its representations are unified and covaries strongly with subjective experience. (vmPFC $r = 0.58$, $p = 1.29e-16$, amPFC $r = 0.60$, $p = 1.12e-17$, dmPFC $r = 0.22$, $p = 3.95e-03$).

**Update-driven coupling with PFC relates to shared neural responses in the Precuneus**
We further explored the relationship between the integration of prefrontal predictions in the Precuneus and individuals' shared experiences of the movie. This was prompted by findings that suggest a close relationship between the Precuneus ISC and experienced arousal dynamics. As a neural measure of the overall degree to which the Precuneus was engaged in integration with PFC, we averaged its ISFC profiles with all 3 PFC regions for each participant. We hypothesised that individuals with similar integration of prefrontal predictions in the Precuneus would share more similar experiences than those with different integration profiles. To test this, we correlated the average ISFC between the Precuneus and PFC during updates with whole-movie Precuneus ISC (Fig. 7d top), which showed a significant correlation ($r = 0.54$, $p = 1.4e-09$). Importantly, this was present only during update time points, and not during the nonupdated time points (Fig. 7d bottom) (see Methods on how these time points were extracted).

We also correlated the whole-movie Precuneus ISC with each of the PFC regions (during its domain-update) separately (Supplementary Fig. 21), which showed significant correlations in each region (though

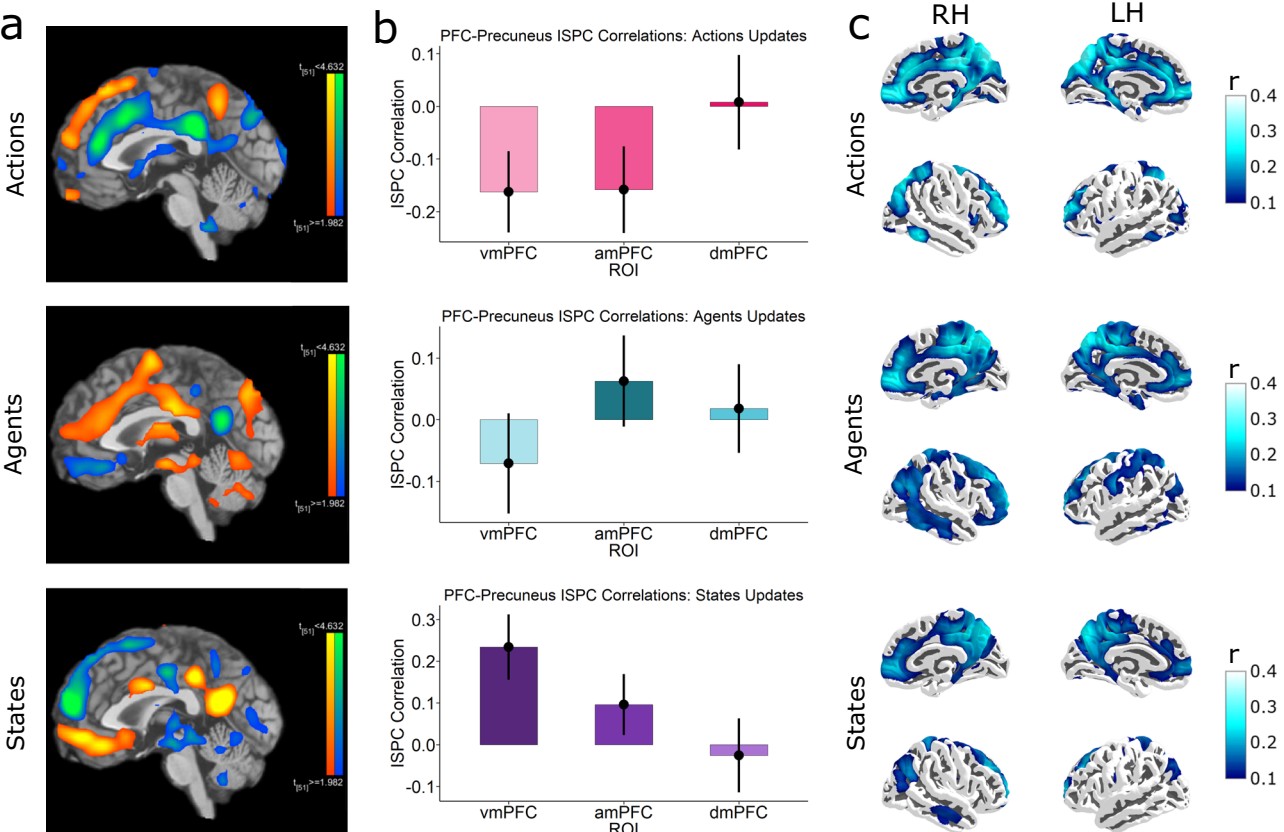

**Fig. 8 | Fragmentation and Multithreading are independent of sensory modality or content. a** Whole brain maps ($p < 0.05$ FDR corrected) show a topographically distinct activation in the midline PFC to revisions of predictions in States (bottom), Agents (middle) and Actions domains (top) during spoken narrative processing. **b** Domain-specific integration between the Precuneus with vmPFC during States and amPFC during Agents, and dmPFC during Actions ($n = 52$).

Error bars denote the standard error of the mean. **c** Intersubject functional connectivity (ISFC) between whole-brain voxels and the Precuneus seed. Unlike the movie, here there is no evidence of visual cortex functional connectivity with the Precuneus (c.f Fig. 6a) during processing of an audio narrative. Maps are visualised at ($r > 0.1$, FDR $p < 0.001$).

lesser for dmPFC/amPFC than vmPFC). These results were also only present during the update as compared to the nonupdated time points, emphasising their role in integrating updates to their current world model. While various unknown factors may influence shaping experience, placing this alongside our broader results suggests that similar levels of prediction integration are associated with comparable shared experience.

The picture emerging suggests that the Precuneus unifies prefrontal predictions with bottom-up sensory data into a coherent, continuously evolving experience during unguided natural settings.

**Generalisation to spoken narratives, input modality and emotional content**

Are these results limited to this movie's content? More importantly, is this result limited to the visual modality? To test this, we replicated our two central results - modular prefrontal predictions and integration with the Precuneus - on a different cohort processing a spoken narrative. The narrative was similar in duration to the movie (8 min vs 9 min) but involved dramatically different content and emotional valence (humorous, c.f. the suspenseful content of the movie). Most importantly, it was presented as spoken audio. This allows us to assess the generalisability of our claims across people, content, sensory modality and emotional salience (and a different pre-processing pipeline, see "Methods").

We first tested for activity covarying with State, Agent and Action belief updates. Whole brain maps show remarkably similar prefrontal fragmentation while listening to a spoken narrative, suggesting that

these prefrontal modules fragment different types of experience in a highly consistent manner (Fig. 8a).

We then compared whether shared patterns during updates in the PFC (via ISPC time courses) showed domain-specific alignment with the Precuneus ISPC (Fig. 8b). We found evidence of multithreaded integration in States ($p < 0.001$) and Agents ($p < 0.001$), and in Actions ($p < 0.001$). We also extended this ISPC analysis over a range of windows (Supplementary Fig. 23) to confirm that these trends persist over other window lengths. These results suggest integrated predictions in the core DMN are modality-agnostic, i.e., abstract.

The movie analyses showed significant functional connectivity between the Precuneus and visual cortex (Fig. 6a). In contrast, during an audio narrative, no relevant information comes through visual channels. Accordingly, a functional connectivity analysis (ISFC) seeded on the Precuneus showed very low coupling with the visual regions in the narrative data (Fig. 8c). We did not observe significant coupling with auditory regions of the superior temporal lobe. This may be because when listening to speech, the auditory signal itself carries little information that can directly influence the world model. The speech signal must first be transformed into lexical-semantic representations. This process involves parts of the ventral and lateral temporal cortex that did show coupling with the Precuneus.

Overall, this replication addresses potential limitations of using a movie stimulus and provides for an independent validation of some of the main results. In doing so, this cements our claim that humans utilise a set of modular predictive models in vision and language inference during general world modelling.

## Discussion

Inductive biases offer useful and computationally advantageous prior knowledge in structuring internal models[10]. By analysing prediction updates across different domains during naturalistic experience, we uncovered how humans might utilise such biases to represent different internal models. We suggest that humans model the world by partitioning it into three distinct domains within the PFC. Each model occupies a topographically distinct portion of the midline PFC. Our analyses of fMRI movie-watching data suggest that these three parallel neuronal systems adaptively guide predictions for each domain; namely, States, Agents and Actions. We found evidence that these top-down predictions are then unified in the Precuneus, the posterior hub of DMN. We propose that the Precuneus continually integrates top-down predictions with bottom-up sensory information to form and update the current model of the world. These results also generalised from a movie to a spoken narrative with very different content. This illustrates how the DMN contains modular representations of abstract predictive models.

Our results support our proposal that the joint world modelling process is divided into dedicated modules of States, Agents and Actions models. Domain activation profiles mapped to distinct regions in the midline PFC. These roles align with insights from various lines of work[20–33]. However, to our knowledge, they have not been previously integrated into a unified theoretical framework, localising them to the prefrontal cortex. First, the vmPFC, traditionally associated with reward learning and decision-making, appears to play a broader role in context-based inference. In our study, the vmPFC responded to context changes within the movie. In more goal-directed situations, relevant states might relate to task instructions or the reward value of different stimuli. This supports the theory that State estimation is a core function of the vmPFC[20,22,25]. We generalise this notion into a model space of States encoded within this region. Here, vmPFC not only tracks but also generates predictions about various States in the environment, updating these predictions as necessary to navigate experiences.

The amPFC plays a crucial role in various forms of complex social cognition[26–28]. Central to these functions is the ability to construct reference frame models of Agents, enabling generalisations to new or familiar individuals across varying contexts. As a social species, our understanding of the world would be dangerously incomplete without having robust models of the people around us. This allows us to anticipate their emotions and behaviour accurately. In the present study, amPFC activity was coupled with updates in beliefs about the characters in the movie. However, amPFC is also highly engaged in reasoning about our own (future) mental states[54], suggesting that agent models also guide interpretation of our own motivations and behaviours[18,29–31]. It's important to reiterate that agent and state- based predictions are often orthogonal. Individuals exhibit personality traits that are stable across various contexts, and environments possess characteristics that remain consistent regardless of the inhabitants. This orthogonality makes it computationally sensible to code State and Agent predictions separately.

Managing a vast state space requires abstracting ways of transitioning, or paths across it. This allows us to navigate through various states to achieve different goals. Modelling temporal properties that evolve over extended periods is crucial for this. The dmPFC, our Action model space, plays a critical role in strategic decision-making[33], hierarchical planning[34], and compressing action sequences over time[32]. These functions are vital for encoding and inference through abstract Action models, where specific actions trigger particular paths or sequences. These models are built and represented separately to underlying reference frames (coded by agent models) or the contexts (coded by state models) in which they occur. In our study, a change in the State or an Agent's behaviour triggers an update in the possible trajectories within the inferred story. This requires adjustments to the

predicted 'paths' across States or future agent behaviours. The ability to generalise actions provides significant advantages in adapting to new goals and compositionally reusing model components elsewhere, a key aspect of human flexibility.

Our data make a case for top-down predictions also arising in a parallel, distributed manner akin to bottom-up visual pathways. Taken together, our data indicate that PFC is a core region from which top-down model predictions can arise. Importantly, it also suggests that specialisation across this region is a simple yet flexible adaptation used by the brain. This strategy processes the continuous and incredibly high-dimensional world by 'carving' it into distinct domains and separately computing predictions in each[55].

If our world models are represented across three modular systems, then why does our subjective experience of the world not feel similarly fragmented? Our results are consistent with the idea that the Precuneus unifies the prefrontal predictions, integrating them with visual sensory data. This is not only a core node of DMN (of which Precuneus is perhaps the central node), but structural[39] and functional connectivity[38] data show that this region interfaces between sensory regions and the PFC. It also acts as a connecting hub between various cortical networks[38,39]. Therefore, it is well-placed to play such an integrative role. The shared representations in this region were selectively aligned to that of each domain-specific PFC region during their corresponding updates. Crucially, these were observed only during the update events and not in non-updated segments, suggesting it's driven by the experience rather than being an intrinsic property of being the core nodes of DMN. In addition, such a form of prioritised, multi-threaded integration of prediction threads was observed to be unique to the Precuneus, compared to a host of other regions. Studies have shown the brain could implement multithreading structurally[56] and that dopamine might be functionally integrating multiple threads of reward prediction errors[17]. Maintaining complex unified representations likely requires such parallel neural architectures[57], and multi-threadedness can be seen as an adaptation to distributed errors. This leads to robust interareal communication, further bolstering this region's increasing evidence in global integration[18,41,51,58].

We also ruled out other forms of regional integration, such as within-prefrontal and with the Hippocampus. Consistent evidence emerged for the Precuneus, whose activity was attuned to changes in the unified experience. Conversely, the PFC was only selective to domain-specific shifts of experience. We found that activation dynamics in the Precuneus aligned with ratings of emotional arousal, which index temporally evolving, emotion-laden engagement with stimuli[48,59]. The usually high correlation values observed in this region across subjects in studies indicate shared representations during shared experiences[53], a proxy for the stimulus-driven states of experience. Such a functional manner of integrating predictions into experience might also underpin neural correlates of consciousness, since a unified conscious experience requires us to integrate the various facets of our current experience of the world.

A dominant question in consciousness research is to adjudicate between various neural theories of conscious processing. Currently, a major theoretical debate is whether it is the prefrontal or posterior parietal zones that mediate access to conscious representations[60]. Our results suggest that the answer might be a holistic gathering of both the PFC and the Precuneus. Perhaps, PFC is required but ultimately generates an incomplete, coarse-grain experience, while parietal integration is critical for the final view[37]. This was in line with our results, where individuals with similar prefrontal predictions integrated with the Precuneus had similar shared global experience. As a consequence, it becomes difficult to falsify competing theories that have neural implementation shared with each of these regions. Indeed, such an ambiguous conclusion was observed in a recent adversarial experiment[61], which pit these two theories against each other.

Alternatively, multiple, concurrent streams of consciousness are central to some philosophical theories of consciousness[62]. Here, different neural modules can have 'control' at different times. Implementing any theory into neuronal machinery to be called as a neural correlate of consciousness (NCC), requires satisfying several different criteria[63]. One such criteria is the differentiation between global and local contents. Fragmented prefrontal representations and their eventual integration within the Precuneus might be seen as a way of differentiating these. Another constraint is that the NCC should be a systematically specific form of conscious processing, rather than an arbitrary or spurious neural association. In our framework, the domain-specificity of these modules (e.g., conscious updates to contexts vs people) satisfies such a requirement. Predictive processing frameworks centred on different cortical networks seem to be a promising avenue to explore here. Despite domain-specificity, these regions still responded in a remarkably similar manner during updates to perceptually and emotionally different input. This suggests that these representations are separated from the concrete textures of the senses, something the DMN is in a legitimate position to fulfil.

A rich literature of cognitive[12,13,18,48,50,51,53,64] and clinical studies[40,41,65] supports the role of DMN in higher-order human cognition. Although classically seen as task-negative, this network is implicated in a variety of cognitive activities associated with subjectivity, such as mind-wandering, creative thought, self-related processing and mental time travel. These tasks are inferential in nature and possibly involve the construction of rich internal models of experience. Situating the DMN within the literature, one finds numerous links to episodic memory, which extends beyond the hippocampal spatial machinery from which it evolved, to encode more abstract temporal and referential properties[66]. Any experience can potentially be represented as a structured set of states, forming a cognitive map[67] with distinct transitions between them. Re-experiencing an event involves traversing this state space, akin to path integration in the physical domains, reinstating both states and their transitions. These transitions might follow meaningful directional trajectories, shaped by multiple abstract factors such as time and agent-specific information. The dmPFC and amPFC may underlie such roles during episodic encoding, sensitive to changes in predicted trajectories. Indeed, episodic memories of experience are thought to be compartmentalised through event segmentations, which are functionally driven by prediction errors[45,68,69].

Situation models often require encoding object-space relational properties as schemas—generalised templates that facilitate rapid learning of similar contexts and prediction of novel contexts, a function linked to the vmPFC[70]. Reference frames are critical in social/self-processing[71] and conceptual integration across viewpoints[72]. Some knowledge is anchored to an external, world-centred (allocentric) reference frame, while others are encoded relative to the self (egocentric) or others. The ability to condense conceptual knowledge across various frames into task-centric reference frames may be a key function of the amPFC. Distinct cognitive maps emerge from different experiences, prompting inference across them in extended contexts. The dmPFC might then aid in traversing these different cognitive maps as observed in exploration[35], and in this context, mind-wandering[73]. Finally, during naturalistic experiences, the DMN often co-activates with not just the medial temporal lobe structures, but also the anterior temporal lobe, which represents various forms of semantic and conceptual content[74]. It's an open question whether the distinct predictions that emerge in the PFC depend on its interaction with specialised representations of conceptual knowledge in this region.

Throughout this paper, we have referred to "fragmentation" of the model space. By fragmentation, we mean the within-network deconstruction and reconstruction of an internal model through separate functional updates. We have argued this is located in distinct PFC subregions. Thus, multiple prediction 'threads' have to be active and integrated with the relevant sensory channels for updating and exploiting high-level models, perhaps through top-down and bottom-up integration, potentially experienced egocentrically in the medial parietal regions. This transient, modular breakdown of the internal model, followed by its reintegration, can explain the resurfacing of this set of regions across superficially distinct but computationally similar tasks in different studies[75]. Fragmentation of experience may be seen as a possible reason behind consequential clinical accounts like blindsight, spatial neglect[76], dissociative consciousness disorders, and in extreme cases commissurotomy-related phenomenona[77]. Inability to integrate these prefrontal predictions can offer a fresh perspective in examining psychiatric conditions with independent (and often rebellious) 'conscious' entities within. Indeed, 'misintegration' by the Precuneus, the hub of DMN, are well reported in clinical studies underlying related phenomena[40,41]. Finally, there is a computational formalism of seeing resting-state DMN activity as the prior models encoded in the cortex, under Bayesian frameworks[4,78]. The present study is suggestive of the dynamic processes and structural constraints by which these priors are updated as an experience unfolds.

From a methodological perspective, one strength of our study is that our neuroimaging participants were not given any specific cognitive task to perform while experiencing the story. Having them explicitly provide conscious ratings of their updates would have changed their experience, evoking metacognitive/response-related neural signatures and precluding a fully natural experience of the movie. Our design was specifically aimed to detect naturally occurring predictive changes rather than perceptual changes, viewed through the lens of an individual's internal model. This approach minimises instructional effects and offered a window into the nature of our internal models. However, it comes with the limitation that we do not have data on precisely when individual participants in the scanner experienced updates. This means that we were not able to investigate how the timing and nature of updates vary between individuals. Like most neuroimaging studies, most of our analyses are correlational, exploiting the model generation and updating processes that occur spontaneously during a naturalistic stimulus. In future works, it will be important to exert more experimental control over the nature and timing of such processes in order to validate our findings. In particular, updates to states, agents, and actions were not fully independent of one another, and some plot developments seemed to result in multiple domains updating simultaneously. This presumably reflects the fact that unfolding events often provide information relevant to more than one domain. This dependence could potentially be avoided in future studies by using artificially-generated stimuli, though this would come at the cost of reduced ecological validity. That said, our replication of the main results across two settings supports the value of studying internal models using naturalistic neuroimaging, where a suite of specific analytic techniques has been established in recent years[12,44,48,50,52,53].

We found that ISPC in each PFC region was most correlated with the Precuneus ISPC during updates in its preferred domain, suggesting that the Precuneus plays a general integratory role. However, while these correlations were positive for vmPFC and amPFC, the Precuneus ISPC was *negatively* correlated with dmPFC during action updates. This negative effect suggests that the neural correlates of action integration may be somewhat different to the other domains. Multiple studies suggest functional specialisation within the Precuneus[79,80], with dorsal and ventral parts showing distinct connectivity profiles. In line with this, we found that dorsal Precuneus was particularly engaged by action updates, while ventral Precuneus was more activated by agent or state updates (see Figs. 3a and 8a). Thus, there may be second-order specialisation within the Precuneus for integration of different elements of the world model, which may explain why action updates showed weaker correlations with the Precuneus as a whole. Future works using functional-gradient methods and fine-grained parcellation

could more precisely map these subregional networks, providing deeper insights into the role of DMN in world modelling.

To summarise, we claim that humans model the world by fragmenting it into different domains first – states, agents and actions. Each of these internal models, potentially leveraging different kinds of inductive biases or computations, are represented along a functionally distinct topography in the PFC. Prediction threads from PFC are integrated within the Precuneus, enabling the DMN to shape and transform these into a unified subjective experience. Such parallel, modular representations highlight the inevitability of distributed processing in the brain. Through the broad framework of predictive coding and ecologically rich designs, we hope to have offered a novel and unificatory account of various phenomena associated with the DMN, capturing its possible role in general world modelling.

## Methods

### Stimuli
We examined the behavioural and neuroimaging responses of participants to two naturalistic stimuli –a movie and a spoken narrative - sourced from two different datasets. The movie was an edited 8 min excerpt from Alfred Hitchcock's "Bang! You're Dead!", obtained from the Cam-CAN dataset[42]. In short, this movie involves a boy discovering a loaded gun and pulling the trigger at various unsuspecting people. It involves various shifts in context and various beliefs about the characters and their possible actions at each juncture of the plot. For the narrative, we used an audio clip (9 min 7 sec) of "It's Not the Fall that Gets You" derived from the publicly available "Narratives" dataset[43]. It concerns a self-narrated account of a person trying to date at a skydiving academy and the various bloopers that occur. Both stimuli were chosen because they are linked with publicly available fMRI data from their large samples and are well-studied in the existing literature. Most of our analyses used the movie stimulus, while the data from the narrative served as a replication dataset with a different modality and content, establishing the generalisation of key results.

### Participants
Our study comprised of 8 experimental groups – one fMRI and three behavioural for each stimulus modality, with no overlap between them (see Fig. 2).

129 participants provided ratings of various aspects of the stimuli. 58 UK-based participants (21–35 years, Mean: 27.6, SD = 3.3, 47% females) were recruited via the online platform Prolific to provide continuous belief update ratings, distributed into 3 separate groups – States ($n = 18$), Agents ($n = 21$) and Actions ($n = 19$). In addition, continuous ratings of Arousal ($n = 17$) were obtained from a previous study[48].

Participants were demographically matched with the fMRI participants (UK residents aged 21–35). All participants were native English speakers and had normal or corrected-to- normal vision and hearing. All participants reported no previous history of watching the movie.

Similarly, prediction ratings on the narrative were collected from 3 independent groups (21–35 years, Mean: 28.5, SD = 3.5, 44% females) of participants using Prolific, for States ($n = 16$), Agents ($n = 18$) and Actions ($n = 20$). The participants reported no previous history of listening to the story.

The online rating studies were approved by the University of Edinburgh School of Philosophy, Psychology & Language Sciences Research Ethics Committee, and all participants gave written informed consent.

### Neuroimaging data sources
fMRI data for the movie were obtained from a subset of the healthy population-derived Cam-CAN cohort ($N = 135$, Age:18–35 years)[42]. The full Cam-CAN cohort includes individuals aged between 18 and 87, but here we focused on the age range typically investigated in fMRI studies of healthy young adults. This age range was also comparable with the narrative dataset described below.

Participants exceeding a maximum framewise displacement of 1 mm or angular rotation >1.5° were excluded from subsequent analyses (see fMRI preprocessing), leading to a sample of 111 participants ($\mu$age = 28.5 ± 4.93, 63 female) for further analyses, as described in a previous study[48]. Functional MRI scans were acquired on a 3 T Siemens TIM Trio system, using a T2*-weighted multi-echo pulse sequence with a TR of 2470 ms, and multiple TEs of 9.4 ms, 21.2 ms, 33 ms, 45 ms, 57 ms, a flip angle of 78°, voxel size of 3 x 3 x 4.4 mm with a thickness of 3.7 mm, FoV = 192 × 192 mm, and 32 axial slices. The movie-watching session lasted for 8 minutes and 13 s (493 s), yielding 193 TRs, of which we discarded the first 4 volumes from all analyses.

The narrative dataset[43] consisted of preprocessed (fMRIPrep) 3-T (Siemens Magnetom Skyra) fMRI T2* weighted BOLD responses (TR 1500 ms) from 52 participants (18–29 years, $\mu$age = 28.5 ± 4.93, 31 female). Functional BOLD images were acquired in an interleaved fashion using gradient-echo echo-planar imaging (EPI) with an in-plane acceleration factor of 2 using GRAPPA: TR/TE = 1500/28 ms, flip angle = 64°, bandwidth = 1445 Hz/Px, in-plane resolution = 3 × 3 mm, slice thickness = 4 mm, matrix size = 64 × 64, FoV = 192 × 192 mm, 27 axial slices with roughly full brain coverage and no gap, anterior–posterior phase encoding, prescan normalisation and fat suppression. The functional scanning session included 400 TRs, totalling 600 s of acquisition time.

### Belief update time-courses
The stimuli were presented to the participants using Testable (testable.org). Before the start of the experiment, participants were given instructions that included a definition of the type of beliefs they were being asked to monitor (State/Agent/Action). An example of a situation in which they would need to update those beliefs (from a scenario unrelated to the stimuli) was also included. Participants were asked to focus on the stimuli (movie/narrative) and press a key immediately when they felt their beliefs in the domain of interest had been updated. Full instructions given to the participants for all three domains can be found in Supplementary Table 1.

For each participant, this yielded a binarised time-course indicating the points at which they signalled that they had experienced a belief update. The precise timing of updates varied between participants. Even when two participants experienced an update in response to the same event in the story, they might respond at slightly different times due to differences in their speed of responding, attentiveness and threshold for deciding an update has occurred. To accommodate this variability, we created a 5 s update window[81] around each belief update point, generating a boxcar timeseries for each participant with a value of 1 for the temporal regions at which they experienced updates and 0 for all other times. We then averaged these time series over participants. The group-mean time-course then represents, for each point during the movie, the likelihood that a person watching the movie would be experiencing an update. Group-averaged time-courses for each domain were then smoothed using local regression smoothing (loess in R, a non-parametric approach) with a 10% span, which was kept the same for both movie and story. These smoothed ratings were downsampled to match the fMRI BOLD time-course, providing an update probability for each image acquired during movie-watching (see Fig. 1c).

We computed the correlation between all three update time courses to check for any strong ($r > 0.5$) coupling between any two domains, which might render the neural analysis less feasible. Additionally, an inter-rater consistency (IRC) analysis was performed to assess the reliability of ratings. For each domain, the participant pool was randomly split into two subgroups. Each subgroup's time courses were smoothed, averaged, and their group-mean correlated with the other subgroup's mean, which was corrected with the Spearman-

Brown formula to assess reliability. This process was repeated 100 times for each domain in both movie and narrative.

The smoothed update probability time courses were used to predict BOLD activation in univariate analyses. Other analyses (HMM, ISC, ISFC, ISPC; see below) required discrete estimates of periods when updates occurred. To generate these, for each domain, we applied a threshold θ to the smoothed group-averaged timeseries. Time periods where the update probability exceeded θ were set to 1, and the rest to 0. Where a sequence of multiple TRs/scans exceeded θ, we selected the final TR/scan in the sequence to represent the update.

We adopted a θ of 2 SD above the group-mean update probability for all movie update time-courses and a θ of 1 SD above the group-mean value for the Arousal ratings (since no timepoints were 2 SD higher than the group-mean). Update timecourses for the narrative used a θ of 1.2 SD as there was less variation in the update timecourses for this stimulus. The selection of θ was driven by a desire to balance detection of sufficient update periods with maintaining temporal specificity in the updates in each domain. Supplementary Figs. 4 and 5 show how these factors vary with θ. If θ was set too high, very few timepoints were classified as updates. If it was set too low, many updates were identified, but these frequently overlapped one another in time, reducing the specificity. Our selected θ values balanced these two constraints. Nevertheless, there were some occasions when updates in two domains overlapped in time (approximately 25% of identified updates). We repeated our main analyses after excluding updates that overlapped over domains, which gave similar results to those we report in the paper (based on all updates).

To rule out whether the observed effects (e.g., increased coupling between the regions) are due to belief updates or an inherent property of the network, we computed these neural results on timepoints where no updates occurred (nonupdates). For each domain, we identified non-update time points by ranking all TRs from lowest to highest rating and then selecting those that (1) lay at least one TR (~5 s) away from any update in any domain (using the same 2 SD threshold as for updates) and (2) matched the total count of update events for that domain. This ensured our non-update samples were low in update signal, temporally isolated from genuine updates and are balanced in terms of update points, allowing a fair comparison across domains.

### Arousal Ratings

The continuous ratings of Arousal were collected as part of a previous study[48]. Participants were instructed to watch the movie while continuously rating it with respect to their emotional intensity (Arousal) using their mouse on a vertical slider ranging from −1 to 1. We used the Arousal timecourse as a measure of the degree to which participants were immersed in the experience of watching the movie at any given time.

### fMRI preprocessing

The fMRI data for the movie were processed using SPM12 using a standard preprocessing pipeline consisting of slice time correction, realignment, co-registration, normalisation and smoothing (6 mm x 6 mm x 9 mm FWHM), as described in a previous study[48]. To account for motion artifacts, six parameters capturing translation and rotation (X-, Y-, Z-displacements, and pitch, roll, yaw) were removed from the functional data through least-squares regression. Participants exceeding a maximum framewise displacement of 1 mm or an angular rotation >1.5° were excluded from subsequent analyses. The processed functional data then underwent voxel-wise detrending and was subjected to a band-pass filter between 0.01 and 0.1 Hz, implemented using a second-order Butterworth filter.

The narrative data we obtained were already preprocessed using fMRIPrep[43]. These images were smoothed with an 8 mm x 8 mm x 8 mm FWHM kernel for the GLM.

For all the voxel pattern analyses, unsmoothed voxel-wise BOLD time series were used.

### Region of interest (ROI) definition

All ROI masks were defined from the Brainnetome Atlas[46]. We created the vmPFC ROI by combining 6 subregions in the Orbital Gyrus region. Similarly, amPFC and dmPFC ROIs were created by combining 6 and 4 (non-overlapping) subregions from the Superior Frontal Gyrus. Precuneus (PCN) ROI was constructed by combining all 8 subregions in the Precuneus region. The four primary ROI masks (PFC and PCN) pertaining to our overarching hypothesis, are visualised in Supplementary Fig. 1. Other regions used in control analyses (Hippocampus, Visual Cortex, Posterior Cingulate Cortex, Retrosplenial Cortex, Angular Gyrus and Middle Temporal Gyrus) were similarly constructed from various subregions as used in a previous study[48], and was mainly used as control to establish specificity of effects.

### General linear model (GLM)

To estimate the neural responses corresponding to the State, Agent and Action updates, we modelled the whole-brain movie-viewing BOLD timeseries by fitting a general linear model (GLM). It included a regressor modelling the temporal dynamics of the entire movie as multiple events with 1 s duration, convolved with the canonical hemodynamic response function (HRF) to model the expected BOLD response. Additionally, the GLM incorporated the three group-averaged smoothed timeseries of the prediction updates as parametric modulators (mean-centred). This allowed us to assess how fluctuations in these prediction time courses modulated brain activity throughout the movie. Six head-motion parameters defined by the realignment were added to the model as nuisance regressors.

We assessed 3 statistical contrast maps – States > Agents + Actions, Agents > Actions + States and Actions > States + Agents with weights [1 − 0.5 − 0.5] for each map. All contrast images were obtained at the participant level, and a group-level random-effects analysis was conducted. Thereafter, we thresholded the statistical maps at $q < 0.05$ FDR with an extent threshold of 25 voxels ($k = 25$), which was performed using NeuroElf (http://neuroelf.net).

The same GLM structure was employed for the Narrative data.

### Intersubject correlation analysis (ISC) during updates

For each domain (State, Agent and Action), we first constructed a 7 scan(TR) window around the update points previously identified, i.e., 3 scans before and after update scans, without any overlapping segments. Thereafter, these segments were extracted and concatenated and used to compute a leave-one-out ISC on each of the concatenated time courses in each PFC ROI. To do this, we computed the average BOLD timecourses in each ROI and calculated the correlation of these timecourses for each participant with the mean of all other participants (minus the selected participant). This method ensured any shared neural activity was a result of stimulus-driven processing than any idiosyncratic noise. The State update time courses consisted of 63 scans, Agents had 28 scans, and Action had 35 scans.

### Bayesian regression for GLM betas & ISC

The estimated GLM beta values (or ISC values) of each PFC region during the updates were tested in a Bayesian hierarchical regression model using participants' beta/ISC values as the dependent variable. This modelled the effects of Domain (States, Agents, Actions), ROI (vmPFC, amPFC, dmPFC) and their interaction on beta/ISC. It included random effects of participants including random slopes for the fixed effects to capture participant-level variance to these terms.

The model equation was therefore $Y_{ij} = \beta_0 + \beta_1(Domain)_{ij} + \beta_2(ROI)_{ij} + \beta_3(Domain \times ROI)_{ij} + \gamma_{0j} + \gamma_{1j}(Domain)ij + \gamma_{2j}(ROI)_{ij} + \varepsilon_{ij}$ Where:

$Y_{ij}$ is the ISC outcome or beta value for subject j in observation i. $\beta_0$ is the overall intercept.

$\beta_1$, $\beta_2$, and $\beta_3$ are the fixed effects coefficients for Domain, ROI, and their interaction, respectively.

$\gamma_{0j}$ is the random intercept for subject j.

$\gamma_{1j}$ and $\gamma_{2j}$ are the random slopes for the effects of Condition and ROI within subject j, respectively.

$\varepsilon_{ij}$ is the residual error for subject j in observation i.

Weak, noninformative priors were applied over the model terms.

$\beta_0 \sim \text{Student } t(3, 0.3, 2.5)$
$\beta \sim \text{Student } t(3, 0, 2.5)$
$\gamma_0 \sim \text{Student } t(3, 0.3, 2.5)$
$\gamma \sim \text{Student } t(3, 0, 2.5)$
$\epsilon \sim \text{Half Student } t(3, 0, 2.5)$

Posterior estimation was done using the "brms" package in R through Hamiltonian Monte Carlo for 2 chains of 6000 samples, with the first 500 discarded from each. Model convergence was assessed through R-hat statistics, which were found to be ~1.00, sufficiently large estimated sample size for stable posterior estimates, and by visually inspecting the chains for convergence and large autocorrelations. Posterior predictive cheques for model validation were conducted by simulating 1000 samples from the posterior and by fitting with the empirical data (Supplementary Figs. 7–10).

We were mainly interested in testing the domain-sensitiveness of each of these three ROIs for their respective domain. That is if vmPFC has higher values than dmPFC and amPFC during State updates, amPFC higher values than vmPFC and dmPFC during Agent updates and dmPFC higher values than vmPFC and amPFC during Action updates. For this, we computed Bayes Factors for (and against) these hypotheses and their associated posterior probabilities.

## Hidden markov modelling

Hidden Markov Models (HMMs) have been extensively used to compute neural state dynamics from BOLD data, and have been particularly powerful in analysing naturalistic stimuli that involve the maintenance of temporally extended internal states[12,13,48]. Our goal was to estimate the timing of BOLD hidden latent transitions in each region, and compare that to the belief update points, and contrast it across the neural regions and types of belief updates. We used the R package depmix6 for constructing the HMMs.

We aimed to implement a group-level HMM. After extracting the ROI BOLD time series for each subject, we fit our bootstrapped HMM as follows. Although HMMs are usually fitted by specifying a number of discrete neural states and then selecting based on some penalising information criteria, we face a problem of intersubject variability here. That is, different subjects would have variability in update timing for each domain, on each ROI. This meant that the transition points of group-level HMM should be robust to this factor. First, we determined the number of neural states present in each ROI. For each ROI, we randomly removed 10 subjects from the dataset (~10% of data) and estimated HMMs over a range of pre-specified latent states from 2 – 9 (for a total of 8 pre-specified states), from which the best-fitting number of states was adjudicated using Bayesian Information Criteria (BIC). This was repeated 30 times with 10 random subjects removed in each run, for a total of 300 subjects' removed or ~thrice the full dataset ($n = 111$). This resulted in a vector of the best-fitting number of states for each HMM run (i.e., 30 such values, one for each run). We took the median as the number of states for the final analysis.

Next, using this median number of states, we fit the group-level HMM and obtained the posterior trajectories using the Viterbi algorithm. Since parameter estimation in HMMs (Expectation Maximisation algorithm) is inherently stochastic (unless one seeds it), it outputs the transitions between the states with slight temporal differences. To ensure robustness against this, we ran this final HMM (using the median number of states) 10 times, and obtained an averaged

transition time course, which indicates the probability of state transitions at each point in time (0-1). Higher values suggest the transition point is consistent across the 10 runs, and hence robust for this ROI. This time course was then binarized by subjecting it to a threshold of 2 standard deviations from its mean (exactly like the belief update time courses), thus identifying scans in which neural state transitions occurred. This allowed us to not incorporate the individual variability in neural transitions in a conservative and quantifiable manner, as well as accounting for the stochasticity in transition points, which is a consequence of the Viterbi algorithm (rather than just seeding it with an arbitrary seed). Thus, our bootstrapped HMM ensured (1) the most reliable number of latent states are acquired, (2) with the most consistent transition times obtained for each ROI, (3) while preserving the update dynamics of each ROI individually, in the most statistically efficient manner. Optimal state distributions and across run similarity can be seen in Supplementary Figs. 11 and 12 respectively.

To investigate the degree to which belief updates were coincident with neural state shifts identified by the HMM, we constructed forward windows of duration $t$ scans beginning at each neural shift in the final HMM state transition time series. We then counted what proportion of belief updates of each type occurred within the scan windows following HMM transitions (see Fig. 4a). We repeated this process for windows of size 0 to 8 scans to ensure that results were robust across different temporal windows. We did not convolve the binary ratings with the HRF.

To assess significance, we used non-parametric permutation testing. We repeated each analysis 10,000 times, each time phase randomising the continuous neural transition time-series (i.e., before binarizing), while keeping the experiential shifts (belief updates) constant. We phase-randomised the transition time-courses by applying Fast Fourier transform and sampling new phases $(-\pi, \pi)$, which returns a permuted time-series with preserved autocorrelation structure as the original one[44,51]. This yielded a distribution of results under the null hypothesis that the timing of belief updates and neural shifts are unrelated. We were then able to assess where our observed result fell within the null distribution and assign a p-value accordingly. For example, an observed result more extreme than 95% of null results would receive a $p$-value of 0.05.

We used this method to test the hypothesis that HMM neural transitions were more likely to coincide with updates in a region's preferred domain than in the other two domains. The test statistic here was conditional on the criterion above (i.e., p_Rate1 > p_Rate2 & p_Rate1 > p_Rate3, where Rate1 is the domain-specific Rating to each ROI, while the other two are the 'nonspecific' ratings).

The exact same approach was later deployed for the HMM involving Arousal as well, wherein the hypothesis then becomes p_Arousal > p_Rate1, p_Arousal > p_Rate2, p_Arousal > p_Rate3 to underscore the specialisation to the integrated form (i.e., Arousal) than the fragmented form.

## Intersubject functional connectivity (ISFC) analysis for functional integration

Intersubject functional connectivity (ISFC) is a measure of the correlation of the BOLD time-series in a region in one participant to the group-averaged BOLD time-series (without this participant) to a second region. It is particularly useful in naturalistic stimuli due to its robustness to stimulus-unrelated processing (i.e., noise). We first used this method to investigate functional connectivity profiles between the three PFC regions with Precuneus, Hippocampus, as well as within them during periods of State, Agent and Action updates.

We computed the ISFC employing a similar approach to that of the ISC, by concatenating the 7-scan windows around the update timepoints (3-scan before and after) for each of State, Agent and Action. This allowed us to assess ISFC between pairs of regions in a period where each type of update was occurring. For each participant,

ISFC between a pair of ROIs (i and j) was computed by taking their concatenated BOLD timecourse for the selected updates for ROI$_i$ and correlating it with the average ROI$_j$ time-series for the other participants.

For Prefrontal Integration, we averaged the ISFC between the three PFC ROIs. This involved averaging the ISFC between vmPFC-amPFC and vmPFC-dmPFC during State updates, amPFC-vmPFC and amPFC-dmPFC during Agent updates, and dmPFC-amPFC and dmPFC-vmPFC during Action updates.

For Hippocampal and Precuneus Integration, this was computed with the hypothesised domain-specific ROI – with vmPFC during States, amPFC during Agents and dmPFC during Actions.

### Bayesian regression for ISFC

The Bayesian Regression for ISFC integration during updates was constructed and executed in the exact manner as for the ISC (and with the same MCMC settings), with the three Domains being replaced by PCN-PFC, PFC-HPC and Within-PFC Integration as the three 'Integration' terms to be tested.

We were mainly interested in hypothesis testing of which form of functional integration was higher in each domain. That is, if PFC-Precuneus has a higher ISFC than PFC-Hippocampus and Within-PFC during each of State, Agent and Action updates. We computed Bayes Factors for (and against) this hypothesis and their associated posterior probabilities.

For the Update vs Nonupdate comparison (see Belief update timecourses for how these points were computed), we used the same Bayesian hierarchical model as above, but with only the Precuneus to each domain-specific PFC during updates and nonupdate timepoints.

### Intersubject pattern correlation analysis

This analysis used ISPC to compare the timing of across-subject alignment of voxel patterns between the Precuneus and PFC regions.

We first obtained the ISPC (intersubject spatial pattern correlations) from the unsmoothed BOLD time-series. For each participant, this was achieved by extracting the multivoxel pattern vector at each time point, and correlating that with the group-mean (sans this participant), for a given ROI. This results in a time-course for this participant which reflects the spatially correlated pattern dynamics for that region. From this, for each participant (and for each ROI), we selected three different subsets of this timeseries. In each domain, we identified update points as described earlier and constructed a 9-scan window around them, resulting in as many segments as there were updates. We extracted data from these segments and concatenated them to give ISPC time courses for the periods of State, Agent and Action updates. To compare how similar the ISPC was across two regions, we then computed the correlation of these time courses across pairs of regions (e.g., computing Precuneus-vmPFC ISPC correlation for each update type).

The main ISPC analysis was done comparing each of prefrontal nodes with the Precuneus. To establish the specificity of the results, PFC regions were also compared with a number of other ROIs (VC, HPC, AG, MTL, RSC & PCC as shown in Supplementary Figs. 15–20).

We inferred multithreading if a region showed a domain-selective increase of similarity in ISPCs with the PFC. This similarity was obtained by computing the relative increase in correlation of a prefrontal region during its preferred updates (e.g., ROI- vmPFC ISPC correlation is higher during State updates than ROI-amPFC/dmPFC). Predictions were tested by computing contrasts between conditions of the same form as those used in the original whole-brain GLM. For example, for States this was achieved by comparing vmPFCStates - ((vmPFCAgents + vmPFCActions)/2) with amPFCStates - ((amPFCAgents + amPFCActions)/2) and dmPFCStates -((dmPFCAgents + dmPFCActions)/2). These contrasts compute the relative increase in ISPC correlation with the Precuneus for State updates and test whether this is higher for vmPFC than

the other two PFC regions. Similar logic was applied in Agents and Actions.

Significance was assessed via non-parametric permutation tests. We phase randomised the ISPC time-courses of the prefrontal nodes by applying Fast Fourier transform and sampling new phases (-π,π), which returns a permuted time-series with preserved autocorrelation structure as the original one, and kept the Precuneus (or control ROI) fixed.

The above-mentioned contrasts were computed in the permuted data (for 1000 permutations) to generate a distribution of expected values under the null hypothesis. As before, a $p$-value for the contrast was calculated using the proportion of permuted values that exceeded the true value.

### Seed-based ISFC

We used this method to investigate ISFC between the Precuneus and other brain regions during periods of State, Agent and Action updates. Precuneus BOLD time-series were extracted from the same ROI mask we used in all other analyses and compared to the group-average time-series of every other voxel in the brain. We used the AAL atlas to obtain a whole-brain mask. For computational tractability, voxels were resampled at 6 mm x 6 mm x 6 mm, such that the seed time course was compared to 6814 voxels throughout the brain. We computed the ISFC for periods of State, Agent and Action updates by employing a similar approach to that of the ISC, i.e., concatenating the 7-scan windows (3.5-scan before and after) around the update timepoints of each type. For each participant, ISFC between the Precuneus and each voxel was computed by correlating the participant's Precuneus BOLD time-series with the group-averaged voxel time-series (i.e., Precuneus to voxel). We then used the same approach to compute seed based ISFC for each of the PFC regions. vmPFC was used as the seed in State updates, amPFC was used as the seed in Agent updates, and dmPFC was used as the seed in Action updates.

Individual participant's correlation maps were averaged to give a group-level correlation map. These whole brain ISFC maps were FDR corrected at $p < 0.001$ (and visualised with a threshold of $r > 0.1$).

### Correlations between PFC-PCN coupling and shared responses in the Precuneus

The dynamic variant of ISC (spatial ISC/sISC) was computed as a leave-1-out ISC using a sliding window[44] of 21TR with 1TR overlap[48] by correlating the BOLD signal of each subject with the mean BOLD signal of the rest, for each window and for each region of interest (ROI). The Pearson correlation coefficients were averaged for each window, Fisher-z-transformed, and averaged across subjects, resulting in a single time course of mean ISC across the whole movie, for a region. We compared both the group-level as well as the individual-level ISC time series with the Arousal ratings (Fig. 7b, c). Whole-brain sISC had a strong positive correlation with arousal in a previous study[48].

For estimating the correlations between prediction integrations and experience, we obtained the correlations between Precuneus and each of the domain-specific PFC regions during the respective updates (from the ISFC analysis), and then averaged these to get a single value for prediction integration strength between PFC and the Precuneus during the updates. This measure was then correlated with the whole movie ISC values of the Precuneus (Fig. 7d). These correlations were also compared to the nonupdated timepoints (see Belief update timecourses for how these points were computed).

### Replication of key findings in spoken narrative dataset

We sought to replicate the two key findings observed in the movie - modular predictions and multithreaded integration. For this, we used the narrative data. Belief update time-courses were obtained from 3 independent groups of participants from Prolific, for States ($n = 16$), Agents ($n = 18$) and Actions ($n = 20$). The exact same instructions were

used (aside from changing the 'watching a short movie' part in the instructions to 'listening to a short story'). Smoothed updates can be seen in Supplementary Fig. 24a, with their reliability rating in Supplementary Fig. 24b.

Similar to the movie, 5 s update window around each update point were used generating a boxcar time-series for each participant, which were averaged and smoothed in the same manner.

The three domain-specific belief updates were used as predictors of the narrative neural data "It's Not the Fall that Gets You" (9 min 7 s). The exact same GLM structure used for the movie was deployed for this, with three resulting contrast maps (States > Agents + Actions, Agents > Actions + States and Actions > States + Agents). As mentioned in the original paper[44], the stimulus started after the first three scans. Statistical maps were thresholded at $q < 0.05$ FDR with an extent threshold of 25 voxels ($k = 25$), which was performed using NeuroElf.

For the multithreading/ISPC analysis, we obtained ISPC from the unsmoothed BOLD data. We obtained the update-time points akin to the movie as well, with one difference. The narrative used a θ of 1.2 SD threshold (see Supplementary Fig. 5 for details). We constructed the same 9-scan window around these update points as in the movie. Relative correlations between PCN ROI with that of domain-specific PFC regions were conducted and significance assessed via non-parametric permutation tests, where we phase-randomised the ISPC time-courses of the prefrontal nodes, as in the movie, while keeping the Precuneus (or control ROI) fixed.

ISFC maps were produced with the exact same procedure as in the movie, and maps were FDR corrected at $p < 0.001$ (and visualised with a threshold of $r > 0.1$).

### Reporting summary

Further information on research design is available in the Nature Portfolio Reporting Summary linked to this article.

## Data availability

The neuroimaging data analysed here are open source and available at (https://camcan-archive.mrc-cbu.cam.ac.uk/dataaccess/) and (https://openneuro.org/datasets/ds002345/versions/1.1.4), for the movie and narrative, respectively.

## Code availability

The processed data underlying all figures are provided as part of the Open Science Framework (OSF) https://osf.io/bpvj4/. The OSF project contains folders for movie and narrative with all source data files, detailing the file names and all scripts (R/Python) relevant for producing the figures.

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

## Acknowledgements

We'd like to thank Rob McIntosh, Chris Lucas, Chris Summerfield, Karl Friston, Joszef Fiser, Mihalyi Banyai and Adam Koblinger for helpful discussions. P.H. was supported by a BBSRC grant (BB/T004444/1). For the purpose of open access, the author has applied a Creative Commons Attribution (CC BY) license to any Author Accepted Manuscript version arising from this submission.

## Author contributions

Conceptualisation – F.Y. and P.H. Formal Analysis – F.Y. Methodology – F.Y., G.M. and P.H. Visualisation – G.M., F.Y. and P.H. Validation - F.Y., G.M., N.B. and P.H. Writing Original Draft – F.Y. Writing Review & Editing – F.Y., G.M., N.B. & P.H. Supervision – P.H. and N.B.

## Competing interests

The authors declare no competing interests.
