## [Transparent Peer Review file · Nature Communications]

Fragmentation and Multithreading of Experience in the Default-Mode Network

Corresponding Author: Dr Fahd Yazin

Version 0:

Reviewer comments:

Reviewer #1

(Remarks to the Author)

Using a movie-watching paradigm and fMRI data, Yazin and colleagues argued that humans possess three distinct world models—State, Agent, and Action—supporting domain-specific organization. They employed multiple analysis methods to explore how the DMN supports fragmenting (vmPFC, amPFC, and dmPFC) and multithreading (precuneus) of experiences. Their results showed that each prefrontal region processes distinct types of belief updates, while the precuneus selectively integrates these updated representations. They also replicated their results using a spoken narrative dataset.

I enjoyed reading this paper and believe that it addresses important and compelling questions about how humans model the world and integrate their internal models of different aspects of the world. However, I have concerns about whether their claims are sufficiently supported by the results presented in the study. Below, I outline several points that need to be addressed to enhance the clarity and robustness of the study.

1. It appears that the authors' definition of belief updating moments (instances when the group-averaged update probability exceeded 2 SD above mean) yielded very few initial datapoints. Given the 7TR window around these moments and "the State update timecourses consisted of 63 scans, Agents had 28 scans and Action had 35 scans", only 9 moments for state updates, 4 moments for agent updates, 5 moments for action updates were initially identified. Such a small and unbalanced number of moments across conditions (Agent and Action updating moments being approximately half of State updating moments) raises concerns about the robustness of subsequent analyses.

1-2. Fig. 1C suggests overlaps among the updating moments (e.g., around TR of 40-50, 70-80, 130-150, etc.). These overlaps would further reduce the number of non-overlapping timepoints across conditions, potentially affecting the reliability of observed differences across different updating moments. To establish the robustness of the results, the authors should provide additional analyses using alternative definition of updating moments and/or ensure more balanced number of timepoints across conditions.

1-3. The authors argue that low correlations (~0.3) between State, Agent, and Action update evaluations indicate domain-specific model updates occurring at distinct moments. However, this evidence alone may not sufficiently support their conclusion that updates occur at different moments. They should provide clearer descriptions of how specific updating moments were identified, indicating model updates in each domain, and consider controlling for potential collinearity between distinct belief updates to strengthen their argument.

1-4. The choice of a 7 TR window appears arbitrary. Analyses using multiple window sizes should be conducted to demonstrate the robustness of the results.

2. The authors interpreted Fig. 3b as showing "the proportion of belief updates that occurred immediately after neural state transitions in each PFC region, for a range of temporal window sizes." However, it remains unclear whether this result can support claims about the timing of corresponding cognitive processes. If it can, the authors should address the theoretical implications of why neural transitions occur earlier for State updates than Agent updates (relative to experienced shifts). A discussion on the potential reasons for these timing differences would enhance the interpretation.

3. The authors claimed that the precuneus selectively integrates updated prefrontal representations based on the result

presented in Fig. 4. However, several concerns limit the strength of this claim. The functional connectivity results in Fig. 4a do not seem to adequately support the claim that “prefrontal representations are integrated within the Precuneus.” The authors need to demonstrate that ISFC between the precuneus and each PFC region is significantly higher (1) specifically during update periods and (2) in a domain-specific manner. The current analysis appears limited to update periods. To support their claim, the authors should demonstrate that these effects are absent during non-update periods. Additionally, it is unclear whether these analyses support the domain specificity of PFC-Precuneus connectivity, as previously shown in Fig. 2. As the authors acknowledge, high PFC-Precuneus connectivity is expected and the ISFC between these regions might consistently exceed the other two integration models (within PFC and PFC-Hippocampus). To establish specificity, comparisons with functional integration of non-corresponding world models-PFC ROIs are needed (e.g., Agent update moments in amPFC-Precuneus vs. vmPFC-Precuneus, etc.).

4. While the analyses in Fig. 4b-d appear to partially address the concerns mentioned in 3, several questions remain. Given that ISPC analysis reflects population dynamics synchronization across participants, how does the similarity in synchronization profiles between the precuneus and PFC (Fig. 4d) support the claim that “the Precuneus selectively integrates updated prefrontal representations”? The authors should clarify how the results of synchronization of population dynamics relate to involvement of particular cognitive processes. Additionally, I have concerns about the interpretation of negative ISPC values. In the top panel of Fig. 4d, all ISPC values are negative. While the authors interpreted this as “Action updates elicited higher dmPFC ISPC correlations with Precuneus ISPC than vmPFC and amPFC,” these negative dmPFC-precuneus correlations might instead indicate a lack of population dynamics synchronization during Action updates rather than meaningful integration.

5. These concerns extend to the results in Fig. 5. The ISFC results between the visual cortex and the precuneus alone seem insufficient to support the claim that “the precuneus integrates updated representations with sensory regions.” If the movie stimulus included audio, the absence of similar results in the auditory cortex requires explanation, especially given the authors’ claim that the precuneus integrates updated representations with “sensory” regions broadly. Addressing these points would strengthen their argument.

6. Are there differences in how sISC (Fig. 6) and ISPC analyses should be interpreted? The authors employed multiple intersubject analysis methods, but their rationale for using specific analysis methods is unclear. For instance, could the sISC results in Fig. 6c-d have been obtained using ISPC analysis? If different methods were necessary, the authors should explain why each analytical approach was chosen for obtaining each specific result.

7. To replicate the findings in Fig. 2 and Fig. 4, the authors conducted the same analysis using a spoken narratives dataset (Fig. 7), stating that “this replication addresses potential limitations of using a movie stimulus and provides for an independent validation of some of the main results.” While this is an important step, I have concerns about whether the replication is successful. For example, the results for State updates did not show significant correlations between the vmPFC and precuneus, limiting meaningful outcomes of this analysis to Agent updates. Moreover, while the authors attribute the absence of significant differences for Action updates to “fewer characters in this story (two, one of whom narrates),” this explanation should equally affect Agent updates. This raises questions about the robustness of Agent update results.

8. The paper suggests directional relationships (e.g., PFC → precuneus → sensory cortex). However, an alternative interpretation could be that PFC regions selectively utilize integrated representations from the precuneus. Given that the analysis cannot determine directionality, a discussion that cautions readers against overinterpreting the findings regarding directionality would be important to avoid misinterpretation.

Minor points:

1. While this study focused on three distinct domains—state, agent, and action—primarily in social contexts, other important domains, such as time, objects, and relations, have been established in previous research (Bottini & Doeller, 2020; Ekstrom & Ranganath, 2018). Including a discussion of these domains would provide a more comprehensive acknowledgment of the diversity of internal models.

2. I found it challenging to fully understand the statement on line 535: “we correlated the average ISFC between the Precuneus and each PFC region during updates with whole-movie Precuneus ISC.” To help readers better understand the significance of this analysis the authors should explain the implications of correlating the average ISFC between the Precuneus and each PFC region during updates and the whole-movie Precuneus ISC.

3. Given existing research on cognitive maps (Behrens et al., 2018; Tang et al., 2023), the connection between the prefrontal cortex and the hippocampus would be expected to play a significant role in world model updating. However, this study highlights the importance of the precuneus-prefrontal cortex connection. A discussion on why similar results were not observed in the hippocampus would help provide further context and address this apparent deviation from prior literature.

4. In Fig. 3b, details of the statistical testing methods should be specified.

5. In Fig. 2a and Fig. 7a, color scales with threshold values should be added.

6. In Fig. 3, clarification is needed regarding whether the analysis accounts for HRF delay.

7. References 10 and 18 need correction.

References

- Behrens, T. E., Muller, T. H., Whittington, J. C., Mark, S., Baram, A. B., Stachenfeld, K. L., & Kurth-Nelson, Z. (2018). What is a cognitive map? Organizing knowledge for flexible behavior. *Neuron*, 100(2), 490-509.
- Bottini, R., & Doeller, C. F. (2020). Knowledge across reference frames: Cognitive maps and image spaces. *Trends in Cognitive Sciences*, 24(8), 606-619.
- Ekstrom, A. D., & Ranganath, C. (2018). Space, time, and episodic memory: The hippocampus is all over the cognitive map. *Hippocampus*, 28(9), 680-687.
- Tang, W., Shin, J. D., & Jadhav, S. P. (2023). Geometric transformation of cognitive maps for generalization across hippocampal-prefrontal circuits. *Cell reports*, 42(3).

(Remarks on code availability)

Reviewer #2

(Remarks to the Author)

In this work, Yazin and colleagues investigate whether the human brain processes internal models' updates in modular, domain-specific fashion or by means of a global integrated representation of events unfolding over time. Quite innovatively, they explored internal model representation using openly available fMRI data collected during visual (movie) and auditory (spoken narratives) naturalistic stimulation along with behavioral ratings of beliefs updates as predictors of neural activity. Through a robust analytical pipeline, the authors showed that three subregions in the human prefrontal cortex (vmPFC, amPFC, dmPFC) exhibit a domain-specific tuning for modelling beliefs about contexts/situations, characters beliefs/ goals and actions, while the precuneus, a core hub of the DMN, integrates such fragmented representations about the world into a coherent and integrated subjective experience. The rationale of the study is clearly articulated, the hypotheses and the aims of the work are characterized in detail. The analytical approach is appropriate to address the main questions and the implications of the major findings are interesting and may potentially have a large impact on the literature fostering further investigations in the field. I believe that the following point needs to be addressed to further improve the article:

1. Although the authors appropriately describes the experimental pipeline in the Methods section, I believe that adding an initial figure composed by different panels to illustrate i) the stimuli (movie and narrative), ii) the sensory modalities involved (multimodal audiovisual vs unimodal audio-only) along with iii) the experimental groups (e.g., 4 distinct groups for each stimulus presentation condition, number of participants, mean age) and iv) the tasks administered (i.e., fMRI, behavioral) would enhance the clarity of the analytical procedures, showcase the richness of the dataset and enable the reader to grasp important information at a glance.

2. Figure 2 and figure 7 show the brain maps of contrasts for each domain against the other two for movie watching and narrative listening respectively. In the panels a, cortical and subcortical regions are displayed. Since the analysis focuses on midline prefrontal areas (i.e., vmPFC, amPFC, dmPFC), I believe that showing the results of the contrast within a cortical mask -so to get rid of subcortical activations, which to my understanding are not the main topic of the work- or within the PFC mask solely (made by the conjunction of the regions of interest) would make these results more readable, clean and elegant.

3. In the text the authors report: "Group-level ratings were only weakly correlated with one another (States vs Agents, $r = 0.31$, Agents vs Actions, $r = 0.29$, Actions vs States, $r = 0.28$). This indicates that models in different domains were being updated at different times during movie-watching". However, visual scrutiny of Figure 1c (in which three smoothed update probability time-courses are represented) shows that peaks of at least two distinct domains often coincide (for instance approximately at 48, 50, 80, 100, 130, 160 TR) or are just slightly delayed. In my opinion this observation indicates that, to some extent, the occurrence of events pertaining to distinct domains shows a non-negligible overlap over time. To this regard, although correlation coefficient plays a pivotal role in quantifying linear relationships between variables, its application to time series data is very challenging due to possible temporal dependencies. A viable alternative could be to compute the cross-correlation since it can effectively show how changes in one time series relate to changes in another taking into consideration slight differences in time lags. Other metrics such as mutual information could be used as well to better understand how much information is shared between timeseries beyond correlation. Additionally, given the intricate relatedness of features in naturalistic stimulation, it could be worthy to evaluate whether and to what extent the three domains share common underlying information which may cause them to be collinear to a certain degree. For instance, changes in the context, actions and agent beliefs, may partially overlap with scene transitions, which mark pivot, break points in the unit of storytelling. Asking a few new raters to tag scene changes in the movie clip, would be sufficient to test whether these events are collinear with the changes in belief the authors are studying in the present work.

4. To assess inter-rater reliability the authors performed a split-half correlation procedure. The values reported in the manuscript indicates that the tags assigned across the two subgroups of raters are fairly consistent for the States domain ($r = 0.80$), while Agents and Actions show fairly low values ($r = 0.49$, $r = 0.57$ respectively). I was wondering whether this could originate from the difficulty raters experienced in identifying changes in beliefs in the tale and in labelling them into separate, distinct domains. For instance, reading the instructions the authors provide in the Supplementary Material (table 1), I framed the domain "States" as indicating an unpredictable event, changing my expectation or "cognitive script" of a typical situation (such as in the restaurant example provided). However, I had troubles in distinguishing it neatly from the "Actions" domain given the fact that changes in the context likely trigger unpredictable switches in the following actions performed by the characters. Did you perhaps train the subjects on a different movie clip and collect their ratings before the actual experiment? This could provide an interesting index of tagging accuracy to evaluate the "goodness" of subjective ratings. Going on with this, even if the split-half correlation is a common and robust method, it would be interesting to combine it with other metrics to gain a comprehensive understanding of the consistency both within the tagging system and among raters. Moreover, to account for the fact that half a test (i.e., half the raters pool) is evaluated, I think the author might want to consider to adopt the Spearman-Brown formula to adjust the split-half correlation upward to estimate the reliability of the full test.

5. At page 19, the authors adopted a dynamic intersubject correlation analysis (dISC) and tested whether the ISC in the precuneus computed over a sliding window, correlates with the emotional Arousal ratings. I suggest the authors to include more details about this analysis such as for example the window width and the time step they used and to motivate better why correlation with arousal means that the region is tracking a unified experience. Another approach I think is suited to talk about "unified vs fragmented" experience, is to create window sizes of increasing width, ranging from the actual TR (2.470 seconds, which leads the same results of the classical ISC pipeline) up to, let's say, 1 minute. This technique has been widely exploited in the field of naturalistic stimulation (see Hasson et al., 2008) with the name of Temporal Receptive Windows (TRW) analysis. Using this approach, a high ISC over a short window (that is, few seconds) would suggest that the correlation was modulated by rapidly changing events, whereas high ISC values over longer segments (that is, tens of seconds) will indicate that the correlation mostly relied on accumulating information (that is the "unified experience"). Therefore, following the authors' hypothesis, the precuneus should display higher ISC values over long windows, while the three prefrontal regions should show greater ISC values over short time windows.

6. As new events and experiences are encountered, the human brain possesses the remarkable capacity to adapt and transform mental representations of people, events, and environments to align with the dynamically changing world. Not only can these representations be modified, but they can also be assigned to a mental timeline, creating a sense of temporal coherence. Further, the presence of new information can prompt individuals to revise and update existing narrative representations, allowing for a flexible and adaptive understanding of personal stories. The change of beliefs in all the three domains (i.e., scene, agent and actions) do coincide with crucial points in the narrative, which can be conceived as "plot twist". This moments in the story capture the viewer/listener alertness by defying expectations and trigger, as the authors discuss, both emotional engagement and cognitive reappraisal. Therefore, evolving cognitive demands would be required to process these "points of rupture" in the narrative in order to make sense of the plot evolving through time. For instance, goal-directed (attributed to the influx of plot-relevant information) and stimulus-driven visuo-spatial attention, semantic (influx of new plot-relevant information, requiring a solid semantic structure to build forthcoming narrative elements) and episodic (increase in information being integrated and updated into long-term memory as the narrative progresses) memory would sustain the formation of an integrated representation. Altogether, distinct cognitive processes inherent to the narrative would give rise to dynamic alterations in neural synchrony within the attentional, semantic, and episodic networks. How would the authors expand the discussion about the results of the present work in light of these considerations? I believe that adding a paragraph in the discussion section about these aspects would enrich the manuscript and prompt further debate.

Minor comments

1. Supplementary Figures 7-12 need a proper caption which briefly comments the results presented.
2. Try to fix where possible the variations in terminology adopted. In my opinion, having a consistent lexicon through the text helps the reader and makes the final message clearer. For example, I suggest to change "State"- "Agent"- "Action" Updates (e.g., lines 406-411) with the plural form "States – Agents – Actions" as stated in the figures and other parts of the text. The same holds for "Supplementary Figure" (line 203) or "Suppl Fig" (e.g., lines 512, 977).
3. Figure 5: since the brain mesh is a pale, light-grey color, I would recommend to use a different colormap to represent the results of the ISFC analysis. A color palette going from red to black or from blue to green excluding the white component (for instance the matplotlib palettes viridis, plasma, inferno, magma or the sequential Color Brewer palettes) would enhance the visual impact and the readability of the connectivity maps.
4. Some figures present the ISC values for the three PFC regions by means of bar plots. I believe that the order of regions on the x axis of the bar plots in Figure 2- Panel b, Figure 4 - Panel d, and Figure 7 – Panel b, should be the same to aid visual comparability. This will be extremely useful in particular for Figure 2 and 7 where the authors show the generalizability of results across content and sensory modalities.

(Remarks on code availability)

I reviewed the code without re-running it. The folders structure is well organized and reflects the analytical steps reported in the paper, thus representing a useful resource for the community.

Reviewer #3

(Remarks to the Author)

Using a publicly available audio-visual movie fMRI dataset, the authors show that three midline frontal brain regions respond selectively to different types of narrative changes, suggesting involvement in the update of state, agent, and action components of situation models. It is further shown that these frontal regions are more functionally connected with the precuneus than with the hippocampus or with each other, and an innovative analysis relates this to correlated consistency of activation patterns. Precuneus activity transitions are also correlated with subjective arousal ratings. A subset of these findings are replicated in the second publicly available audio-only dataset, showing consistent results for states and agents, but not for actions.

The paper is generally well written and presented. The topic and results are interesting. The methods and statistics are sophisticated and generally appropriate (but see specific queries). Overall, I enjoyed reading the manuscript and I feel it will make a valuable contribution to the literature. However, I have several concerns and queries as detailed below.

1. I wonder whether the claim “it is unknown whether humans acquired an assembly of many separable and highly specialized models” (e.g. line 45-46) might be overstated. There is little doubt that the brain is somewhat modular with domain-specific specializations. In particular, in the context of narrative understanding (both reading and movie-watching), it has long been known that different brain regions track different aspects of a narrative, including locations, characters, and characters’ goals, e.g. Speer et al., 2009, “Reading Stories Activates Neural Representations of Visual and Motor Experiences”, and Zacks et al., 2010, “The brain’s cutting-room floor: segmentation of narrative cinema”. The proposal in these studies is also that prediction failures based on various feature changes trigger updates of situation models, and the brain response at these changes mediates subjective ratings of event boundaries. It has also been proposed that midline parietal DMN has a particular role in integrating event models and incoming sensory information, through coupling with midline frontal regions (e.g. Stawarczyk et al., 2021, “Event Representations and Predictive Processing: The Role of the Midline Default Network Core”). That said, the current findings of differentiation within the medial frontal lobe, the particular framework of states, agents and actions, and functional connectivity with the precuneus, provide a novel and useful addition to this literature.

2. Throughout most of the manuscript, key domains of world knowledge are defined as “states”, “agents” and “actions”. It is confusing that the abstract does not use these terms but instead refers to “spatial”, “referential,” and “temporal” domains. Some of these seem more specific (“spatial” vs “states”); others more general (“referential” vs “agents”). “Spatial” is especially confusing, because the instruction given to participants (Supplementary Table 1) uses an example of a character change within a fixed spatial location. I think it would be more consistent to use “states”, “agents” and “actions” in the abstract, and if using alternative terms then clarify their relation in the introduction. (This issue also arises on line 613: I disagree that states are necessarily spatial, and I’m not persuaded that actions are necessarily “temporal”.)

3. Although the theoretical focus is on medial prefrontal cortex, in none of the figures does this brain region seem particularly unique. E.g. in figs 2, 5, & 7 responses that are at least as strong tend to be seen across many other brain regions. I do like the story of a tripartite division within mPFC, but I wonder whether this ignores a bigger picture.

4. Some methodical choices are unclear, especially when these vary across analyses. E.g. why was a 5 s window chosen for combining update ratings, a 7 TR window for the ISC analyses, and a 9 TR window for the ISPC analyses? Why use frequentist statistics for the activation analyses and Bayesian statistics for the ISC analyses? Why use FDR correction for results in Fig 2 but not results in Fig 5?

5. For several of the statistical tests, I am concerned about potentially inflated significance due to non-independence of the samples. Whenever significance is based on correlations across time, autocorrelation can bias significance if not accounted for. I think this applies to the HMM analyses and ISPC analyses, where random shuffling of time-points would be invalid if it does not preserve temporal autocorrelation. Circular shifts and phase shuffling could be permutation approaches that better preserve autocorrelation. A related but different concern arises for the Bayesian regressions of ISC and ISFC, and seed-based ISFC: although here the significance is based on variation across subjects, the values per subject are not fully independent because the leave-one out approach means that there is overlap across folds (as noted in reference 41).

6. I do not agree with the statement on line 221-222 that the activation effects seen in the GLM do “not confirm that these effects are truly driven by the movie stimulus alone (rather than stimulus-unrelated internally driven thoughts).” Stimulus-unrelated activity would be temporally idiosyncratic and so average out in the GLM. Responses that are statistically significant in the cross-subject GLM must be consistent across subjects and therefore evoked by the stimulus (or something that is time-locked to it).

7. Related to the above point, the GLM analysis, ISC analysis, and the HMM analysis do not seem independent, but rather three slightly different ways of probing related aspects of the same data. I.e. having shown that subjects’ activity time-courses on average correlate with behavioral ratings (in a domain specific way), it seems necessarily the case that subjects’ activity time-courses will correlate with each other (ISC) in a similar way, and that transitions extracted from them (HMM) will similarly correlate with the ratings. A similar point about non-independent analyses applies to those in Fig 6 panels a and b.

8. Regarding the HMM analyses, several aspects were unclear to me:

a. Is it surprising that only two (or three) latent states were selected per ROI (which seems to suggest a binary response per region)?

b. Is it a problem that the inter-run time-series correlation for dmPFC is close to zero (panel A of supplementary figure 5)?

c. I would also query the description on line 290 of “exhaustively bootstrapping”... If I understood correctly, 30 random samples were taken, which is far from exhaustive, and samples were not taken with replacement, as would normally be the case for a bootstrap?

d. How can there be any belief updates within a time window of zero scans?

- e. Line 306 says, “This suggests that the subjective experience of an update to predictions occurs sometime after the neural model is reconfigured”, however this could not really be otherwise when only considering time windows that follow rather than precede the neural transition? Relatedly, the preceding GLM implies, instead, that the neural effects occur sometime after the belief updates. Also, lines 875-7 state, “Where a sequence of multiple TRs/scans exceeded θ , we selected the final TR/scan in the sequence to represent the update.” This decision will bias the belief updates to be later in time, relative to the neural signal. Further, there will be some reaction time between when a belief update occurs and when a participant makes the button press, and this does not seem to have been measured or accounted for. Overall, any inferences about temporal precedence seem to be on shaky ground.
- f. Being pedantic, the word “discrete” on lines 308 & 682 is not really justified because discreteness is an assumption of the HMM rather than something being tested.
- g. Finally, the claim on line 310 that the neural shifts “mediate” the belief shifts is not statistically tested; the presented results simply show a correlation.
9. Line 376 states “...above results suggest prefrontal representations are integrated within the Precuneus.” However, essentially this is a functional connectivity analysis based on correlated activation, so doesn’t really say anything about “representations” or “integration”, just that the frontal regions are more correlated with the precuneus than with the hippocampus or with each other. The following section describes some clever ISPC analyses that do get closer to representational integration, so the phrasing here just needs to be a bit more careful.
10. In Figure 4, how do you interpret the significantly negative ISPC correlations for action updates? (In panel d, error bars are not defined, and it would help to order the x axis to match panel b of Fig 2. Also, some error bars are missing from supplementary figures 7-11.)
11. For the seed-based ISFC (Fig. 5), can you rule out the possibility that visual cortex was more correlated with precuneus than frontal seeds simply due to anatomical proximity?
12. I was very confused by the sections “Similar Predictions Accompany Similar Experience” and “Correlations between Subjective Experience and Predictions,” because the headings talk about “predictions” and “experience,” whereas the described analyses seem to involve neither of these things, instead showing that [precuneus with prefrontal connectivity] correlates with [precuneus inter-subject consistency]?
13. The results on arousal come rather out of the blue, having not been mentioned in the abstract or the introduction.
14. In what sense are activation profiles “mapped to a ventral-dorsal gradient” (line 614)? Does that mean that functional preferences are smoothly varying (a necessary consequence of smoothing during preprocessing)? Or that “agent” representations are intermediate between “action” and “state” representations along a continuum (what would that mean)?
15. The term “fragmentation” in the title suggests decomposition of an initially unified model. However, the manuscript itself generally suggests an opposite flow, with an integrated model being built in the precuneus from initially separate components. Some clarification of when models are seen as fragmenting vs integrating might be helpful. More generally, the paper suggests (implicitly and explicitly) causal relationships such that frontal regions influence the precuneus and that these regions influence sensory activity and conscious experience. However, since all analyses are correlational, it seems possible that these relationships might be reversed, be bidirectional, or even independent.
16. The abstract claims “Results generalized across sensory modalities... suggesting...abstract, modular predictive models for both vision and language.” Although this is likely, both datasets include audio/language, so it seems possible that language-based models could account for all findings, with minimal evidence of whether they might apply in a purely vision-based situation?
17. For the Bayesian analyses, what prior effect size distributions were assumed?
18. In Fig 2, panel A is missing a color bar, while the color bar in panel C seems to be neither defined nor used. In panel B, the error bars are not defined.
19. Discussion of links to consciousness, and to blind-sight, spatial neglect, and commissurotomy-related phenomena seem rather tangential.
20. The Cam-CAN dataset is usually treated as a lifespan cohort. It would be useful to state that only a subset of the full dataset was analyzed here, and mention the reason for choosing the age cut-off at 35 years. For this dataset, what was the voxel size and slice coverage?
21. When theta is defined in terms of SD from the mean, is this the mean across participants per time-point, or the mean across time for the group-average?

(Remarks on code availability)

Reviewer #4

(Remarks to the Author)

This study investigates the neural basis of predictive processing and integration in the brain, focusing on how different prefrontal cortex (PFC) regions and the Precuneus contribute to forming a unified subjective experience. I found this to be a creative study addressing an important question in cognitive neuroscience. However, the study suffers from several critical issues that significantly undermine the validity and strength of its conclusions. The authors frame their work within predictive coding and Bayesian inference models. However, the connections to these theoretical frameworks appear tenuous and poorly substantiated. The study’s correlational nature precludes causal claims about predictive coding mechanisms, and the presented Bayesian inference models lack robust justification. More rigorous analyses, including direct measurements of prediction errors and clear temporal sequencing, are necessary to validate the proposed mechanisms and strengthen the claimed links to predictive coding, Bayesian inference frameworks, and cognitive maps.

The core proposition – that humans model the world by partitioning it into domains of states, agents, and actions,

represented topographically in the PFC and integrated in the Precuneus – is intriguing but inadequately supported by the presented data. The evidence for integration in the Precuneus, a crucial aspect of their model, is not convincingly demonstrated and no rationale is provided for overall analysis approach is not clear or justified. The heavy reliance on correlational analyses fails to establish the causal relationships implied by their predictive coding framework.

Key Methodological Issues:

1. Prediction Error Analysis: The authors claim "Prediction-error-driven neural transitions... preceded subjective belief changes in a domain-specific manner." However, the study lacks any direct analysis or measurement of prediction errors, a fundamental omission given the centrality of this concept to their claims.
2. Temporal Precedence: Despite assertions about neural transitions preceding subjective belief changes, no specific analysis demonstrates this temporal relationship. The absence of clear temporal analysis to establish the sequence of neural transitions and subjective experiences is a significant limitation.
3. Event Concatenation: The concatenation of events across update segments is problematic, potentially introducing artifactual relationships in the data. It's unclear whether these events were correlated across the three types (states, agents, actions).
4. Replication and Generalizability: The replication results, particularly as shown in Figure 7, are weak and do not strongly support the authors' claims. The inconsistency in the number of update scans across domains (63 for States, 28 for Agents, 35 for Actions) raises questions about the comparability of these measures and the potential influence of estimation statistics.
5. Inter-Subject Correlation (ISC): The rationale for using ISC rather than activation to each event type is not clearly justified, especially considering that event detection might differ across individuals. This approach may obscure important individual differences in cognitive processing.
6. Participant Consistency: It is unclear whether the measures of agent, state, and action updates are consistent between the participants who provided these ratings and those who underwent fMRI scanning, potentially introducing significant confounds. The reliability of ratings is weak and reliability of event segmentation in the second task is not shown.

Other/Minor Issues:

Citations to key relevant manuscripts on DMN and PFC function are missing.

The presentation of results could be improved, for instance, by including color scales in figures to enhance clarity.

Brain areas included for ROI analysis are not shown.

Figure 1: Inter-rater reliability is shown as ISC.

(Remarks on code availability)

Reviewer #5

(Remarks to the Author)

(Remarks on code availability)

I tried to run the experimental code, but it seems that the dependencies of the R libraries are incorrect. I installed all the libraries, but the code didn't work.

Version 1:

Reviewer comments:

Reviewer #1

(Remarks to the Author)

I appreciate the authors' detailed and thoughtful responses to the previous comments from all reviewers. The revised manuscript shows substantial improvements in both methodological clarity and interpretational precision. The additional analyses, expanded explanations, and careful revisions indicate that the authors have taken the feedback seriously and addressed each point with considerable care. I sincerely appreciate the time and effort the authors have invested in

thoroughly responding to all of the comments.

However, I believe several important concerns remain insufficiently addressed. Below, I outline the key issues that, in my view, require further revision before the manuscript can be considered for acceptance.

Major points

1-2. The authors argue that the use of a 2 SD threshold strikes a balance between the number of belief update moments and the degree of temporal overlap across domains. While I appreciate the attempt to justify this parameter choice, the supplementary figures provided indicate that overlap across domains persists even at the 2 SD level (~25%). This raises a concern about whether the belief updates for each domain can truly be considered distinct, as the authors claim.

Given this, I believe additional analyses are needed to further validate the domain specificity of belief updates. First, the authors should perform analyses that exclude timepoints that overlap between domains and show whether their main findings still hold under this more conservative condition. Second, it would be important to examine whether the main results replicate when using alternative thresholds, such as 1.6 SD. Based on the figures provided, the 1.6 SD threshold appears to yield more distinct domain-specific update patterns than the 2 SD threshold. If the key findings are sensitive to the choice of threshold, this should be acknowledged and directly addressed in the manuscript.

1-4. While I appreciate the authors' effort to test the stability of the ISC and ISPC results across multiple window sizes (Figure 3C, Figure 5D, Supplementary Figures 10 and 14), I remain concerned regarding the robustness of the findings. Specifically, the domain-specific effects highlighted in the manuscript appear to be most consistent at a 7-TR (or now 9-TR?) window. At shorter or longer window sizes, the effects noticeably weaken or become inconsistent, raising questions about whether these findings reflect a stable phenomenon or are dependent on specific analytical choices. The GLM analyses already show strong domain-specific effects using continuous update ratings—suggesting that the inclusion of ISC/ISPC results, particularly given their window-size sensitivity, may not substantially strengthen the manuscript's main claims. To clarify the contribution of these results, it would be helpful for the authors to articulate what unique insights ISC/ISPC analyses offer beyond the GLM findings and to address the extent to which these effects can be considered robust across a range of analytical choices.

3. While the authors provided additional analyses addressing domain specificity of PFC–precuneus coupling, the question of whether this coupling is specific to belief updating periods remains unresolved. The authors chose not to compare update and non-update periods directly, citing an imbalance in the number of timepoints. This concern could be addressed by randomly sampling non-update timepoints to match the number of update periods, which would help clarify whether the observed coupling is update-specific or instead reflects more general characteristics of the data.

4. In both Figure 5D and Figure 8B, ISPC correlations during Action updates are consistently negative across PFC regions, with the dmPFC showing values that are less negative or closer to zero. The authors interpret this as evidence of domain-specific coupling between the dmPFC and the precuneus. However, negative ISPC values are inherently difficult to interpret, and the manuscript does not clearly explain what such negative (or less negative) correlations mean, either mechanistically or functionally. If these values are intended to reflect desynchronization, the authors should clarify what desynchronization entails in this context and how it specifically relates to Action updates and belief updating more broadly. A review of relevant literature documenting similar patterns and proposing plausible neural mechanisms that might underlie this phenomenon would strengthen the claim about dmPFC's selective role in Action updates.

5. The authors argue that the absence of precuneus–visual cortex ISFC in the story-based dataset supports that the precuneus integrates updated representations with sensory regions. While this finding is informative, I believe this conclusion remains incomplete without considering the auditory modality. Because the story dataset is presented entirely through auditory input, one might expect to see increased coupling between the precuneus and the auditory cortex. However, no such effect is reported or discussed.

As a result, the current pattern of findings appears to be specific to precuneus–visual cortex connectivity rather than broadly supporting the idea that the precuneus integrates with sensory regions. If the authors wish to maintain this broader interpretation, I believe it is necessary to show that such integration is flexible across modalities or revise their claim to reflect the more specific nature of the observed effect.

6. While I now better understand the rationale for employing multiple intersubject analyses such as ISC, ISPC, and sISC, this rationale is currently presented in a fragmented manner throughout the manuscript. Given the diversity of analytical approaches used, it would greatly benefit the reader to include a brief overview—perhaps in the Methods section or at the beginning of each finding—explaining why each method was chosen and what kind of hypothesis or neural property it is intended to capture. This would enhance readability and help readers more easily navigate the analyses, while also highlighting the unique strengths of each approach.

Minor points

While most of the minor points have been addressed, some issues would still benefit from additional clarification.

2. I would like to raise a potential concern regarding the interpretability of the correlation between PFC–precuneus ISFC and precuneus ISC. Because both measures are derived from activity within the precuneus, the observed relationship may reflect intrinsic properties of this region—such as signal stability or general responsiveness—rather than intersubject convergence in belief updating. If the authors intend to use this correlation to support claims about representational integration, alternative explanations should be ruled out or explicitly acknowledged.

Furthermore, the authors suggest that individuals with stronger or more similar “integration profiles” with PFC also exhibit more similar precuneus activity across participants. However, it is not clearly explained why the degree of functional coupling should necessarily translate into shared processing or subjective experience at the representational level. If this correlation is intended to support claims about shared belief updating or predictive coding, I encourage the authors to clarify this interpretational link more directly.

(Remarks on code availability)

Reviewer #2

(Remarks to the Author)

The authors successfully addressed all the issues and points I asked for during the first review of the article. I'm happy with the new version and I want to congratulate all the authors for the tremendous amount of work!

(Remarks on code availability)

Reviewer #3

(Remarks to the Author)

The authors have thoughtfully and adequately addressed my previous concerns and queries. I am now left with four relatively minor comments:

1. Error bars still appear to be missing from some bars in supplementary figures 15-19, but I now wonder whether this is because the line width is too thin for them to render consistently?
2. Supplementary figure 10 is redundant, because all information is also present in supplementary figure 22.
3. My previous comment 9 concerned the interpretation of functional connectivity between precuneus and frontal regions, but I realise that the same point applies to functional connectivity between precuneus and visual cortex in the section beginning on line 455. I.e. I don't think one can conclude "integration of representations" from what are merely correlated time-courses. Although I wonder why the more interesting ISPC method used for frontal connectivity was not applied here, I feel it would be sufficient just to tone-down the claims in the section heading. (Similar claims within the text of the section are already expressed sufficiently cautiously.)
4. The most substantive comment concerns the new analysis of precuneus functional connectivity in the replication dataset. One claim of the paper is that "the precuneus integrates updated representations with sensory regions," with evidence including functional connectivity between precuneus and visual regions (previous comment notwithstanding). The new analysis in Figure 8c usefully shows that that such connectivity is weaker for the audio-only narrative. However, the general hypothesis surely predicts connectivity between precuneus and auditory regions in this case, which is not apparent? Should the conclusion, then, be that precuneus is specifically coupled with visual regions, rather than sensory regions more generally? I feel this deserves at least some acknowledgement/consideration in the results/discussion.

(Remarks on code availability)

Reviewer #4

(Remarks to the Author)

The authors have done a great job of addressing reviewer concerns, including mine. I think a limitations section would be useful for the reader, given the substantive conceptual issues raised and the clarifications the authors have provided (e.g. prediction error implied from belief update etc). In parallel, readers will benefit from the reviewer response so the strengths/weaknesses and issues raised can advance future research.

(Remarks on code availability)

Reviewer #5

(Remarks to the Author)

(Remarks on code availability)

While I did not run the full analysis code, it appears well-structured and reasonably documented.

Version 2:

Reviewer comments:

Reviewer #1

(Remarks to the Author)

The authors have adequately addressed my concerns, and the manuscript has been strengthened.

(Remarks on code availability)

As the entire analysis code currently runs in a single block, I recommend partitioning the code into distinct sections by

analysis step to facilitate individual review of each result and enhance usability.

Reviewer #5

(Remarks to the Author)

(Remarks on code availability)

Reviewer comments are in Black. Our responses are in Blue. Changes/quotes from the revised manuscript are in red. Quotes from other references are in green where applicable.

REVIEWER COMMENTS

Summary of changes

We thank all the Reviewers for their extensive comments on methodology and interpretations. We have made a number of major revisions to the paper, which we explain in detail in the responses below. Here is a summary of the major changes:

1. **Update Threshold Selection.** In the original submission, we used a threshold of 2SD to identify belief updates from the movie-rating data and 1SD for the narrative. We have now investigated alternative thresholds for selecting updates (Supplementary Fig 4 & 5).
2. **Window Size Analysis.** Some parts of the study involve analyses of a “window” of scans following a belief update, and Reviewers questioned whether the choice of window size would affect the results. In the ISC and ISPC analysis, we now report effects for a range of window sizes.
3. **Story Replication.** We have performed additional analyses on the audio story data that we treat as a replication dataset. In the course of doing this we found that in our original analyses, the ratings were misaligned with the fMRI data by 3TRs. We have now corrected this error, which means that all of the results presented in Figure 8 have changed slightly.
4. **Rating Robustness.** We have performed additional analyses of the reliability and consistency of participants’ ratings of belief updates.
5. **Summary Figure.** In response to Reviewer 2’s suggestion, we have added a new figure (Figure 2) that illustrates our experimental and analytic approach.
6. **Temporal Claims.** Several Reviewers questioned our statement that neural activity changes precede the reported experience of a belief update. This statement was not central to the claims we are making in the paper and we have now removed it, as our data do not permit this kind of inference about temporal precedence.

Reviewer #1 (Remarks to the Author):

Using a movie-watching paradigm and fMRI data, Yazin and colleagues argued that humans possess three distinct world models—State, Agent, and Action—supporting domain-specific organization. They employed multiple analysis methods to explore how the DMN supports fragmenting (vmPFC, amPFC, and dmPFC) and multithreading (precuneus) of experiences. Their results showed that each prefrontal region processes distinct types of belief updates, while the precuneus selectively integrates these updated representations. They also replicated their results using a spoken narrative dataset.

I enjoyed reading this paper and believe that it addresses important and compelling questions about how humans model the world and integrate their internal models of different aspects of the world. However, I have concerns about whether their claims are sufficiently supported by the results presented in the study. Below, I outline several points that need to be addressed to enhance the clarity and robustness of the study.

Thank you for the thorough review and kind words.

(We combine 1 and 1-2 together for a joint response)

1. It appears that the authors' definition of belief updating moments (instances when the group-averaged update probability exceeded 2 SD above mean) yielded very few initial datapoints. Given the 7TR window around these moments and "the State update timecourses consisted of 63 scans, Agents had 28 scans and Action had 35 scans", only 9 moments for state updates, 4 moments for agent updates, 5 moments for action updates were initially identified. Such a small and unbalanced number of moments across conditions (Agent and Action updating moments being approximately half of State updating moments) raises concerns about the robustness of subsequent analyses.

1-2. Fig. 1C suggests overlaps among the updating moments (e.g., around TR of 40-50, 70-80, 130-150, etc.). These overlaps would further reduce the number of non-overlapping timepoints across conditions, potentially affecting the reliability of observed differences across different updating moments. To establish the robustness of the results, the authors should provide additional analyses using alternative definition of updating moments and/or ensure more balanced number of timepoints across conditions.

We agree that the method used to identify update points is important and that we did not provide sufficient justification for this. In the movie data, we defined an update as occurring when the update probability exceeded two standard deviations of its mean. The Reviewer raises two important concerns about this threshold. First, it should identify a sufficient number of updates in each domain to conduct analyses. Second, it should avoid too much overlap in the times when different domains overlap.

We have now investigated how a range of different thresholds (from 1SD to 3.2SD) meet these requirements. As the threshold increases from 1 SD to 3.2 SD, fewer updates are identified in the data. This is shown in Supplementary Figure 4 for per domain (left) and as an average across

domains (middle). Thus, using a lower threshold helps to find more updates. However, lower thresholds have a negative effect on selectivity, as the updates in different domains overlap one another more frequently. This is shown in Supplementary Figure 4, right (here we implemented a sliding window approach (10TRs) to quantify cross-domain overlaps. An overlap occurs when at least two updates from different domains fall within a window).

As shown in Supplementary Figure 4, more liberal thresholds (less than 2SD) identify more update points during the movie, which is desirable. However, they also result in greater overlap between domains, which is undesirable. Conversely, very conservative thresholds avoid overlap but the number of updates decreases, particularly for Actions. Thus, in selecting a threshold of 2SD we balance these.

We performed a similar analysis on the Story data (in Supplementary Figure 5) and here found less overlap, allowing us to use a more liberal threshold of 1.2SD for these data (previously used 1SD).

Finally, we note that the whole-brain GLM analyses (Fig 3a,b and Fig 8a) used the continuous update probabilities as parametric modulators, and thus are independent of the thresholding decisions discussed here. These analyses shown clear domain-specific patterns of PFC recruitment, indicating that effects are not dependent on a particular method of isolating belief update moments.

All these figures are now in the Supplementary Figures 4 & 5

1-3. The authors argue that low correlations (~ 0.3) between State, Agent, and Action update evaluations indicate domain-specific model updates occurring at distinct moments. However, this evidence alone may not sufficiently support their conclusion that updates occur at different moments. They should provide clearer descriptions of how specific updating moments were identified, indicating model updates in each domain, and consider controlling for potential collinearity between distinct belief updates to strengthen their argument.

To be clear, we do not believe there is a unique set of moments in the movie when belief updates occur. Instead, we view model updating as a continuous process happening throughout the movie, but the likelihood of updating varies at different moments. Hence our initial GLM analysis that treats update as a continuous variable (for movie & narrative) and since all three variables are included simultaneously in the model, it accounts for potential collinearity.

However, we also sought to investigate what is happening at times when the most significant updating is occurring. Thus, we used our thresholding approach to identify times when the most participants are aware of an update occurring. Our further analyses based on this approach show clear and specific relationships between updates in different domains and activity in different PFC sub-regions.

Additionally, we have performed extensive analyses of the relationships between the update timecourses in the three domains, which indicate they are only weakly correlated with one another. In addition to the correlations presented in Figure 1, we also performed a lagged cross-correlation analyses among States, Agents and Actions ratings as well as mutual information analyses; as was suggested by Reviewer 2. MI values were low (Supplementary Figure 2) and cross-correlation (Supplementary Figure 3) did not show a strong influence of one rating on the others.

1-4. The choice of a 7 TR window appears arbitrary. Analyses using multiple window sizes should be conducted to demonstrate the robustness of the results.

We agree and have therefore investigated the main intersubject analysis (ISC & ISPC) over a range of window sizes.

We have conducted the ISC analysis (3c in the main text) over a range of windows and found them to stay consistent in their domain preference during synchronization in each update (Supplementary Figure 10).

We have also done the same window analysis for the Movie ISPC (5d) over a range of windows, and found the general trend to remain stable for windows up to 10TR in length (Supplementary Figure 14).

We repeated the same for the Story ISPC (8b) and found similar results (Supplementary Figure 23).

2. The authors interpreted Fig. 3b as showing “the proportion of belief updates that occurred immediately after neural state transitions in each PFC region, for a range of temporal window sizes.” However, it remains unclear whether this result can support claims about the timing of corresponding cognitive processes. If it can, the authors should address the theoretical implications of why neural transitions occur earlier for State updates than Agent updates (relative to experienced shifts). A discussion on the potential reasons for these timing differences would enhance the interpretation.

Several Reviewers commented on this. With hindsight, it’s clear we overstated this aspect of our results. In fact, we did not intend to make any strong claims about relative timing differences. We suspect an M/EEG+task study would be better suited for this question than our fMRI+movie data.

Accordingly, we have adjusted the wording in several places in the paper

We changed the wording in abstract from preceded to coincided with

“Prediction-error-driven neural transitions in these regions, indicative of model updates, coincided with subjective belief changes in a domain-specific manner. “

We changed the result title in Figure 4 (new version) similarly

“Neural Transitions Coincided with Subjectively Experienced Belief Updates”

We have now removed this sentence

“This suggests that the subjective experience of an update to predictions occurs sometime after the neural model is reconfigured.”

3. The authors claimed that the precuneus selectively integrates updated prefrontal representations based on the result presented in Fig. 4. However, several concerns limit the strength of this claim. The functional connectivity results in Fig. 4a do not seem to adequately support the claim that “prefrontal representations are integrated within the Precuneus.” The authors need to demonstrate that ISFC between the precuneus and each PFC region is significantly higher (1) specifically during update periods and (2) in a domain-specific manner. The current analysis appears limited to update periods. To support their claim, the authors should demonstrate that these effects are absent during non-update periods. Additionally, it is unclear whether these analyses support the domain specificity of PFC-Precuneus connectivity, as previously shown in Fig. 2. As the authors acknowledge, high PFC-Precuneus connectivity is expected and the ISFC between these regions might consistently exceed the other two integration models (within PFC and PFC-Hippocampus). To establish specificity, comparisons with functional integration of non-corresponding world models-PFC ROIs are needed (e.g., Agent update moments in amPFC-Precuneus vs. vmPFC-Precuneus, etc.).

Thank for your raising this point. We used connectivity to compare and contrast the within-PFC (owing to their anatomical adjacency) and PFC-HPC integration hypotheses with PFC-PCN connectivity. This is a valuable starting point but we agree that it is important to build on this result by testing that PFC-PCN interactions demonstrate domain-specificity.

We did not directly compare update period with non-update periods because there are many more non-update scans than update scans and this imbalance could impact the reliability of the estimated statistics (i.e., correlations might be higher for non-update scans because they are based on a much larger set of data and therefore less affected by noise). Instead, we have conducted the comparisons across domains that the Reviewer suggests, within the ISFC analyses reported in Figure 5d and in the main text. They confirm that during Agent updates, the amPFC-PCN correlation is higher than either vmPFC-PCN or dmPFC-PCN. During Action updates, the dmPFC-PCN correlation is higher than either vmPFC-PCN or amPFC-PCN. And during State updates, the vmPFC-PCN correlation is higher than either amPFC-PCN or dmPFC-PCN. Together, we believe these results convincingly demonstrate that PCN is most strongly correlated with different PFC subregions, depending on which world model is currently being updated.

4. While the analyses in Fig. 4b-d appear to partially address the concerns mentioned in 3, several questions remain. Given that ISFC analysis reflects population dynamics synchronization across participants, how does the similarity in synchronization profiles between the precuneus and PFC (Fig. 4d) support the claim that “the Precuneus selectively integrates updated prefrontal representations”? The authors should clarify how the results of synchronization of population dynamics relate to involvement of particular cognitive processes.

We have clarified the logic behind this analysis in the text: “ISPC indexes the degree to which the pattern of activation across the voxels in a region is similar across participants (Fig 5b). By computing ISPC in the Precuneus for each TR, we can construct a time-course of shared patterns across participants. High ISPC values indicate points during the movie when (stimulus-driven) neural representations are similar across participants. This similarity suggests that regions are actively engaged in stimulus-driven processing, such as updating beliefs about movie content. To determine whether the Precuneus is involved in global integration across domains, we compared the ISPC values in this region with the ISPC values in PFC regions. If the Precuneus performs a global integration role, then we would expect its ISPC dynamics to resemble those seen in different PFC regions in a domain-specific manner (Fig 4c). Around State updates, we would expect the Precuneus ISPC timecourse to be most aligned to that of vmPFC, since these regions should both be engaged in processing State-related changes at these times. For Agent updates, it should be most similar to amPFC and for Action updates, to dmPFC. Thus, we tested the correlation between two regions' ISPC values during belief updates of each type.”

Additionally, I have concerns about the interpretation of negative ISPC values. In the top panel of Fig. 4d, all ISPC values are negative. While the authors interpreted this as “Action updates elicited higher dmPFC ISPC correlations with Precuneus ISPC than vmPFC and amPFC,” these negative dmPFC-precuneus correlations might instead indicate a lack of population dynamics synchronization during Action updates rather than meaningful integration.

We agree it is important to make this clear to readers and have added the following text:

“However, it is important to note that the PFC-Precuneus correlations were negative during Action updates (though least negative in dmPFC). The reason for this is unclear but it may indicate that the PFC processes triggered by Action updates are somewhat desynchronized from those in the Precuneus. Finally, we extended this ISPC analysis over a range of temporal windows surrounding the updates (Supplementary Fig 14) to assess whether these trends persist over other window lengths. Similar patterns were found for windows of up to 10TRs surrounding updates.”

5. These concerns extend to the results in Fig. 5. The ISFC results between the visual cortex and the precuneus alone seem insufficient to support the claim that “the precuneus integrates updated representations with sensory regions.” If the movie stimulus included audio, the absence of similar results in the auditory cortex requires explanation, especially given the authors’ claim that the precuneus integrates updated representations with “sensory” regions broadly. Addressing these points would strengthen their argument.

Thank you for this interesting point. Our hypothesis was that PCN would integrate the top-down information with the relevant sensory stimuli to build/maintain high level situation/event models on the fly. The Hitchcock movie participants watched was heavily visual biased: although there is some dialogue, most of the critical plot information is delivered non-verbally (changes in behaviour, appearance of objects/entities triggering context shifts, abrupt actions not seen before etc). Hence, we observed strong connectivity with visual regions.

Our hypothesis predicts that this visual connectivity should be absent or much lower when participants listen to an auditory story. In line with this, we now added a new analysis of the Story with whole-brain ISFC maps seeded on the Precuneus (Fig 8c). This shows very limited connectivity with visual cortex, in contrast to the movie stimulus (now Fig 6).

6. Are there differences in how sISC (Fig. 6) and ISPC analyses should be interpreted? The authors employed multiple intersubject analysis methods, but their rationale for using specific analysis methods is unclear. For instance, could the sISC results in Fig. 6c-d have been obtained using ISPC analysis? If different methods were necessary, the authors should explain why each analytical approach was chosen for obtaining each specific result.

Thank you for this. There are indeed methodological constraints that required us to use different methods for these two analyses. We chose to use ISPC to investigate coupling between precuneus and PFC regions, which places the strong constraint that patterns of activity must be similar across participants, not simply levels of activity. Because this method computes spatial similarity over multiple voxels, it allowed us to compute inter-subject similarity for each individual scan in the movie independently (i.e., with high temporal resolution). This was important for the analyses in Fig 5, which involved computing temporal ISPC correlations over relatively short periods of time.

In contrast, the analyses in Fig 7c-d (previously Fig 6) compute correlations over the entire course of the movie. Here, the sISC approach is more appropriate as it constructs a similarity time-course by computing temporal inter-subject similarity over short moving windows. This results in a smoother and less temporally precise measure of similarity, which is acceptable when conducting analyses across the movie as a whole.

7. To replicate the findings in Fig. 2 and Fig. 4, the authors conducted the same analysis using a spoken narratives dataset (Fig. 7), stating that “this replication addresses potential limitations of using a movie stimulus and provides for an independent validation of some of the main results.” While this is an important step, I have concerns about whether the replication is successful. For example, the results for State updates did not show significant correlations between the vmPFC and precuneus, limiting meaningful outcomes of this analysis to Agent updates. Moreover, while the authors attribute the absence of significant differences for Action updates to “fewer characters in this story (two, one of whom narrates),” this explanation should equally affect Agent updates. This raises questions about the robustness of Agent update results.

Thank you for this. In response to Reviewers’ comments, in this revision we have performed extra analyses of the replication dataset. In the process of doing these, we identified an error in our analysis pipeline.

As stated in the original Narrative dataset paper, “The stimulus file was started after 3 TRs (4.5 seconds) as indicated in the events.tsv files accompanying each scan.” Our original analyses of these data did not take the 3 TR delay into account, hence the ratings were misaligned with the fMRI data by 4.5s. We have now correctly shifted the ratings accordingly (by 3 TRs) and the results (Figure 8) are more consistent with the movie data.

Comparison of previous GLM (top) and new (below) in (left to right Actions Agents States)

The GLM (Figure 8a) shows stronger effects in vmPFC, amPFC and dmPFC for States, Agents and Actions respectively. The ISPC analysis (Fig 8b) now shows successful replication in of the effects for Action domain update with dmPFC-PCN correlations significantly higher amPFC/vmPFC.

We have also conducted a seed-based ISFC (PCN seed) on the story. The underlying logic was as discussed in comment 5 above. In the movie, PCN and Visual cortex was strongly coupled as shown in Fig 6a (new version). In an audio-only story, there should be weaker correlations with visual cortex here compared to the movie. We found this to be true across all domains (Fig 8c new version), which supports our that PCN integrates top-down predictions with the necessary external input.

8. The paper suggests directional relationships (e.g., PFC → precuneus → sensory cortex). However, an alternative interpretation could be that PFC regions selectively utilize integrated representations from the precuneus. Given that the analysis cannot determine directionality, a discussion that cautions readers against overinterpreting the findings regarding directionality would be important to avoid misinterpretation.

We agree that our methods cannot infer directionality in information flow. We have now added this in our limitation in the results itself.

“These results cannot confirm whether there is a unidirectional or bi-directional flow of information between these two zones. However, they do highlight a network-centric representation of the unified internal model.”

Minor points:

1. While this study focused on three distinct domains—state, agent, and action—primarily in social contexts, other important domains, such as time, objects, and relations, have been established in previous research (Bottini & Doeller, 2020; Ekstrom & Ranganath, 2018). Including a discussion of these domains would provide a more comprehensive acknowledgment of the diversity of internal models.

Agreed. These are interesting topics and we have now contextualized our framework with these domains and properties in this Discussion paragraph (red text was from original version with **bold** as the new addition)

A rich literature of cognitive^{12,13,18,47,50,51,53,65} and clinical studies^{40,41,66} supports the role of DMN in higher-order human cognition. Although classically seen as task-negative, this network is implicated in a variety of cognitive activities associated with subjectivity, such as mind-wandering, creative thought, self-related processing and mental time travel. These tasks are inferential in nature and possibly involve construction of rich internal models of experience. **Situating the DMN within the literature, one finds numerous links to episodic memory, which extends beyond the hippocampal spatial machinery from which it evolved, to encode more abstract temporal and referential properties⁶⁷. Any experience can potentially be represented as a structured set of states, forming a cognitive map⁶⁸ with distinct transitions between them. Re-experiencing an event involves traversing this state space, akin to path integration in the physical domains, reinstating both states and their transitions. These transitions might follow meaningful directional trajectories, shaped by multiple abstract factors such as time, and agent-specific information. The dmPFC and amPFC may underlie such roles during episodic encoding, sensitive to changes in predicted trajectories.** Indeed, episodic memories of experience are thought to be compartmentalized through event segmentations, which are functionally driven by prediction errors^{44,69,70}.

Situation models often require encoding object-space relational properties as schemas—generalized templates that facilitate rapid learning of similar contexts and prediction of novel contexts, a function linked to the vmPFC⁷¹. Reference frames are critical in social/self-processing⁷² and conceptual integration across viewpoints⁷³. Some knowledge is anchored to an external, world-centered (allocentric) reference frame, while others are encoded relative to the self (egocentric) or others. The ability to condense conceptual knowledge across various frames into task-centric reference frames may be a key function of the amPFC. Distinct cognitive maps emerge from different experiences, prompting inference across them in extended contexts. The dmPFC might then aid in traversing these different cognitive maps as observed in exploration³⁵, and in this context mind-wandering⁷⁴. Finally, during naturalistic experiences, the DMN often co-activates with not just the medial temporal lobe structures, but also the anterior temporal lobe, which represents various forms of semantic and conceptual content⁷⁵. It's an open question whether the distinct predictions that emerge in the PFC depend on its interaction with specialized representations of conceptual knowledge in this region.

2. I found it challenging to fully understand the statement on line 535: “we correlated the average ISFC between the Precuneus and each PFC region during updates with whole-movie Precuneus ISC.” To help readers better understand the significance of this analysis the authors should explain the

implications of correlating the average ISFC between the Precuneus and each PFC region during updates and the whole-movie Precuneus ISC.

We apologize for the lack of exposition here, and have now changed it

“As a neural measure of the overall degree to which Precuneus was engaged in integration with PFC, we averaged its ISFC profiles with all 3 PFC regions for each participant. We hypothesized that individuals with similar integration of prefrontal predictions in the Precuneus would share more similar experiences than those with different integration profiles. To test this, we correlated the average ISFC between the Precuneus and PFC during updates with whole-movie Precuneus ISC (Fig 7d), which showed a significant correlation ($r = 0.54$, $p = 1.4e-09$).”

We also changed the result title to a more focused one for clarity

“Update-driven coupling with PFC relates to shared neural responses in the Precuneus”

We have also added the correlation of the whole movie PCN ISC with each of the PFC-PCN connectivity separately in the supplementary, while adding it in the main text

“We also correlated the whole-movie Precuneus ISC with each of the PFC region (during its domain-update) separately (Supplementary Fig 21), which showed significant correlations in each (though lesser for dmPFC/amPFC than vmPFC).”

3. Given existing research on cognitive maps (Behrens et al., 2018; Tang et al., 2023), the connection between the prefrontal cortex and the hippocampus would be expected to play a significant role in world model updating. However, this study highlights the importance of the precuneus-prefrontal cortex connection. A discussion on why similar results were not observed in the hippocampus would help provide further context and address this apparent deviation from prior literature.

This is an excellent point! Given the legacy of Hippocampus in the original cognitive map hypothesis, we've added the following under the ISFC results

“The above results suggest prefrontal representations are preferentially coupled with the Precuneus in all the updates. Interestingly, the Hippocampus also showed more coupling with the vmPFC (States), suggesting greater Hippocampal involvement in more contextual, State-related interaction. Strong PFC-Precuneus coupling is perhaps not surprising given the usually high connectivity between the PFC and Precuneus within DMN.”

4. In Fig. 3b, details of the statistical testing methods should be specified.

Thank you for this. We have now added this (we have performed phase randomizing than simple shuffling, a point raised by Reviewer 3)

“Statistical significance was assessed via permutation test on phase randomized neural time-series”

5. In Fig. 2a and Fig. 7a, color scales with threshold values should be added.

We have now added this in Fig 3a and Fig 8a now using the same color bar scales for easier comparison.

6. In Fig. 3, clarification is needed regarding whether the analysis accounts for HRF delay.

We did not convolve the binarized ratings with HRF, and have mentioned it in the methods.

“We did not convolve the binary ratings with HRF.”

7. References 10 and 18 need correction.

Thank you for finding this! We have now corrected this in the main text

10. Tenenbaum, J. B., Kemp, C., Griffiths, T. L. & Goodman, N. D. How to Grow a Mind: Statistics, Structure, and Abstraction. *Science* **331**, 1279–1285 (2011).

20. Takahashi, Y. K. *et al.* Expectancy-related changes in firing of dopamine neurons depend on orbitofrontal cortex. *Nat. Neurosci.* **14**, 1590–1597 (2011).

References

Behrens, T. E., Muller, T. H., Whittington, J. C., Mark, S., Baram, A. B., Stachenfeld, K. L., & Kurth-Nelson, Z. (2018). What is a cognitive map? Organizing knowledge for flexible behavior. *Neuron*, 100(2), 490-509.

Bottini, R., & Doeller, C. F. (2020). Knowledge across reference frames: Cognitive maps and image spaces. *Trends in Cognitive Sciences*, 24(8), 606-619.

Ekstrom, A. D., & Ranganath, C. (2018). Space, time, and episodic memory: The hippocampus is all over the cognitive map. *Hippocampus*, 28(9), 680-687.

Tang, W., Shin, J. D., & Jadhav, S. P. (2023). Geometric transformation of cognitive maps for generalization across hippocampal-prefrontal circuits. *Cell reports*, 42(3).

Reviewer #2 (Remarks to the Author):

In this work, Yazin and colleagues investigate whether the human brain processes internal models' updates in modular, domain-specific fashion or by means of a global integrated representation of events unfolding over time. Quite innovatively, they explored internal model representation using openly available fMRI data collected during visual (movie) and auditory (spoken narratives) naturalistic stimulation along with behavioral ratings of beliefs updates as predictors of neural activity. Through a robust analytical pipeline, the authors showed that three subregions in the human prefrontal cortex (vmPFC, amPFC, dmPFC) exhibit a domain-specific tuning for modelling believes about contexts/situations, characters beliefs/ goals and actions, while the precuneus, a core hub of the DMN, integrates such fragmented representations about the world into a coherent and integrated subjective experience. The rationale of the study is clearly articulated, the hypotheses

and the aims of the work are characterized in detail. The analytical approach is appropriate to address the main questions and the implications of the major findings are interesting and may potentially have a large impact on the literature fostering further investigations in the field. I believe that the following point needs to be addressed to further improve the article:

Thank you for the meticulous review and kind words.

1. Although the authors appropriately describes the experimental pipeline in the Methods section, I believe that adding an initial figure composed by different panels to illustrate i) the stimuli (movie and narrative), ii) the sensory modalities involved (multimodal audiovisual vs unimodal audio-only) along with iii) the experimental groups (e.g., 4 distinct groups for each stimulus presentation condition, number of participants, mean age) and iv) the tasks administered (i.e., fMRI, behavioral) would enhance the clarity of the analytical procedures, showcase the richness of the dataset and enable the reader to grasp important information at a glance.

Thank you for this suggestion. We have created a pipeline with all the important detail about the data including modality, duration, groups involved for the tasks as well as a high-level depiction of the analysis rationale/methodology. We believe this should serve as an aid for the reader in navigating the big picture easily. This is now Fig 2 in the new version

2. Figure 2 and figure 7 show the brain maps of contrasts for each domain against the other two for movie watching and narrative listening respectively. In the panels a, cortical and subcortical regions are displayed. Since the analysis focuses on midline prefrontal areas (i.e., vmPFC, amPFC, dmPFC), I believe that showing the results of the contrast within a cortical mask -so to get rid of subcortical activations, which to my understanding are not the main topic of the work- or within the PFC mask solely (made by the conjunction of the regions of interest) would make these results more readable, clean and elegant.

Thank you for this point. You are right that the central focus is on the midline PFC. We chose to show the full maps so as to give readers the general picture of the brain in all three contrasts. Comparing the movie and story (please see new Story replication in General section), the reader can infer that the consistent regions across the datasets are the PFC (and Posteromedial zones). All our subsequent results focus on the ROIs of these sectors to further show the specificity. Although our focus is on specific PFC regions, we are also aware that other researchers may be interested in other regions (including subcortical regions).

3. In the text the authors report: "Group-level ratings were only weakly correlated with one another (States vs Agents, $r = 0.31$, Agents vs Actions, $r = 0.29$, Actions vs States, $r = 0.28$). This indicates that models in different domains were being updated at different times during movie-watching". However, visual scrutiny of Figure 1c (in which three smoothed update probability time-courses are represented) shows that peaks of at least two distinct domains often coincide (for instance approximately at 48, 50, 80, 100, 130, 160 TR) or are just slightly delayed. In my opinion this observation indicates that, to some extent, the occurrence of events pertaining to distinct domains shows a non-negligible overlap over time. To this regard, although correlation coefficient plays a

pivotal role in quantifying linear relationships between variables, its application to time series data is very challenging due to possible temporal dependencies. A viable alternative could be to compute the cross-correlation since it can effectively show how changes in one time series relate to changes in another taking into consideration slight differences in time lags. Other metrics such as mutual information could be used as well to better understand how much information is shared between timeseries beyond correlation. Additionally, given the intricate relatedness of features in naturalistic stimulation, it could be worthy to evaluate whether and to what extent the three domains share common underlying information which may cause them to be collinear to a certain degree. For instance, changes in the context, actions and agent believes, may partially overlap with scene transitions, which mark pivot, break points in the unit of storytelling. Asking a few new raters to tag scene changes in the movie clip, would be sufficient to test whether these events are collinear with the changes in belief the authors are studying in the present work.

This is an excellent point and we have now performed the cross-correlation analyses on the three ratings in movie and story. We have added this report in Supplementary Fig 3. We found that correlations with various time lags were always smaller than the basic unlagged correlations. Thus, the correlations we report in the main text represent the upper bound on collinearity. We also conducted the mutual information Supplementary Fig 2 as it is a distinct metric of shared information, and it returned low values.

It is also important to note that the GLM results reported in Figs 3a and 8a included all three ratings as simultaneous predictors to account for collinearity.

Overlap between update periods is another important concern, which we attempted to minimise through the choice of threshold we used to binarise the ratings data and identify updates. As shown in Supplementary Figures 4 and 5, the thresholds we used result in relatively low levels of overlap in updates of different types (less than 25%).

4. To assess inter-rater reliability the authors performed a split-half correlation procedure. The values reported in the manuscript indicates that the tags assigned across the two subgroups of raters are fairly consistent for the States domain ($r = 0.80$), while Agents and Actions show fairly low values ($r = 0.49$, $r = 0.57$ respectively). I was wondering whether this could originate from the difficulty raters experienced in identifying changes in beliefs in the tale and in labelling them into separate, distinct domains. For instance, reading the instructions the authors provide in the Supplementary Material (table 1), I framed the domain "States" as indicating an unpredictable event, changing my expectation or "cognitive script" of a typical situation (such as in the restaurant example provided). However, I had troubles in distinguishing it neatly from the "Actions" domain given the fact that changes in the context likely trigger unpredictable switches in the following actions performed by the characters. Did you perhaps train the subjects on a different movie clip and collect their ratings before the actual experiment? This could provide an interesting index of tagging accuracy to evaluate the "goodness" of subjective ratings.

Thank you for this point. We did not train the subjects on any clips before because we wanted the experience to be akin to the scanning participants in terms of the movie watching. The demographic (age, country, education, gender info) was matched to that of the scanning participants.

We used the exact same instructions for the movie and the narrative (except to watch/listen), and the same method to average and smooth the data to regress on the BOLD data. We believe the consistent and highly significant results in the GLM and ISPC across datasets (please see the new story replication results with technical changes in Summary section) suggest that participants were interpreting the instructions precluding any instructional comprehension difficulties.

We do agree Actions (in the way we define the domain) might be difficult to map on a movie/story since there is no active strategies deployed by the participants. On an ideal setting, as an example, if one was going to a movie and found out during the commute that there are no more tickets, one can redeploy a different 'way' to obtain entertainment (another movie, park, meetup etc). Movement through various different kinds of state spaces/maps is what abstract/temporal actions here signify. In the context of a movie/story this roughly means the shift a viewer experiences when events that induce huge changes to the expected plot, thus requiring them to 'move' and entertain other state space configurations that satisfies the data. E.g. seeing a kid use a loaded gun for the first time would invariably change one's expectation of this movie from being funny to more suspenseful as more negative states have now opened up as possibilities. Situating this in the literature, Actions in decision-making can be seen as explorations over various state spaces to find the right models (in line with Kolling et al). In more naturalistic-settings it can also be seen heavily involved in mind-wandering, where one traverse wide swathes of seemingly different cognitive maps. This was strikingly specific in Fig 2 of Gordon et al. We've now added this in the discussion to improve clarity

"The dmPFC might then aid in traversing these different cognitive maps as observed in exploration³⁵ and in this context mind-wandering⁷⁵."

Kolling, N. & O'Reilly, J. X. State-change decisions and dorsomedial prefrontal cortex: the importance of time. *Curr Opin Behav Sci* **22**, 152–160 (2018).

Christoff, K., Gordon, A. M., Smallwood, J., Smith, R. & Schooler, J. W. Experience sampling during fMRI reveals default network and executive system contributions to mind wandering. *Proceedings of the National Academy of Sciences* **106**, 8719–8724 (2009).

Going on with this, even if the split-half correlation is a common and robust method, it would be interesting to combine it with other metrics to gain a comprehensive understanding of the consistency both within the tagging system and among raters. Moreover, to account for the fact that half a test (i.e., half the raters pool) is evaluated, I think the author might want to consider to adopt the Spearman-Brown formula to adjust the split-half correlation upward to estimate the reliability of the full test.

We have now adjusted the split-half correlation values with the Spearman-Brown correction and it has increased reliability scores on both movies (Fig 1d) and story (Supplementary Fig 24b).

5. At page 19, the authors adopted a dynamic intersubject correlation analysis (sISC) and tested whether the ISC in the precuneus computed over a sliding window, correlates with the emotional Arousal ratings. I suggest the authors to include more details about this analysis such as for example

the window width and the time step they used and to motivate better why correlation with arousal means that the region is tracking a unified experience.

Thank you for pointing this out. A previous work (Majumdar et al) found high correlation between whole brain sISC and behavioural engagement (arousal), motivating us to use this approach. We used the same window size as this previous work. We have now included more details on this analysis (sISC and Arousal) in the methods, and referenced related works

“The dynamic variant of ISC, (spatial ISC/sISC) was computed as a leave-1-out ISC using a sliding window⁴⁸ of 21TR with 1TR overlap⁴⁷ by correlating the BOLD signal of each subject with the mean BOLD signal of the rest, for each window and for each region of interest (ROI). The Pearson correlation coefficients were averaged for each window, Fisher-z-transformed, and averaged across subjects, resulting in a single time course of mean ISC across the whole movie, for a region. We compared both the group-level as well as the individual-level ISC time series with the Arousal ratings (Fig 7b, c). Whole-brain sISC had strong positive correlation with arousal in a previous study⁴⁷”

47. Majumdar, G., Yazin, F., Banerjee, A. & Roy, D. Emotion dynamics as hierarchical Bayesian inference in time. *Cerebral Cortex* 33, 3750–3772 (2023).

Another approach I think is suited to talk about “unified vs fragmented” experience, is to create window sizes of increasing width, ranging from the actual TR (2.470 seconds, which leads the same results of the classical ISC pipeline) up to, let’s say, 1 minute. This technique has been widely exploited in the field of naturalistic stimulation (see Hasson et al., 2008) with the name of Temporal Receptive Windows (TRW) analysis. Using this approach, a high ISC over a short window (that is, few seconds) would suggest that the correlation was modulated by rapidly changing events, whereas high ISC values over longer segments (that is, tens of seconds) will indicate that the correlation mostly relied on accumulating information (that is the “unified experience”). Therefore, following the authors’ hypothesis, the precuneus should display higher ISC values over long windows, while the three prefrontal regions should show greater ISC values over short time windows.

We have done the ISC for different domain updates over windows ranging from 5s (2.47s in before and after the update point) to ~60s (or 24 TRs) for PFCs and PCN. These results are shown below. In general, ISC values are largely stable across window sizes, suggesting the choice of window is not a major factor in driving these results. The exception is perhaps the ISC around Action updates where, as the Reviewer predicts, Precuneus ISC does seem to be higher for longer windows.

We have now added this in the Supplementary Figure 22 for the interested reader

6. As new events and experiences are encountered, the human brain possesses the remarkable capacity to adapt and transform mental representations of people, events, and environments to align with the dynamically changing world. Not only can these representations be modified, but they can also be assigned to a mental timeline, creating a sense of temporal coherence. Further, the presence of new information can prompt individuals to revise and update existing narrative representations, allowing for a flexible and adaptive understanding of personal stories. The change of beliefs in all the three domains (i.e., scene, agent and actions) do coincide with crucial points in the narrative, which can be conceived as “plot twist”. These moments in the story capture the

viewer/listener alertness by defying expectations and trigger, as the authors discuss, both emotional engagement and cognitive reappraisal. Therefore, evolving cognitive demands would be required to process these “points of rupture” in the narrative in order to make sense of the plot evolving through time. For instance, goal-directed (attributed to the influx of plot-relevant information) and stimulus-driven visuo-spatial attention, semantic (influx of new plot-relevant information, requiring a solid semantic structure to build forthcoming narrative elements) and episodic (increase in information being integrated and updated into long-term memory as the narrative progresses) memory would sustain the formation of an integrated representation. Altogether, distinct cognitive processes inherent to the narrative would give rise to dynamic alterations in neural synchrony within the attentional, semantic, and episodic networks. How would the authors expand the discussion about the results of the present work in light of these considerations? I believe that adding a paragraph in the discussion section about these aspects would enrich the manuscript and prompt further debate.

Thank you for the rich comment! We added on how these prefrontal operations (and more generally the DMN) underpins episodic memory updates and interplays with semantic stores in the following paragraph in discussion.

A rich literature of cognitive^{12,13,18,47,50,51,53,65} and clinical studies^{40,41,66} supports the role of DMN in higher-order human cognition. Although classically seen as task-negative, this network is implicated in a variety of cognitive activities associated with subjectivity, such as mind-wandering, creative thought, self-related processing and mental time travel. These tasks are inferential in nature and possibly involve construction of rich internal models of experience. **Situating the DMN within the literature, one finds numerous links to episodic memory, which extends beyond the hippocampal spatial machinery from which it evolved, to encode more abstract temporal and referential properties⁶⁷. Any experience can potentially be represented as a structured set of states, forming a cognitive map⁶⁸ with distinct transitions between them. Re-experiencing an event involves traversing this state space, akin to path integration in the physical domains, reinstating both states and their transitions. These transitions might follow meaningful directional trajectories, shaped by multiple abstract factors such as time, and agent-specific information. The dmPFC and amPFC may underlie such roles during episodic encoding, sensitive to changes in predicted trajectories.** Indeed, episodic memories of experience are thought to be compartmentalized through event segmentations, which are functionally driven by prediction errors^{44,69,70}.

Situation models often require encoding object-space relational properties as schemas—generalized templates that facilitate rapid learning of similar contexts and prediction of novel contexts, a function linked to the vmPFC⁷¹. Reference frames are critical in social/self-processing⁷² and conceptual integration across viewpoints⁷³. Some knowledge is anchored to an external, world-centered (allocentric) reference frame, while others are encoded relative to the self (egocentric) or others. The ability to condense conceptual knowledge across various frames into task-centric reference frames may be a key function of the amPFC. Distinct cognitive maps emerge from different experiences, prompting inference across them in extended contexts. The dmPFC might then aid in traversing these different cognitive maps as observed in exploration³⁵, and in this context mind-wandering⁷⁴. Finally, during naturalistic experiences, the DMN often co-activates with not just the medial temporal lobe structures, but also the anterior temporal lobe, which represents various forms of semantic and conceptual content⁷⁵. It's an open question whether the distinct predictions that emerge in the PFC depend on its interaction with specialized representations of conceptual knowledge in this region.

Minor comments

1. Supplementary Figures 7-12 need a proper caption which briefly comments the results presented.

We have now added captions to all these

2. Try to fix where possible the variations in terminology adopted. In my opinion, having a consistent lexicon through the text helps the reader and makes the final message clearer. For example, I suggest to change “State”– “Agent”- “Action” Updates (e.g., lines 406-411) with the plural form “States – Agents – Actions” as stated in the figures and other parts of the text. The same holds for “Supplementary Figure” (line 203) or “Suppl Fig” (e.g., lines 512, 977).

We have now made the plural form wherever applicable and Supplementary throughout, including the figures

3. Figure 5: since the brain mesh is a pale, light-grey color, I would recommend to use a different colormap to represent the results of the ISFC analysis. A color palette going from red to black or from blue to green excluding the white component (for instance the matplotlib palettes viridis, plasma, inferno, magma or the sequential Color Brewer palettes) would enhance the visual impact and the readability of the connectivity maps.

Thanks for this suggestion. We have tried a range of color schemes, including as suggested as shown below.

We feel although the blue-green variant is attractive, this color scheme might confuse as this is similar to deactivation profiles in the GLMs elsewhere. The red to black is visually most striking, but black might be unusual for some readers to map interpretations onto. For this reason, we chose the blue turquoise white color scheme.

4. Some figures present the ISC values for the three PFC regions by means of bar plots. I believe that the order of regions on the x axis of the bar plots in Figure 2- Panel b, Figure 4 - Panel d, and Figure 7 – Panel b, should be the same to aid visual comparability. This will be extremely useful in particular for Figure 2 and 7 where the authors show the generalizability of results across content and sensory modalities.

This is a good point and we have now changed Figure 2 panel b from barplots to density plots in panel b, Fig 3 (as per a suggestion from Reviewer 3). This is exactly the same as panel c making it easier for the reader now.

Reviewer #3 (Remarks to the Author):

Using a publicly available audio-visual movie fMRI dataset, the authors show that three midline frontal brain regions respond selectively to different types of narrative changes, suggesting involvement in the update of state, agent, and action components of situation models. It is further shown that these frontal regions are more functionally connected with the precuneus than with the hippocampus or with each other, and an innovative analysis relates this to correlated consistency of activation patterns. Precuneus activity transitions are also correlated with subjective arousal ratings. A subset of these findings are replicated in the second publicly available audio-only dataset, showing consistent results for states and agents, but not for actions.

The paper is generally well written and presented. The topic and results are interesting. The methods and statistics are sophisticated and generally appropriate (but see specific queries). Overall, I enjoyed reading the manuscript and I feel it will make a valuable contribution to the literature. However, I have several concerns and queries as detailed below.

Thank you for the in-depth review and kind words.

1. I wonder whether the claim “it is unknown whether humans acquired an assembly of many separable and highly specialized models” (e.g. line 45-46) might be overstated. There is little doubt that the brain is somewhat modular with domain-specific specializations. In particular, in the context of narrative understanding (both reading and movie-watching), it has long been known that different brain regions track different aspects of a narrative, including locations, characters, and characters’ goals, e.g. Speer et al., 2009, “Reading Stories Activates Neural Representations of Visual and Motor Experiences”, and Zacks et al., 2010, “The brain’s cutting-room floor: segmentation of narrative cinema”. The proposal in these studies is also that prediction failures based on various feature changes trigger updates of situation models, and the brain response at these changes mediates subjective ratings of event boundaries. It has also been proposed that midline parietal DMN has a particular role in integrating event models and incoming sensory information, through coupling with midline frontal regions (e.g. Stawarczyk et. al., 2021, “Event Representations and Predictive Processing: The Role of the Midline Default Network Core”). That said, the current findings of differentiation within the medial frontal lobe, the particular framework of states, agents and actions, and functional connectivity with the precuneus, provide a novel and useful addition to this literature.

This is a fair point and we have rephrased the sentence to emphasize the tripartite differentiation in the MPFC more in the main text, along with some of these references

“Yet it is unknown the extent to which humans use an assembly of many separable models each specialized for different kinds of prediction^{2,16,17}, versus a single unified model.”

2. Throughout most of the manuscript, key domains of world knowledge are defined as “states”, “agents” and “actions”. It is confusing that the abstract does not use these terms but instead refers to “spatial,” “referential,” and “temporal” domains. Some of these seem more specific (“spatial” vs “states”); others more general (“referential” vs “agents”). “Spatial” is especially confusing, because the instruction given to participants (Supplementary Table 1) uses an example of a character change within a fixed spatial location. I think it would be more consistent to use “states”, “agents” and “actions” in the abstract, and if using alternative terms then clarify their relation in the introduction. (This issue also arises on line 613: I disagree that states are necessarily spatial, and I’m not persuaded that actions are necessarily “temporal”.)

Thank you for this point. We originally had abstract spaces in mind when referring to the state models but we can now see why this can be confusing. We have now removed ‘spatial’ as a way of referring to them generally, and replaced them with ‘contextual’. This also is more in line with the instructions imparted to the participants.

We have now edited this in the main text and Abstract.

“Representing these **contextual** State models seems necessary but insufficient for a full world model.”

We have also amended the abstract in the main text and connect it with the MPFCs adding more clarity from the start

“Using fMRI during naturalistic experiences (movie watching and narrative listening), we show that three topographically distinct midline prefrontal cortical regions perform distinct predictive operations. The ventromedial PFC updates contextual predictions (States), the anteromedial PFC governs reference frame shifts for social predictions (Agents), and the dorsomedial PFC predicts transitions across the abstract state spaces (Actions).”

3. Although the theoretical focus is on medial prefrontal cortex, in none of the figures does this brain region seem particularly unique. E.g. in figs 2, 5, & 7 responses that are at least as strong tend to be seen across many other brain regions. I do like the story of a tripartite division within mPFC, but I wonder whether this ignores a bigger picture.

Thank you for this observation. It is true that PFCs (and PCN) are not the only zones implicated, especially with the sensory and multisensory regions showing strong effects. However, comparing Figs 3a and 8a, it is clear that the consistent effects across movie and story analyses are largely confined to mPFC and PCN. Effects in other regions (e.g., the occipital effects for Actions observed only in the movie analysis) most likely represent modality-specific processes that depend on the

nature of sensory input. This hypothesis is borne out by additional ISFC analyses requested by another Reviewer. Strong connectivity between PCN and occipital cortex was present in the movie data but not in the auditory narrative data (compare Fig 6a with Fig 8c).

4. Some methodical choices are unclear, especially when these vary across analyses. E.g. why was a 5 s window chosen for combining update ratings, a 7 TR window for the ISC analyses, and a 9 TR window for the ISPC analyses?

These are fair and extremely important technical questions which prompted us to investigate the results over a range of windows for these analyses.

We used 5s windows centred on update points, mainly with the movie in mind (1 TR = 2.47s/~2.5s). 5s is a commonly-used temporal window for segmenting naturalistic/narrative stimuli (Hoffman et al). Too small a window (say 1s) and it might not have any effect, and too long and it would incorporate unnecessary information (Fig 1b for an illustration).

We chose 7 TR for ISC. In order to investigate whether the patterns observed is dependent on this choice, we have now run the analysis on a range of plausible windows (please see Summary section, Window analysis for these figures & Supplementary figures 10). We found the ISC domain-specific pattern was consistent across a range of windows (below).

We used 9 TR for ISPC since pattern-level integration might persist for a bit longer than simple BOLD-level coupling. The same trends of similar integration profiles with PCN were also observed here (Supplementary figures 14, 23).

The story also used the same TRs for ISFC and ISPC (Fig 8b, 8c, in the new version) as in the movie for fair comparison. We also did the window analysis on the story ISPC, and found the same consistent pattern (also please see technical changes to the story replication in Summary Section, Story Replication).

1. Hoffman, P. Reductions in prefrontal activation predict off-topic utterances during speech production. *Nat Commun* 10, 515 (2019).

Why use frequentist statistics for the activation analyses and Bayesian statistics for the ISC analyses?

We have now used Bayesian hierarchical regression on the movie GLM betas (c.f. Frequentist statistics as before), and put this in the main figure (Fig 3b in the new version).

Why use FDR correction for results in Fig 2 but not results in Fig 5?

This is a good point. We have now applied a threshold of FDR-corrected $p < 0.001$ to these results. The results are largely unchanged (see new Fig 6).

5. For several of the statistical tests, I am concerned about potentially inflated significance due to non-independence of the samples. Whenever significance is based on correlations across time, autocorrelation can bias significance if not accounted for. I think this applies to the HMM analyses and ISPC analyses, where random shuffling of time-points would be invalid if it does not preserve temporal autocorrelation. Circular shifts and phase shuffling could be permutation approaches that better preserve autocorrelation.

We agree random shuffles is not the optimum approach so we have now changed that to phase randomization as recommended in reference 41 (in the original manuscript). For both HMM and ISPC analyses, the neural data is now phase randomized and submitted to permutation as before. The critical results are unchanged (see Fig 4, 5).

A related but different concern arises for the Bayesian regressions of ISC and ISFC, and seed-based ISFC: although here the significance is based on variation across subjects, the values per subject are not fully independent because the leave-one out approach means that there is overlap across folds (as noted in reference 41).

The authors of ref 41 (Nastase et al) state that in the case of one-sample tests (i.e., testing whether a mean inter-subject correlation is significantly greater than zero), a simple parametric t-test is invalid due to non-independence in the data of different subjects, particularly when pairwise ISCs are used. This is not a major concern for our analyses for two reasons. First, we use the leave-one-out method to compute ISCs rather than the pairwise method, which Nastase et al. state is less affected by the non-independence issue. Second, and more importantly, we do not perform any one-sample tests – all of our Bayesian regressions employ two-sample tests for differences in ISC values between two experimental conditions/regions (e.g dmPFC v vmPFC in ISC State updates/ PCN-PFC v within-PFC integration during Agent updates etc). Nastase et al. states that it is acceptable to use parametric tests in the two-sample case, since non-independence affects both conditions equally.

6. I do not agree with the statement on line 221-222 that the activation effects seen in the GLM do “not confirm that these effects are truly driven by the movie stimulus alone (rather than stimulus-unrelated internally driven thoughts).” Stimulus-unrelated activity would be temporally idiosyncratic and so average out in the GLM. Responses that are statistically significant in the cross-subject GLM must be consistent across subjects and therefore evoked by the stimulus (or something that is time-locked to it).

We have now removed this sentence.

7. Related to the above point, the GLM analysis, ISC analysis, and the HMM analysis do not seem independent, but rather three slightly different ways of probing related aspects of the same data. I.e. having shown that subjects’ activity time-courses on average correlate with behavioral ratings (in a domain specific way), it seems necessarily the case that subjects’ activity time-courses will correlate with each other (ISC) in a similar way, and that transitions extracted from them (HMM) will similarly correlate with the ratings. A similar point about non-independent analyses applies to those in Fig 6 panels a and b.

Thank you for this comment. As is common practice in fMRI research, we have subjected our data to a range of different analyses. Since these analyses operate on the same data, it's true that they're not strictly independent and the patterns they uncover are likely generated by the same underlying neural processes. However, we believe that the analyses do provide distinct ways of characterising and understanding the underlying neural processes.

The GLM used continuous and smoothed measures of updating in one single model, while the ISC/HMMs used discretized binary updates. The latter was mainly to probe neural dynamics *within a region* (conditional on an update) in case of HMM and *across the subjects* in case of IS analyses. This can be within region (ISC) or comparing similarity among regions (ISPC). One uses averaged BOLD while the other is pattern-level similarity expressed as correlation time series, which is then compared across various regions. In Fig 6 (old version), panel a showed HMM analyses that identify transitions in neural states, while panel b showed dynamic ISC analyses that use a sliding window approach to quantify shared synchronization. While the central data is about a movie experience, we believe these analyses show very different expressions of the neural data – continuous modulation, discrete dynamics, shared responses and inter-areal integration profiles.

Importantly, we applied the same GLM, ISPC analysis to an entirely independent dataset that uses distinct content in a different sensory modality (the narrative data in Fig 8). Since we found comparable patterns here, we are confident that our key results generalise across participants and content.

8. Regarding the HMM analyses, several aspects were unclear to me:

a. Is it surprising that only two (or three) latent states were selected per ROI (which seems to suggest a binary response per region)?

This does seem low but we did bootstrapping with selection of the median number of states over many runs of the analysis, to ensure that this result was robust. It is perhaps movie-related. Usual movie-HMM studies use longer, richer movies (>30m) with a variety of content, whereas ours was a short movie (8 mins) . We have also focused on *when* an update occurs rather than the contents of a specific HMM latent state, which is not in line with the scope of this work.

b. Is it a problem that the inter-run time-series correlation for dmPFC is close to zero (panel A of supplementary figure 5)?

We believe the dmPFC did exhibit more variability across participants in many ways. While it is difficult to ascribe an explanation to this post-hoc, it is possible that there is more variability in how people expect the movie to unfold than compared to beliefs about characters and stateful contexts, which are more or less 'controlled' by the director.

c. I would also query the description on line 290 of "exhaustively bootstrapping"... If I understood correctly, 30 random samples were taken, which is far from exhaustive, and samples were not taken with replacement, as would normally be the case for a bootstrap?

Thank you for this critique. Our focus was on maximizing diversity in subject exclusion across different iterations, meaning every excluded group is distinct and does not have duplicate subjects in a single iteration, hence sampled without replacement. Each iteration would remove 10 subjects without replacement, and this was performed 30 times. The lack of replacement would ensure any

one subject is not overrepresented, and thus the whole dataset ($n = 111$) is cycled ~thrice. To avoid any confusion, we have changed the reference to “exhaustive bootstrapping” to “repeatedly model-fitting across subsets of participants”

d. How can there be any belief updates within a time window of zero scans?

This was poor reporting on our part. We have now shown the correct HMM x axis in Fig 4, 7 and Supplementary Fig 13, starting from 1 TR window.

e. Line 306 says, “This suggests that the subjective experience of an update to predictions occurs sometime after the neural model is reconfigured”, however this could not really be otherwise when only considering time windows that follow rather than precede the neural transition? Relatedly, the preceding GLM implies, instead, that the neural effects occur sometime after the belief updates. Also, lines 875-7 state, “Where a sequence of multiple TRs/scans exceeded θ , we selected the final TR/scan in the sequence to represent the update.” This decision will bias the belief updates to be later in time, relative to the neural signal. Further, there will be some reaction time between when a belief update occurs and when a participant makes the button press, and this does not seem to have been measured or accounted for. Overall, any inferences about temporal precedence seem to be on shaky ground.

Several Reviewers commented on this. With hindsight, it’s clear we overstated this aspect of our results. In fact, we did not intend to make any strong claims about the relative timing of behavioural and neural effects, as we acknowledge that this problematic for the reasons the Reviewer mentions.

Accordingly, we have adjusted the wording in several places in the paper

We changed the wording in abstract from preceded to coincided with

“Prediction-error-driven neural transitions in these regions, indicative of model updates, coincided with subjective belief changes in a domain-specific manner. “

We changed the result title in Figure 4 (new version) similarly

“Neural Transitions Coincided with Subjectively Experienced Belief Updates”

We have now removed this sentence

“This suggests that the subjective experience of an update to predictions occurs sometime after the neural model is reconfigured.”

f. Being pedantic, the word “discrete” on lines 308 & 682 is not really justified because discreteness is an assumption of the HMM rather than something being tested.

Fair point and we have now removed these from the main text.

g. Finally, the claim on line 310 that the neural shifts “mediate” the belief shifts is not statistically tested; the presented results simply show a correlation.

Agreed and we have now changed it to **coincide with**

“Three separate internal models in the prefrontal cortex appear to **coincide with** these shifts in three key domains during unguided naturalistic experience.”

9. Line 376 states “...above results suggest prefrontal representations are integrated within the Precuneus.” However, essentially this is a functional connectivity analysis based on correlated activation, so doesn’t really say anything about “representations” or “integration”, just that the frontal regions are more correlated with the precuneus than with the hippocampus or with each other. The following section describes some clever ISPC analyses that do get closer to representational integration, so the phrasing here just needs to be a bit more careful.

Thanks for this and we have limited the interpretation to just (shared) coupling, which is, what it is really. We added this sentence as well as substituted integration with **connectivity/coupling/synchrony** for the ISFC interpretations from

“This allowed us to measure and compare coupling strengths of the updated prefrontal representations within the above three hypothesized ways.”

10. In Figure 4, how do you interpret the significantly negative ISPC correlations for action updates? (In panel d, error bars are not defined, and it would help to order the x axis to match panel b of Fig 2. Also, some error bars are missing from supplementary figures 7-11.)

We agree it is important to note the overall negative effects for Action updates and have added the following text:

“However, it is important to note that the PFC-Precuneus correlations were negative during Action updates (though least negative in dmPFC). The reason for this is unclear but it may indicate that the PFC processes triggered by Action updates are somewhat desynchronized from those in the Precuneus.”

Error bars are standard error of the mean, and we have defined it in all places in main and Supplementary figures now.

We’ve now changed the Fig 3b (new version) barplots to posterior densities which makes it easier to read the following ISC (panel c) now as well. All ordering of similar structure are now same for easier comprehension.

11. For the seed-based ISFC (Fig. 5), can you rule out the possibility that visual cortex was more correlated with precuneus than frontal seeds simply due to anatomical proximity?

This is a great question To rule this out, we have now implemented the same seed-based ISFC (seeded PCN), on the narrative data, which was delivered exclusively via audio. Indeed, we found a

conspicuous absence of visual connectivity with PCN in these data, ruling out simpler explanations of anatomical adjacency (Fig 8c).

12. I was very confused by the sections “Similar Predictions Accompany Similar Experience” and “Correlations between Subjective Experience and Predictions,” because the headings talk about “predictions” and “experience,” whereas the described analyses seem to involve neither of these things, instead showing that [precuneus with prefrontal connectivity] correlates with [precuneus inter-subject consistency]?

We agree it might be misleading and has changed it accordingly

We changed the title of results to

“Update-driven coupling with PFC relates to shared neural responses in the Precuneus”

We changed the caption of results in Figure 7 to

“Relation between PFC-PCN coupling during updates and shared neural responses in the Precuneus”

We changed the title of methods to

“Correlations between PFC-PCN coupling and shared responses in the Precuneus”

13. The results on arousal come rather out of the blue, having not been mentioned in the abstract or the introduction.

Good point. We have now mentioned this in the introduction to give prior to the reader.

“In other words, the evolving unified experience should be integrated and maintained here, while the fragmented modular predictions persist in the PFC.”

As well as inserting in the next paragraph, “We hypothesize that these predictions are integrated in the Precuneus, contributing to a unified representation of one’s current environment. This process may underlie the subjective experience of the world, which we operationalize in this study by measuring participants’ moment-to-moment arousal levels.”.

14. In what sense are activation profiles “mapped to a ventral-dorsal gradient” (line 614)? Does that mean that functional preferences are smoothly varying (a necessary consequence of smoothing during preprocessing)? Or that “agent” representations are intermediate between “action” and “state” representations along a continuum (what would that mean)?

These are really interesting questions but we don’t think our data can really answer them conclusively. Our findings do not imply a strictly continuous nor a discrete tripartite organization in the prefrontal cortex, but rather that state, agent, and action representations are *functionally and spatially dissociable*. It also appears robust across both movie-watching and narrative comprehension tasks, suggesting a more general organization.

To avoid potential misinterpretation, we have therefore rephrased as “Domain activation profiles mapped to distinct regions in the midline PFC”

15. The term “fragmentation” in the title suggests decomposition of an initially unified model. However, the manuscript itself generally suggests an opposite flow, with an integrated model being built in the precuneus from initially separate components. Some clarification of when models are seen as fragmenting vs integrating might be helpful. More generally, the paper suggests (implicitly and explicitly) causal relationships such that frontal regions influence the precuneus and that these regions influence sensory activity and conscious experience.

We acknowledge that “fragmentation” may imply decomposition, and we appreciate the opportunity to clarify its meaning in the context of our study.

We have clarified our use of this term in the Discussion, after the reader is familiarized with the results (**bold** for the new addition).

“Throughout this paper, we have referred to “fragmentation” of the model space. By fragmentation, we mean the within-network deconstruction and reconstruction of an internal model through separate functional updates. We have argued this is located in distinct PFC subregions. Thus, multiple prediction 'threads' have to be active and integrated with the relevant sensory channels for updating and exploiting high-level models, perhaps through top-down and bottom-up integration, potentially experienced egocentrically in the medial parietal regions. This transient, modular breakdown of the internal model followed by its reintegration can explain the resurfacing of this set of regions across superficially distinct but computationally similar tasks in different studies. Fragmentation of experience may be seen as a possible reason behind consequential clinical accounts like blind-sight, spatial neglect⁷⁸, dissociative consciousness disorders, and in extreme cases commissurotomy-related phenomena⁷⁹. Inability to integrate these prefrontal predictions can offer a fresh perspective in examining psychiatric conditions with independent (and often rebellious) 'conscious' entities within. Indeed 'misintegration' by Precuneus, the hub of DMN, are well reported in clinical studies underlying related phenomena^{40,41}. Finally, there is a computational formalism of seeing resting-state DMN activity as the prior models encoded in the cortex, under Bayesian frameworks^{4,80}. The present study is suggestive of the dynamic processes and structural constraints by which these priors are updated as an experience unfolds.”

However, since all analyses are correlational, it seems possible that these relationships might be reversed, be bidirectional, or even independent.

To avoid any directional claim, we have now added this on the results which touches this topic

“These results cannot confirm whether there is a unidirectional or bi-directional flow of information between these two zones. However, they do highlight a network-centric representation of the unified internal model.”

16. The abstract claims “Results generalized across sensory modalities... suggesting...abstract, modular predictive models for both vision and language.” Although this is likely, both datasets include audio/language, so it seems possible that language-based models could account for all findings, with minimal evidence of whether they might apply in a purely vision-based situation?

The Hitchcock movie participants watched was heavily visual biased: although there is some dialogue, most of the critical plot information is delivered non-verbally (changes in behaviour, appearance of objects/entities triggering context shifts, abrupt actions not seen before etc). Thus, we don't believe that participants could form a coherent model of this particular movie's content without using the visual information present. We certainly accept that there may be other audio-visual stimuli where language information is more central to comprehension.

17. For the Bayesian analyses, what prior effect size distributions were assumed?

We used weak, noninformative priors over the model terms, similar to Majumdar et al. We have now added this to methods in the main text under Bayesian regression (of which ISC and Betas now have same model structure)

“Weak, noninformative priors were applied over the model terms.

$\beta_0 \sim \text{Student } t(3,0.3,2.5)$

$\beta \sim \text{Student } t(3,0,2.5)$

$\gamma_0 \sim \text{Student } t(3,0.3,2.5)$

$\gamma \sim \text{Student } t(3,0,2.5)$

$\epsilon \sim \text{Half Student } t(3,0,2.5)$ ”

Majumdar, G., Yazin, F., Banerjee, A. & Roy, D. Emotion dynamics as hierarchical Bayesian inference in time. *Cerebral Cortex* 33, 3750–3772 (2023).

18. In Fig 2, panel A is missing a color bar, while the color bar in panel C seems to be neither defined nor used. In panel B, the error bars are not defined.

We have added the color bars back to Fig 3 (in new version) and Fig 8 GLMs now, which uses the same scale now for easier comparison. We have defined error bars wherever applicable. For all the density plots, we used a single color bar to denote posterior evidence (0.5-1) with respect to 0 (in either direction). We agree this might not be informative for the ISC/ISFC plots (which are all >0) but it makes for consistency (and avoids confusion).

19. Discussion of links to consciousness, and to blind-sight, spatial neglect, and commissurotomy-related phenomena seem rather tangential.

We agree that these topics were not central to the specific research questions we are addressing. However, we believe making this link is important for placing the research in its broader context, particularly given our interest in the fragmentation vs. integration of experience.

20. The Cam-CAN dataset is usually treated as a lifespan cohort. It would be useful to state that only a subset of the full dataset was analyzed here, and mention the reason for choosing the age cut-off at 35 years. For this dataset, what was the voxel size and slice coverage?

We used 18-35 as a cut off for young healthy adults as used in (Majumdar et al), since this age range is commonly used in aging studies as a healthy young cohort, and as this is similar to the age range in the narrative dataset We have added a statement to this effect in the Methods.

Voxel size in CAMCAN dataset was 3x3x4.4mm with a thickness of 3.7mm and FOV 192 x 192 mm. We have now added this in methods

“Functional MRI scans were acquired on a 3T Siemens TIM Trio system, using a T2*-weighted multi-echo pulse sequence with a TR of 2470 ms, and multiple TEs of 9.4 ms, 21.2 ms, 33 ms, 45 ms, 57 ms, a flip angle of 78°, **voxel size of 3x3x4.4mm with a thickness of 3.7 mm, FoV = 192 x 192 mm, and 32 axial slices**”

47. Majumdar, G., Yazin, F., Banerjee, A. & Roy, D. Emotion dynamics as hierarchical Bayesian inference in time. *Cerebral Cortex* 33, 3750–3772 (2023).

21. When theta is defined in terms of SD from the mean, is this the mean across participants per time-point, or the mean across time for the group-average?

It is the mean across time for the group-average, and we have clarified our wording in the methods under Belief Updates.

Reviewer #4 (Remarks to the Author):

This study investigates the neural basis of predictive processing and integration in the brain, focusing on how different prefrontal cortex (PFC) regions and the Precuneus contribute to forming a unified subjective experience. I found this to be a creative study addressing an important question in cognitive neuroscience. However, the study suffers from several critical issues that significantly undermine the validity and strength of its conclusions. The authors frame their work within predictive coding and Bayesian inference models. However, the connections to these theoretical frameworks appear tenuous and poorly substantiated. The study's correlational nature precludes causal claims about predictive coding mechanisms, and the presented Bayesian inference models lack robust justification. More rigorous analyses, including direct measurements of prediction errors and clear temporal sequencing, are necessary to validate the proposed mechanisms and strengthen the claimed links to predictive coding, Bayesian inference frameworks, and cognitive maps.

The core proposition – that humans model the world by partitioning it into domains of states, agents, and actions, represented topographically in the PFC and integrated in the Precuneus – is intriguing but inadequately supported by the presented data. The evidence for integration in the Precuneus, a crucial aspect of their model, is not convincingly demonstrated and no rationale is provided for overall analysis approach is not clear or justified. The heavy reliance on correlational analyses fails to establish the causal relationships implied by their predictive coding framework.

We thank the Reviewer for their thorough review. In this revision, we have added a number of new analyses and substantially revised the text of the paper, in ways that we believe strengthen and

refine our claims. We respond fully to this Reviewer's specific concerns in the sections below. Before proceeding, we respond to the claim that our use of Bayesian inference models is not justified. We use Bayesian statistical models for hypothesis testing, in contrast to the frequentist approach that dominates in the field. To be clear, this choice does not indicate a theoretical position about how the brain functions (i.e., the so-called Bayesian brain theory), about which we remain agnostic. It simply reflects our preference for Bayesian statistical models as tools for testing hypotheses, the advantages of which have been articulated by various authors, e.g., van de Schoot et al. (2021), Keyzers et al (2020).

van de Schoot, R., Depaoli, S., King, R., Kramer, B., Märtens, K., Tadesse, M. G., ... & Yau, C. (2021). Bayesian statistics and modelling. *Nature Reviews Methods Primers*, 1(1), 1.

Keyzers, C., Gazzola, V. & Wagenmakers, E.J. Using Bayes factor hypothesis testing in neuroscience to establish evidence of absence. *Nat Neurosci* 23, 788–799 (2020).

Key Methodological Issues:

1. Prediction Error Analysis: The authors claim "Prediction-error-driven neural transitions... preceded subjective belief changes in a domain-specific manner." However, the study lacks any direct analysis or measurement of prediction errors, a fundamental omission given the centrality of this concept to their claims.

This is an important point. It's true that we did not directly sample participants' predictions while they watched the movie, so we cannot determine with certainty when prediction errors occurred. Instead, we asked raters to make a behavioural response whenever they updated their beliefs about the movie in one of the three domains of interest. We assume that such belief updates are the consequence of prediction errors – that the mismatch between the participant's current model of the movie and the incoming data trigger the model to be revised. By aggregating such data over a group of participants, we aimed to identify the points in the movie where such revisions were most likely to occur.

Although this is not a direct measurement of prediction error, it's not clear how one *could* directly measure prediction error during comprehension of a complex, dynamic naturalistic stimulus like a movie. For language stimuli, there are quantitative measures like surprisal which index the likelihood of a given word in a particular context. Even these methods do not directly measure what an individual participant has predicted at any given point, instead assuming that prediction errors occur whenever someone encounters a statistically improbable word. Computing statistics like this for natural audio-visual stimuli is very challenging. Given these issues, we believe our approach of simply asking people when their beliefs have changed is an appropriate one to take.

We do agree that these assumptions should be clearly stated in the paper so we have added the following clarifying assumption at the start of Results

"We assume that people signal a belief update whenever their existing model of the movie mismatches incoming information (i.e., when prediction errors occur)."

We also consider potential limitations of this approach in the Discussion.

2. Temporal Precedence: Despite assertions about neural transitions preceding subjective belief

changes, no specific analysis demonstrates this temporal relationship. The absence of clear temporal analysis to establish the sequence of neural transitions and subjective experiences is a significant limitation.

Several Reviewers commented on this. With hindsight, it's clear we overstated this aspect of our results. In fact, we did not intend to make any strong claims about the relative timing of behavioural and neural effects, and we believe fMRI data is not ideally suited to answering these specific questions about timing.

Accordingly, we have adjusted the wording in several places in the paper

We changed the wording in abstract from preceded to coincide with

“Prediction-error-driven neural transitions in these regions, indicative of model updates, coincide with subjective belief changes in a domain-specific manner. “

We changed the result title in Figure 4 (new version) similarly

“Neural Transitions Coincide with Subjectively Experienced Belief Updates”

We have now removed this sentence

“This suggests that the subjective experience of an update to predictions occurs sometime after the neural model is reconfigured.”

3. Event Concatenation: The concatenation of events across update segments is problematic, potentially introducing artifactual relationships in the data. It's unclear whether these events were correlated across the three types (states, agents, actions).

We have now investigated the effects of this in two ways.

For mitigating/controlling any artifactual relations in the data, we have applied phase randomization for the permutation tests in HMM and ISPCs (Fig 4,5 and 8 in the new version). We have also performed an extended window analysis (please see under Summary Section, Window analysis) for ISC and ISPC, two of our core analysis for domain-specific synchronization and integration profiles, which involve taking segments of time around update points. We have found largely consistent patterns for a range of window sizes, and have now put these results in the Supplementary Materials (Figures 10,14,23) for the reader.

In the original paper, we reported in the update timecourses for states, agents and actions ((States vs Agents, $r = 0.31$, Agents vs Actions, $r = 0.29$, Actions vs States, $r = 0.28$). In response to queries from Reviewer 2, we have also performed mutual information and cross-correlation analysis, which provide further evidence of weak relationships between domains. These results are presented in Supplementary Figure 2 & 3.

4. Replication and Generalizability: The replication results, particularly as shown in Figure 7, are weak and do not strongly support the authors' claims.

In response to Reviewer comments, we have re-run and expanded the analyses of the narrative dataset. Upon further scrutinizing the story we have found some technical changes to be necessary for the correct analysis.

As mentioned in the Summary section, upon deeper scrutiny we did find our GLM needed to be shifted by 3 TRS, and the ISPC required the raw ratings (which was mistakenly performed on smoothed ratings). . In the process of doing these, we identified an error in our analysis pipeline. As stated in the original Narrative dataset paper, "The stimulus file was started after 3 TRs (4.5 seconds) as indicated in the events.tsv files accompanying each scan." Our original analyses of these data did not take the 3 TR delay into account, hence the ratings were misaligned with the fMRI data by 4.5s. We have now correctly shifted the ratings accordingly (by 3 TRs).

Comparison of previous GLM (top) and new (below)in (left to right Actions Agents States)

The GLM (Figure 8a) shows stronger effects in vmPFC, amPFC and dmPFC for States, Agents and Actions respectively. The ISPC analysis (Fig 8b) now shows successful replication in of the effects for Action domain update with dmPFC-PCN correlations significantly higher amPFC/vmPFC.

We have also conducted a seed-based ISFC (PCN seed) on the story to further strengthen our claims of integration in the Precuneus. In the movie, PCN and Visual cortex was strongly coupled as shown in Fig 6a (new version). In an audio-only story, there should be weaker correlations with visual cortex here compared to the movie. We found this to be true across all domains (Fig 8c new version), which supports our that PCN integrates top-down predictions with the necessary external input.

The inconsistency in the number of update scans across domains (63 for States, 28 for Agents, 35 for Actions) raises questions about the comparability of these measures and the potential influence of estimation statistics.

It is true that we identified more update points for States than for the other two domains. To investigate whether it would be possible to use a different threshold that would identify more Agent and Action updates, we have tested a range of different thresholds. These new analyses are reported in the Method and Supplementary Figures 4 and 5. Although more lenient thresholds identify more update points, they also lead to greater temporal overlap in updates in different domains, and thus jeopardise our ability to discriminate between domains. Thus, we have retained our original approach. The difference in the numbers may simply reflect the fact the beliefs about the state change more frequently than those about the agents or actions in this particular movie.

We also note that the update profiles are somewhat different in the narrative dataset, yet this showed similar results in the PFC and core DMN (as shown in Figure 8).

5. Inter-Subject Correlation (ISC): The rationale for using ISC rather than activation to each event type is not clearly justified, especially considering that event detection might differ across individuals. This approach may obscure important individual differences in cognitive processing.

Thank you for raising this point. We do agree that the intersubject analyses we used might not be sensitive to individual differences in timing of their own updates.

This is one reason we binarized the update time-courses over a threshold. This approach removes weak and inconsistent updates ; even if some individuals update earlier or later, only the most prominent, shared updates survive thresholding.

That being said, we have added a sentence in the discussion highlighting this potential drawback of individual timing in the discussion

“However, it comes with the limitation that we do not have data on precisely when individual participants in the scanner experienced updates. This means that we were not able to investigate how the timing and nature of updates vary between individuals.”

6. Participant Consistency: It is unclear whether the measures of agent, state, and action updates are consistent between the participants who provided these ratings and those who underwent fMRI scanning, potentially introducing significant confounds. The reliability of ratings is weak and reliability of event segmentation in the second task is not shown.

Thank you for raising this concern. Since the robustness of the ratings are so critical to our main analysis and interpretation, we did conduct additional analysis of this.

We have performed cross-correlation between the ratings (for both movie and story) added this report in Supplementary Fig 3. We found that correlations with various time lags were always smaller than the basic unlagged correlations. Thus, the correlations we report in the main text

represent the upper bound on collinearity. We also conducted the mutual information Supplementary Fig 2 as it is a distinct metric of shared information, and it returned low values.

The reliability analysis (split-half correlation, now adjusted with Spearman-Brown correction) of the second task (narrative replication) is in Supplementary Fig 24b.

Other/Minor Issues:

Citations to key relevant manuscripts on DMN and PFC function are missing.

We have now added the following references as well as added a paragraph in discussion that contextualize the results within the broader literature spanning these works (bold for the new content).

A rich literature of cognitive^{12,13,18,47,50,51,53,65} and clinical studies^{40,41,66} supports the role of DMN in higher-order human cognition. Although classically seen as task-negative, this network is implicated in a variety of cognitive activities associated with subjectivity, such as mind-wandering, creative thought, self-related processing and mental time travel. These tasks are inferential in nature and possibly involve construction of rich internal models of experience. **Situating the DMN within the literature, one finds numerous links to episodic memory, which extends beyond the hippocampal spatial machinery from which it evolved, to encode more abstract temporal and referential properties⁶⁷. Any experience can potentially be represented as a structured set of states, forming a cognitive map⁶⁸ with distinct transitions between them. Re-experiencing an event involves traversing this state space, akin to path integration in the physical domains, reinstating both states and their transitions. These transitions might follow meaningful directional trajectories, shaped by multiple abstract factors such as time, and agent-specific information. The dmPFC and amPFC may underlie such roles during episodic encoding, sensitive to changes in predicted trajectories.** Indeed, episodic memories of experience are thought to be compartmentalized through event segmentations, which are functionally driven by prediction errors^{44,69,70}.

Situation models often require encoding object-space relational properties as schemas—generalized templates that facilitate rapid learning of similar contexts and prediction of novel contexts, a function linked to the vmPFC⁷¹. Reference frames are critical in social/self-processing⁷² and conceptual integration across viewpoints⁷³. Some knowledge is anchored to an external, world-centered (allocentric) reference frame, while others are encoded relative to the self (egocentric) or others. The ability to condense conceptual knowledge across various frames into task-centric reference frames may be a key function of the amPFC. Distinct cognitive maps emerge from different experiences, prompting inference across them in extended contexts. The dmPFC might then aid in traversing these different cognitive maps as observed in exploration³⁵, and in this context mind-wandering⁷⁴. Finally, during naturalistic experiences, the DMN often co-activates with not just the medial temporal lobe structures, but also the anterior temporal lobe, which represents various forms of semantic and conceptual content⁷⁵. It's an open question whether the distinct predictions that emerge in the PFC depend on its interaction with specialized representations of conceptual knowledge in this region.

The presentation of results could be improved, for instance, by including color scales in figures to enhance clarity.

We have added color scales to movie and story GLMs now, and with same bounds/thresholds and midline slices for easier comparison

Brain areas included for ROI analysis are not shown.

We have shown the ROI masks used (PFC and PCN) in the Supplementary Fig 1

Figure 1: Inter-rater reliability is shown as ISC.

Thanks for pointing out this error! We have now changed this to correlation in the y axis (Fig 1d)

Reviewer #5 (Remarks to the Author):

Thank you for the review.

I tried to run the experimental code, but it seems that the dependencies of the R libraries are incorrect. I installed all the libraries, but the code didn't work.

Thank for debugging the code. We've ran this on a different machine with libraries installed fresh, and the ones provided in the OSF code should be enough to reproduce all the analysis/plots. It seems to be working on RStudio 2023.06.1 & 2023.12.1 versions as well. We've added the new updated code for some analysis now so please let us know which ones are not running currently.

Reviewer comments are in Black. Our responses are in Blue. Changes/quotes from the revised manuscript are in Red. Quotes from other references are in Green where applicable.

REVIEWER COMMENTS

Reviewer #1 (Remarks to the Author):

I appreciate the authors' detailed and thoughtful responses to the previous comments from all reviewers. The revised manuscript shows substantial improvements in both methodological clarity and interpretational precision. The additional analyses, expanded explanations, and careful revisions indicate that the authors have taken the feedback seriously and addressed each point with considerable care. I sincerely appreciate the time and effort the authors have invested in thoroughly responding to all of the comments.

Thank you for the fantastic review throughout and kind words.

However, I believe several important concerns remain insufficiently addressed. Below, I outline the key issues that, in my view, require further revision before the manuscript can be considered for acceptance.

Major points

1-2. The authors argue that the use of a 2 SD threshold strikes a balance between the number of belief update moments and the degree of temporal overlap across domains. While I appreciate the attempt to justify this parameter choice, the supplementary figures provided indicate that overlap across domains persists even at the 2 SD level (~25%). This raises a concern about whether the belief updates for each domain can truly be considered distinct, as the authors claim.

Given this, I believe additional analyses are needed to further validate the domain specificity of belief updates. First, the authors should perform analyses that exclude timepoints that overlap between domains and show whether their main findings still hold under this more conservative condition.

Thanks for raising this point in more scrutiny. We have now re-run our main movie analyses after excluding update points that weren't specific to a single domain. As shown below (original analyses on left; after exclusions on right), the major differences between domains remain under the more conservative approach. In the paper, we continue to report the data using the original approach, which we believe is preferable because (a) it makes use of all of the available data, which is important given the limited data available (~8mins) for each participant and (b) it better reflects the reality of natural experiences, in which changes to state, agent and action beliefs occur in a correlated fashion and cannot be fully separated from one another.

However, we do note this important issue in the Discussion: "In future works it will be important to exert more experimental control over the nature and timing of such processes, in order to validate our findings. In particular, updates to states, agents and actions were not fully independent of one another and some plot developments seemed to result in multiple domains updating simultaneously."

This presumably reflects the fact that unfolding events often provide information relevant to more than one domain. This dependence could potentially be avoided in future studies by using artificially-generated stimuli, though this would come at the cost of reduced ecological validity.”

This is the original HMMs (left) vs unique time points only (where no overlapping times occur with $\sim 4TR$ gap between different domain updates), under the same threshold ($2SD$) (right)

Qualitatively, the domain-specificity of each PFCs were conserved even under more specific updates.

Similarly, this is the main ISPC result (left) vs using only the Unique time points (right)

The trend of relative specificity of PCN to each of the PFC during domain update remained same.

Second, it would be important to examine whether the main results replicate when using alternative thresholds, such as 1.6 SD. Based on the figures provided, the 1.6 SD threshold appears to yield more distinct domain-specific update patterns than the 2 SD threshold. If the key findings are sensitive to the choice of threshold, this should be acknowledged and directly addressed in the manuscript.

We compared these results under 1.6SD threshold as the Reviewer suggested (left original, right 1.6SD)

The HMM remained largely similar to the main results under this threshold. The PFC-Precuneus ISPC correlations were also similar, except that the correlation during State updates was similar in strength for vmPFC and amPFC. This may be because the less stringent threshold resulted in greater overlap between State and Agent update points.

We point this out in the Methods now

“Nevertheless, there were some occasions when updates in two domains overlapped in time (approximately 25% of identified updates). We repeated our main analyses after excluding updates that overlapped over domains, which gave similar results to those we report in the paper (based on all updates).”

1-4. While I appreciate the authors' effort to test the stability of the ISC and ISPC results across multiple window sizes (Figure 3C, Figure 5D, Supplementary Figures 10 and 14), I remain concerned regarding the robustness of the findings. Specifically, the domain-specific effects highlighted in the manuscript appear to be most consistent at a 7-TR (or now 9-TR?) window. At shorter or longer window sizes, the effects noticeably weaken or become inconsistent, raising questions about whether these findings reflect a stable phenomenon or are dependent on specific analytical choices.

Thank you for this point. We observed the same qualitative pattern of results in all domains for all window sizes between 4TR and 10TR (e.g., for Agents, the amPFC correlation was higher than vmPFC and dmPFC correlations across all these window sizes). We agree that effects weaken somewhat towards the extremes of this range and don't always hold beyond 10TRs. However, we believe we have demonstrated that our findings are stable across a reasonable range of parameters. Indeed, one could reasonably expect patterns to break down outside this range of window lengths. At very short window sizes, the small number of observations make it difficult to observe reliable effects. At longer durations beyond 10TR (10TR = 25s) the window will include neural data that occur some time after the update and are unlikely to be related to it. We do however mention on this point explicitly now in the manuscript.

“We consider reasons for this in the Discussion...The same overall pattern of results was observed for windows of length 4 to 10 TRs, before dissipating when the window length exceeded 10 TRs. It should be emphasized that for the intersubject analyses, at very short windows, the small number of observations could lead noise to drive the results while at very long windows, the window will include neural data that occur some time after the update and are unlikely to be related to it. Updates themselves are somewhat transient events but our data suggest that their effects on neural patterns persist for some time after they occur.”

The GLM analyses already show strong domain-specific effects using continuous update ratings—suggesting that the inclusion of ISC/ISPC results, particularly given their window-size sensitivity, may not substantially strengthen the manuscript's main claims. To clarify the contribution of these results, it would be helpful for the authors to articulate what unique insights ISC/ISPC analyses offer beyond the GLM findings and to address the extent to which these effects can be considered robust across a range of analytical choices.

We have addressed this comment by adding a new paragraph to the beginning of the Results section. This outlines the various analytical approaches we have taken, what questions they were designed to test and what information the HMM/ISC approaches provide beyond the initial GLM results (see response to point 6 below).

3. While the authors provided additional analyses addressing domain specificity of PFC–precuneus coupling, the question of whether this coupling is specific to belief updating periods remains unresolved. The authors chose not to compare update and non-update periods directly, citing an imbalance in the number of timepoints. This concern could be addressed by randomly sampling non-

update timepoints to match the number of update periods, which would help clarify whether the observed coupling is update-specific or instead reflects more general characteristics of the data.

Thanks for this comment and for the great suggestion of subsampling the non-update times to avoid data imbalances.

We conducted a way to sample the nonupdated points as follows.

For each domain rating, we took the lowest values under two constraints

- The first constraint is that, within 1 TRs in front or back (~5s), there should not be any updates from any ratings. Update here is defined as the same update we took elsewhere i.e. thresholded at 2SD. This ensures the non update time points are not only of lowest update values but also never overlapping close to the main updates

- The second constraint was to keep sampling these points until the number of nonupdated time points is matched with the update time points, for each domain.

This process ensured a fair comparison in each domain, between its most likely updates across participants with the most likely nonupdated events.

We then submitted these values into a Bayesian hierarchical regression (same structure we used in the main) and estimated posterior evidence of update vs nonupdated ISFC values between PCN and the domain-specific PFC.

We find that States had highest difference in Update times having more ISFC values than NonUpdate points (Update>Nonupdate $BF_{for} >1000$, $P = 0.99$). Followed by Actions (Update>Nonupdate $BF_{for} = 262.16$, $P = 0.99$), and less by Agents (Update>Nonupdate $BF_{for} = 4$, $P = 0.80$). Overall, there was more evidence for Update events having more coupling within DMN than non-update points.

We have added this to the main Figure 5 as well as its needed information in methods and the Supplementary.

4. In both Figure 5D and Figure 8B, ISPC correlations during Action updates are consistently negative across PFC regions, with the dmPFC showing values that are less negative or closer to zero. The authors interpret this as evidence of domain-specific coupling between the dmPFC and the precuneus. However, negative ISPC values are inherently difficult to interpret, and the manuscript does not clearly explain what such negative (or less negative) correlations mean, either mechanistically or functionally. If these values are intended to reflect desynchronization, the authors should clarify what desynchronization entails in this context and how it specifically relates to Action updates and belief updating more broadly.

A review of relevant literature documenting similar patterns and proposing plausible neural mechanisms that might underlie this phenomenon would strengthen the claim about dmPFC's selective role in Action updates.

We agree this aspect of our results is difficult to interpret. We have reflected on the potential explanation for this.

Extensive studies by various groups have shown functional specialization in its subregions. In our GLM analyses, we found that the dorsal part of Precuneus, rather the whole, was more consistently activated during action updates in both movie and story GLMs. This specialisation could potentially explain why action updates did not correlate with ISPC values derived from the Precuneus as a whole.

We have added a sentence on this in the Discussion for the interested reader.

We found that ISPC in each PFC region was most correlated with the Precuneus ISPC during updates in its preferred domain, suggesting that the Precuneus plays a general integratory role. However, while these correlations were positive for vmPFC and amPFC, the Precuneus ISPC was negatively correlated with dmPFC during action updates. This negative effect suggests that the neural correlates of action integration may be somewhat different to the other domains. Multiple studies suggest functional specialization within the Precuneus, with dorsal and ventral parts showing distinct connectivity profiles. In line with this, we found that dorsal Precuneus was particularly engaged by action updates, while ventral Precuneus was more activated by agent or state updates (see Fig 3a and 8a). Thus, there may be second-order specialization within the Precuneus for integration of different elements of the world model, which may explain why action updates showed weaker correlations with the Precuneus as a whole. Future works using functional-gradient methods and fine-grained parcellation could more precisely map these subregional networks, providing deeper insights into the role of DMN in world modelling.

Zhang, Y. *et al.* Connectivity-Based Parcellation of the Human Posteromedial Cortex. *Cerebral Cortex* **24**, 719–727 (2014).

Cunningham, S. I., Tomasi, D. & Volkow, N. D. Structural and functional connectivity of the precuneus and thalamus to the default mode network. *Human Brain Mapping* **38**, 938–956 (2017).

5. The authors argue that the absence of precuneus–visual cortex ISFC in the story-based dataset supports that the precuneus integrates updated representations with sensory regions. While this finding is informative, I believe this conclusion remains incomplete without considering the auditory modality. Because the story dataset is presented entirely through auditory input, one might expect to see increased coupling between the precuneus and the auditory cortex. However, no such effect is reported or discussed.

As a result, the current pattern of findings appears to be specific to precuneus–visual cortex connectivity rather than broadly supporting the idea that the precuneus integrates with sensory regions. If the authors wish to maintain this broader interpretation, I believe it is necessary to show that such integration is flexible across modalities or revise their claim to reflect the more specific nature of the observed effect.

We agree that this is an important point and have made two main changes to the paper in response.

1. At key points in the text, we have clarified that our results are specific to visual sensory integration
2. We have explicitly acknowledged the lack of coupling with auditory cortex in the story data and have provided a possible explanation for this: “We did not observe significant coupling with auditory regions of the superior temporal lobe. This may be because when listening to speech, the auditory signal itself carries little information that can directly influence the world model. The speech signal must first be transformed into lexical-semantic representations. This process involves parts of the ventral and lateral temporal cortex that did show coupling with the Precuneus.”

6. While I now better understand the rationale for employing multiple intersubject analyses such as ISC, ISPC, and sISC, this rationale is currently presented in a fragmented manner throughout the manuscript. Given the diversity of analytical approaches used, it would greatly benefit the reader to include a brief overview—perhaps in the Methods section or at the beginning of each finding—explaining why each method was chosen and what kind of hypothesis or neural property it is intended to capture. This would enhance readability and help readers more easily navigate the analyses, while also highlighting the unique strengths of each approach.

Thank you for this suggestion. As replied to 1-4, we have now edited Figure 2 and added a prior for the analyses at the beginning section of result.

We performed a number of analyses designed to identify the neural correlates of belief updates in the domains of states, agents and actions, and to investigate the within- and between-region neural dynamics associated with them. Each of these analyses is described in more detail later, but we begin with a general overview. First, we used update probability time-courses as predictors of BOLD response, to identify regions that show heightened activity during updates of each type (Fig 2b). Next, we isolated the most significant update moments in each domain by thresholding the update time-courses (Fig 2c). We used Hidden Markov Modelling to test whether these domain update moments coincide with transitions in the neural states of specific PFC regions. This analysis tested whether the transient activity increases observed during model updates were associated with sustained shifts in the activation states of the region, specific to each domain. We also performed a number of analyses

based on inter-subject correlations (ISC; Fig 2D). The rationale of ISC analyses is that when multiple participants are scanned while experiencing the same stimulus, neural responses that are consistent (i.e., correlated) across participants reflect reliable brain responses to the content of the stimuli. We tested ISC in the activity of PFC regions to determine whether correlation increased during a region's preferred updates, indicating heightened stimulus-driven activity. We used a related method, inter-subject functional connectivity, to investigate whether stimulus-driven PFC-Precuneus activity correlations were increased during belief updates, compared to other regions and non-updates. And finally, we used inter-subject pattern correlations to quantify the alignment of multi-voxel activation patterns between participants and test the degree to which these pattern time-courses of PFC and Precuneus displayed representational alignment at similar times.

Minor points

While most of the minor points have been addressed, some issues would still benefit from additional clarification.

2. I would like to raise a potential concern regarding the interpretability of the correlation between PFC–precuneus ISFC and precuneus ISC. Because both measures are derived from activity within the precuneus, the observed relationship may reflect intrinsic properties of this region—such as signal stability or general responsiveness—rather than intersubject convergence in belief updating. If the authors intend to use this correlation to support claims about representational integration, alternative explanations should be ruled out or explicitly acknowledged.

Furthermore, the authors suggest that individuals with stronger or more similar “integration profiles” with PFC also exhibit more similar precuneus activity across participants. However, it is not clearly explained why the degree of functional coupling should necessarily translate into shared processing or subjective experience at the representational level. If this correlation is intended to support claims about shared belief updating or predictive coding, I encourage the authors to clarify this interpretational link more directly.

The reviewer is correct that in this analysis, Precuneus activity contributes to both of the measures being correlated which makes it difficult to link the effect specifically to belief updating. Our original analysis showed a strong correlation between participants' whole-movie PCN ISC and their PCN-PFC ISFC during model update times. We have now repeated this analysis with PCN-PFC when no updates are occurring. As shown below, there is no significant correlation. Since the correlation is specific to update periods, we interpret this as evidence for integration of belief updates specifically between PFC and Precuneus. The degree of this coupling is correlated with the whole movie Precuneus ISC, a measure of the movie experience (Arousal), only during the updates, implying the integration profile of a participant during updates specifically is associated with their general movie experience. Put another way, how participants' beliefs get updated during certain events is associated with their subjective experience.

We found that their association was more during updates than nonupdates (these are now in the new Fig 7). Furthermore, we also compared to each PFC ROI separately and found this to be consistent. These are now in the new Supplementary Fig 21.

Reviewer #2 (Remarks to the Author):

The authors successfully addressed all the issues and points I asked for during the first review of the article. I'm happy with the new version and I want to congratulate all the authors for the tremendous amount of work!

We greatly appreciate Reviewer 2's constructive suggestions and enthusiasm for our work throughout the review process.

Reviewer #3 (Remarks to the Author):

The authors have thoughtfully and adequately addressed my previous concerns and queries. I am now left with four relatively minor comments:

Thank you for the amazing review throughout and kind words.

1. Error bars still appear to be missing from some bars in supplementary figures 15-19, but I now wonder whether this is because the line width is too thin for them to render consistently?

We suspect this too since on some devices the same code is rendered with thinner error bars. We have now added thicker bars wherever applicable.

2. Supplementary figure 10 is redundant, because all information is also present in supplementary figure 22.

Agreed and removed.

3. My previous comment 9 concerned the interpretation of functional connectivity between precuneus and frontal regions, but I realise that the same point applies to functional connectivity between precuneus and visual cortex in the section beginning on line 455. I.e. I don't think one can conclude "integration of representations" from what are merely correlated time-courses. Although I wonder why the more interesting ISPC method used for frontal connectivity was not applied here, I feel it would be sufficient just to tone-down the claims in the section heading. (Similar claims within the text of the section are already expressed sufficiently cautiously.)

We have now changed this and toned down the result title.

Update-driven coupling of Precuneus with Visual Regions

4. The most substantive comment concerns the new analysis of precuneus functional connectivity in the replication dataset. One claim of the paper is that “the precuneus integrates updated representations with sensory regions,” with evidence including functional connectivity between precuneus and visual regions (previous comment notwithstanding). The new analysis in Figure 8c usefully shows that that such connectivity is weaker for the audio-only narrative. However, the general hypothesis surely predicts connectivity between precuneus and auditory regions in this case, which is not apparent? Should the conclusion, then, be that precuneus is specifically coupled with visual regions, rather than sensory regions more generally? I feel this deserves at least some acknowledgement/consideration in the results/discussion.

Thanks for this comment. Reviewer 1 noted this as well. We agree that this is an important point and have made two main changes to the paper in response.

1. At key points in the text, we have clarified that our results are specific to visual sensory integration
2. We have explicitly acknowledged the lack of coupling with auditory cortex in the story data and have provided a possible explanation for this: “We did not observe significant coupling with auditory regions of the superior temporal lobe. This may be because when listening to speech, the auditory signal itself carries little information that can directly influence the world model. The speech signal must first be transformed into lexical-semantic representations. This process involves parts of the ventral and lateral temporal cortex that did show coupling with the Precuneus.”

Reviewer #4 (Remarks to the Author):

The authors have done a great job of addressing reviewer concerns, including mine. I think a limitations section would be useful for the reader, given the substantive conceptual issues raised and the clarifications the authors have provided (e.g. prediction error implied from belief update etc). In parallel, readers will benefit from the reviewer response so the strengths/weaknesses and issues raised can advance future research.

Thank you for the solid review and kind words. We have now added these in the limitation sections in discussion to inform the reader on current shortcomings and future opportunities.

“In particular, updates to states, agents and actions were not fully independent of one another and some plot developments seemed to result in multiple domains updating simultaneously. This presumably reflects the fact that unfolding events often provide information relevant to more than one domain. This dependence could potentially be avoided in future studies by using artificially-generated stimuli, though this would come at the cost of reduced ecological validity

We found that ISPC in each PFC region was most correlated with the Precuneus ISPC during updates in its preferred domain, suggesting that the Precuneus plays a general integratory role. However, while these correlations were positive for vmPFC and amPFC, the Precuneus ISPC was negatively correlated with dmPFC during action updates. This negative effect suggests that the neural correlates of action integration may be somewhat different to the other domains. Multiple studies suggest functional specialization within the Precuneus, with dorsal and ventral parts showing distinct connectivity profiles. In line with this, we found that dorsal Precuneus was particularly engaged by action updates,

while ventral Precuneus was more activated by agent or state updates (see Fig 3a and 8a). Thus, there may be second-order specialization within the Precuneus for integration of different elements of the world model, which may explain why action updates showed weaker correlations with the Precuneus as a whole. Future works using functional-gradient methods and fine-grained parcellation could more precisely map these subregional networks, providing deeper insights into the role of DMN in world modelling. “

Reviewer #5 (Remarks to the Author):

While I did not run the full analysis code, it appears well-structured and reasonably documented.

Thank you for the review!

Reviewer comments are in Black. Our responses are in Blue.

REVIEWER COMMENTS

Reviewer #1 (Remarks to the Author):

The authors have adequately addressed my concerns, and the manuscript has been strengthened.

As the entire analysis code currently runs in a single block, I recommend partitioning the code into distinct sections by analysis step to facilitate individual review of each result and enhance usability.

Thank you for the fantastic review throughout!